# Glycogen synthase GYS1 overactivation contributes to glycogen insolubility and malto-oligoglucan-associated neurodegenerative disease

Silvia Nitschke [iD][1], Alina P Montalbano [iD][1], Megan E Whiting[1], Brandon H Smith[1,8], Neije Mukherjee-Roy[1], Charlotte R Marchioni[1,9], Mitchell A Sullivan[2,3], Xiaochu Zhao[4], Peixiang Wang [iD][4], Howard Mount [iD][5], Mayank Verma [iD][1], Berge A Minassian [iD][1,6✉] & Felix Nitschke [iD][1,6,7✉]

## Abstract

Polyglucosans are glycogen molecules with overlong chains, which are hyperphosphorylated in the neurodegenerative Lafora disease (LD). Brain polyglucosan bodies (PBs) cause fatal neurodegenerative diseases including Lafora disease and adult polyglucosan body disease (APBD), for which treatments, biomarkers, and good understanding of their pathogenesis are currently missing. Mutations in the genes for the phosphatase laforin or the E3 ubiquitin ligase malin can cause LD. By depleting PTG, an activator of the glycogen chain-elongating enzyme glycogen synthase (GYS1), in laforin- and malin-deficient LD mice, we show that abnormal glycogen chain lengths and not hyperphosphorylation underlie polyglucosan formation, and that polyglucosan bodies induce neuroinflammation. We provide evidence indicating that a small pool of overactive GYS1 contributes to glycogen insolubility in LD and APBD. In contrast to previous findings, metabolomics experiments using in situ-fixed brains reveal only modest metabolic changes in laforin-deficient mice. These changes are not replicated in malin-deficient or APBD mice, and are not normalized in rescued LD mice. Finally, we identify a pool of metabolically volatile malto-oligoglucans as a polyglucosan body- and neuroinflammation-associated brain energy source, and promising candidate biomarkers for LD and APBD, including malto-oligoglucans and the neurodegeneration marker CHI3L1/YKL40.

**Keywords** Lafora Disease; Adult Polyglucosan Body Disease; Neuroinflammation; Biomarkers; Glycogen
**Subject Categories** Metabolism; Molecular Biology of Disease; Neuroscience

## Introduction

Glycogen, a branched glucose polymer of up to 60,000 glucose units and $10^4$ kDa in size, is synthesized in various cell types, primarily serving as energy storage with important roles in diabetes, cancer, muscle physiology, and the brain (Dauer and Lengyel, 2019; Hargreaves and Spriet, 2020; Jensen et al, 2011; Markussen et al, 2023; Sullivan and Forbes, 2019). Glycogen water solubility is a prerequisite for glycogen-metabolizing enzymes to access their substrate. The formation of insoluble glycogen, which gradually aggregates into polyglucosan bodies (PBs), is associated with disease (Roach et al, 2012). Lafora disease (LD, OMIM #254780) and adult polyglucosan body disease (APBD, OMIM #263570) are examples of autosomal recessive genetic neurological diseases caused by PB formation in the nervous system. LD, manifesting as childhood-onset progressive myoclonus neurodegenerative epilepsy (Minassian, 2001), is caused by loss-of-function mutations in *EPM2A or NHLRC1* encoding glycogen phosphatase laforin and E3 ubiquitin ligase malin, respectively. APBD, an adult-onset leukodystrophy characterized by neurogenic bladder, progressive spastic gait, and peripheral neuropathy (Mochel et al, 2012), is caused by mutations in *GBE1*, leading to reduced glycogen branching enzyme activity. To date, no curative treatments or established clinical biomarkers are available for these diseases.

In LD and APBD mouse models, skeletal muscle PB glycogen is poorly branched, while soluble glycogen exhibits normal branching and is unchanged in quantity (Sullivan et al, 2019). The coexistence of structurally different soluble and insoluble glycogen suggested that at any given time, few glycogen molecules acquire branching defects, which drive precipitation, aggregation, and accumulation of these particular molecules to over time exceed the levels of normal soluble glycogen (Sullivan et al, 2017; Sullivan et al, 2019). Glycogen synthesis requires glycogen synthase (GYS1 in brain and muscle) for chain elongation with UDP glucose as substrate

[1]Division of Neurology, Department of Pediatrics, University of Texas Southwestern Medical Center, Dallas, TX 75390, USA. [2]Glycation and Diabetes Complications, Mater Research Institute – The University of Queensland, Translational Research Institute, Brisbane, QLD 4102, Australia. [3]School of Health, University of the Sunshine Coast, Sippy Downs, QLD 4556, Australia. [4]Program in Genetics and Genome Biology, The Hospital for Sick Children Research Institute, Toronto, ON M5G 0A4, Canada. [5]Tanz Centre for Research in Neurodegenerative Diseases, Departments of Psychiatry and Physiology, University of Toronto, Toronto, ON M5T 0S8, Canada. [6]O'Donnell Brain Institute, University of Texas Southwestern Medical Center, Dallas, TX 75390, USA. [7]Department of Biochemistry, University of Texas Southwestern Medical Center, Dallas, TX 75390, USA. [8]Present address: Department of Neuroscience, University of Texas Southwestern Medical Center, Dallas, TX 75390, USA. [9]Present address: Biochemistry and Molecular Genetics Department, University of Colorado School of Medicine, Aurora, CO 80045, USA. ✉E-mail: berge.minassian@utsouthwestern.edu; felix.nitschke@utsouthwestern.edu

and glycogen branching enzyme (GBE1), which introduces branch points by cleaving off α1,4-linked glycogen chains and reattaching them via α1,6-linkages. The action of GYS1 and GBE1 determines chain length, i.e. the glycogen branching structure. An imbalance between the activity of both enzymes could lead to glycogen insolubility via the formation of abnormally long chains that tend to form α-helices, which extrude water and promote precipitation (Damager et al, 2010).

It has been hypothesized that glycogen hyperphosphorylation underlies glycogen insolubility in LD because laforin possesses glycogen phosphatase activity, and LD glycogen is hyperphosphorylated (Tagliabracci et al, 2008). Yet, phosphatase-inactive laforin prevents PB formation despite persistent glycogen hyperphosphorylation (Gayarre et al, 2014; Nitschke et al, 2017). Furthermore, APBD mice strongly accumulate insoluble glycogen due to insufficient branching but without glycogen hyperphosphorylation (Sullivan et al, 2019). Hence, a causal relationship between insoluble glycogen and glycogen hyperphosphorylation has been questioned, and the role of glycogen phosphate remains unclear.

While abnormal glycogen branching in APBD is directly linked to GBE1 deficiency (Akman et al, 2015; Sullivan et al, 2019), the reason for abnormal glycogen branching and PB accumulation in LD is unclear. Studies show that reducing GYS1 activity decreases PB accumulation in LD and APBD mice. The knockout of PTG (encoded by *Ppp1r3c*) completely prevents PB formation in LD mice (Turnbull et al, 2011; Turnbull et al, 2014) and partially in APBD mice (Chown et al, 2020). PTG is a non-catalytic subunit of protein phosphatase 1 (PP1), which targets PP1 to glycogen, where PP1 activates GYS1 by dephosphorylation (Roach et al, 2012). PB prevention by PTG knockout in LD suggests that in LD, PBs are caused by dysregulation of glycogen synthesis, which is counteracted when GYS1 activity is reduced. However, the effect of PTG knockout on glycogen structure is unknown, and in LD no differences in expression and activity of glycogen-synthesizing enzymes were detected (DePaoli-Roach et al, 2010; Sullivan et al, 2019; Tagliabracci et al, 2008; Tagliabracci et al, 2007; Tiberia et al, 2012; Valles-Ortega et al, 2011).

PTG and GYS1 knockout also prevent other characteristics in LD mice, i.e. increased seizure susceptibility, myoclonus, and neuroinflammation (Pederson et al, 2013; Turnbull et al, 2011; Turnbull et al, 2014), while transgenically increased GYS1 activity induces deposition of insoluble glycogen, neuroinflammation, and autophagy dysregulation (Duran et al, 2014). Additional phenotypes demonstrated in LD include glycogen hyperphosphorylation (Nitschke et al, 2013; Tagliabracci et al, 2008) and abnormal metabolism (Brewer et al, 2019; Duran et al, 2021). It is unclear whether the latter two phenotypes are causative of or secondary to PB accumulation, and how they relate to LD-associated neuroinflammation. By modulating GYS1 activity through PTG knockout, these phenotypes can be studied in the presence and absence of PBs.

Analyses of glycogen metabolism and any associated metabolic processes in the brain are confounded by rapid postmortem glycogen degradation (Ponten et al, 1973; Wu et al, 2019). Postmortem changes in brain metabolism can be avoided by using high-power focused microwave (FM) fixation systems that inactivate all enzymatic activity in the brain within approximately one second. Using FM, 4-15-fold higher mouse brain glycogen levels were measured (Oe et al, 2016) compared to brains dissected after cervical dislocation and immediate cryopreservation (CP) in liquid nitrogen (DePaoli-Roach et al, 2010; Duran et al, 2014; Nitschke et al, 2017; Tagliabracci et al, 2008). Recently, the comparison of metabolomes from CP- and FM-fixed mouse brain tissue showed that postmortem, several metabolites and associated pathways change within seconds, with drastic changes observed in glucose, glucose-6-phosphate, and glycogen pools (Juras et al, 2023). FM brain fixation also preserves global protein phosphorylation (O'Callaghan and Sriram, 2004), which is otherwise susceptible to rapid postmortem changes as shown for several phosphoproteins (Fernandes and Li, 2017; Scharf et al, 2008; Wang et al, 2015). Note that the rate-limiting enzymes of glycogen synthesis and degradation, GYS and glycogen phosphorylase, respectively, are modulated allosterically as well as through phosphorylation. GYS1 phosphorylation leads to inactivation, while dephosphorylation activates GYS1 (Roach et al, 2012). Allosteric binding of glucose-6-phosphate (G6P) overrides the phosphorylation-imposed inhibition, restoring full GYS1 activity (Bouskila et al, 2010; Hunter et al, 2015). Despite the clear evidence that increased glycogen chain length is the cause of glycogen insolubility (Nitschke et al, 2017; Sullivan et al, 2019), it is unclear what causes the abnormal glycogen structure in LD as no differences in the expression or activity of glycogen-metabolizing enzymes could be found to date. FM brain fixation prevents postmortem processes, which could alter the metabolic state and/or enzyme activities potentially eliminating the disease-related changes that caused the shift in chain length. Therefore, brain glycogen levels, GYS1 phosphorylation, and the LD brain metabolome need to be revisited in FM-fixed mouse brains.

In this study, the effect of PTG knockout on glycogen structure and phosphate, as well as on the brain metabolome and neuroinflammatory markers, was studied in the context of two LD mouse models. For comparison, the Gbe1YS APBD mouse model was included, which accumulates PBs but for a different genetic and biochemical reason than in LD. Microwave-fixation of the brain was employed to enable in situ snapshots of brain glycogen and metabolome as well as the phosphorylation-dependent activation state of GYS1. In conjunction with the study of disease-related mRNA expression changes, the roles of several previously described molecular hallmarks of PB diseases were clarified, and new ones were identified. This includes the discovery of abundant malto-oligoglucans accumulating in the brains of PB disease models. We define malto-oligoglucans as chains of glucose units that are distinct from soluble macromolecular glycogen as they are much smaller and evidently metabolized faster. Malto-oligoglucan metabolism is prominent in bacteria (Gänzle and Follador, 2012) but is here demonstrated to play a role in the context of neurodegenerative disease.

## Results

### Microwave-fixation protects brain glycogen from postmortem degradation

A previous study has shown unexpectedly high brain glycogen levels after FM fixation (Oe et al, 2016), therefore we directly compared glycogen levels from CP- and FM-fixed mouse brains from non-disease mice (LD heterozygous). Total glycogen levels

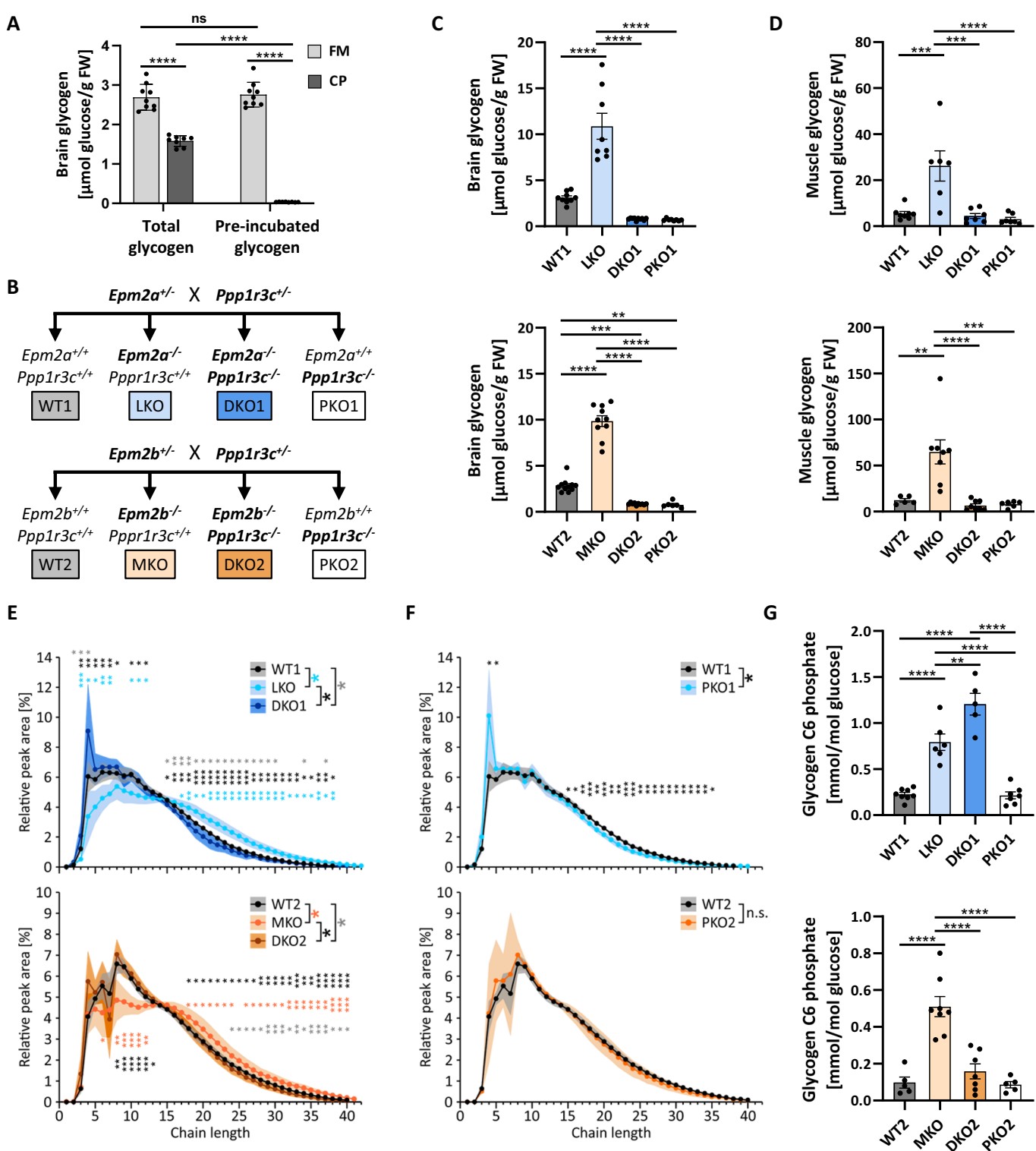

were ~40% lower in CP-fixed brains. While tissue preincubation at RT resulted in complete depletion of glycogen after cryopreservation, there was complete recovery of glycogen in FM-fixed brains (Fig. 1A). Hence, FM fixation successfully deactivated glycogen-degrading enzymes, as indicated by preincubation stability and 1.6-fold increased total glycogen content compared to CP.

## PTG knockout prevents glycogen accumulation in LD brain and muscle and impacts glycogen chain length and phosphorylation

To study the role of PBs in LD pathogenesis as well as their impact on previously described molecular disease characteristics, we generated

**Figure 1. PTG knockout prevents LB formation in the brain and muscle of LD mice and impacts glycogen chain length and phosphorylation.**

(A) Brain total and preincubated glycogen in CP- and FM-fixed brains of heterozygous non-disease mice. $n = 8$-9. (B) Overview of experimental genotypes and respective names and color codes used throughout the study. (C, D) Brain (C) and muscle (D) total glycogen following FM fixation in laforin and malin cohort, respectively. $n = 6$-11 (C), $n = 5$-8 (D). (E, F) Chain length distributions (CLDs) of total muscle glycogen from LD and DKO (E) and PKO mice (F) compared to WT. $n = 5$-8. (G) Muscle total glycogen C6 phosphate in both, laforin and malin, cohorts. $n = 4$-8. Data information: Data were presented as mean ± SEM (A–D, G) or ±SD (E, F). *$p < 0.05$; **$p < 0.01$; ***$p < 0.001$; ****$p < 0.0001$ by two-way ANOVA with Tukey post hoc analysis (A–D, G) or by Welch's $t$-test (E, F); ns not significant, CP cryopreservation, FM focused microwave; n signifies biological replication of measurements in different animals of the same group. P values: (A) FM (Tot. vs. Pre) 0.844, CP (Tot. vs. Pre) 9.55E-12, Tot. (FM vs. CP) 9.56E-12, Pre (FM vs. CP) 9.55E-12; (C) WT1 vs. LKO 1.3E-08, LKO vs. DKO1 1.7E-11, LKO vs. PKO1 4.9E-11, WT2 vs. MKO 5.6E-14, MKO vs. DKO2 5.5E-14, MKO vs. PKO2 5.5E-14, WT2 vs. DKO2 0.00063, WT2 vs. PKO2 0.0022; (D) WT1 vs. LKO 0.00024, LKO vs. DKO1 0.00019, LKO vs. PKO1 8.7E-05, WT2 vs. MKO 0.0011, MKO vs. DKO2 6.2E-05, MKO vs. PKO2 0.00023; (E) Chain lengths: WT1 vs. LKO (3 - 5.08E-05, 4 - 0.02, 5 - 0.016, 6 - 0.007, 7 - 0.0037, 10 - 0.014, 11 - 0.013, 12 - 0.033, 17 - 0.015, 18 - 0.0089, 19 - 0.011, 20 - 0.012, 21 - 0.0088, 22 - 0.0067, 23 - 0.0055, 24 - 0.0044, 25 - 0.0046, 26 - 0.0052, 27 - 0.0065, 28 - 0.0069, 29 - 0.0072, 30 - 0.0076, 31 - 0.0085, 32 - 0.011, 33 - 0.012, 34 - 0.013, 35 - 0.015, 36 - 0.0096, 37 - 0.013, 38 - 0.0095), WT1 vs. DKO1 (2 - 0.0196, 3 - 0.0249, 4 - 0.0218, 15 - 0.0152, 16 - 0.0069, 17 - 0.00521, 18 - 0.00979, 19 - 0.016, 20 - 0.0119, 21 - 0.0139, 22 - 0.0184, 23 - 0.0257, 24 - 0.0334, 25 - 0.0321, 26 - 0.0291, 27 - 0.0292, 28 - 0.0404, 29 - 0.0334, 30 - 0.0499, 34 - 0.0366, 37 - 0.0325), LKO vs. DKO1 (3 - 0.00455, 4 - 0.0038, 5 - 0.00483, 6 - 0.00409, 7 - 0.00129, 8 - 0.0341, 10 - 0.0103, 11 - 0.016, 12 - 0.0373, 15 - 0.0426, 16 - 0.00511, 17 - 0.00251, 18 - 0.00105, 19 - 0.000798, 20 - 0.000674, 21 - 0.000554, 22 - 0.000545, 23 - 0.000757, 24 - 0.000624, 25 - 0.000712, 26 - 0.000859, 27 - 0.00267, 28 - 0.00241, 29 - 0.000257, 30 - 0.00332, 31 - 0.00555, 32 - 0.00892, 33 - 0.0098, 34 - 0.0115, 35 - 0.0165, 36 - 0.00936, 37 - 0.00961, 38 - 0.011), WT2 vs. MKO (6 - 0.0479, 8 - 0.0057, 9 - 0.0006, 10 - 0.00037, 11 - 0.00068, 12 - 0.0072, 18 - 0.0407, 19 - 0.0296, 20 - 0.0389, 21 - 0.0494, 22 - 0.0465, 23 - 0.0452, 24 - 0.05, 26 - 0.0286, 27 - 0.0368, 28 - 0.0294, 29 - 0.025, 30 - 0.0188, 31 - 0.0152, 32 - 0.00902, 33 - 0.0085, 34 - 0.0054, 35 - 0.0085, 36 - 0.0036, 37 - 0.0015, 38 - 0.00036, 39 - 7.86E-05, 40 - 3.95E-05), WT2 vs. DKO2 (24 - 0.0351, 25 - 0.0271, 26 - 0.0245, 27 - 0.0317, 28 - 0.0129, 29 - 0.0078, 30 - 0.0058, 31 - 0.0077, 32 - 0.0111, 33 - 0.0044, 34 - 0.0103, 35 - 0.0049, 36 - 0.00995, 37 - 0.0091, 38 - 0.021, 39 - 0.025, 40 - 0.034), MKO vs. DKO2 (8 - 0.00198, 9 - 0.00081, 10 - 9.31E-05, 11 - 0.00011, 12 - 0.00067, 13 - 0.014, 18 - 0.033, 19 - 0.033, 20 - 0.038, 21 - 0.037, 22 - 0.021, 23 - 0.017, 24 - 0.014, 25 - 0.014, 26 - 0.0104, 27 - 0.019, 28 - 0.009, 29 - 0.0067, 30 - 0.0047, 31 - 0.0038, 32 - 0.0021, 33 - 0.00069, 34 - 0.0015, 35 - 0.00101, 36 - 0.000314, 37 - 0.00012, 38 - 6.28E-05, 39 - 9.36E-05, 40 - 0.00049); (F) Chain lengths: WT1 vs. PKO1 (4 - 0.0111, 5 - 0.0103, 15 - 0.0379, 16 - 0.0163, 17 - 0.00361, 18 - 0.000955, 19 - 0.00134, 20 - 0.000852, 21 - 0.00143, 22 - 0.00179, 23 - 0.000844, 24 - 0.000882, 25 - 0.00116, 26 - 0.00219, 27 - 0.00154, 28 - 0.00184, 29 - 0.0017, 30 - 0.00204, 31 - 0.0018, 32 - 0.00122, 33 - 0.00214, 34 - 0.00435, 35 - 0.0287); (G) WT1 vs. LKO 1.22E-05, WT1 vs. DKO1 3.49E-09, LKO vs. DKO1 0.0024, LKO vs. PKO1 1.43E-05, DKO1 vs. PKO1 4.67E-09, WT2 vs. MKO 1.17E-05, MKO vs. DKO2 2.79E-05, MKO vs. PKO2 7.79E-06. Source data are available online for this figure.

10-month-old LD mice with PTG knockout, enabling the study of laforin- and malin-deficient LD mice (LKO/MKO) and PTG single knockout mice (PKO) in comparison with their corresponding double knockout (DKO) and wild-type (WT) mice (Fig. 1B). Both, the laforin and malin cohort, underwent FM fixation, after which brain (FM-fixed) and skeletal muscle (CP-fixed) were collected and glycogen levels determined in both tissues. Our results confirm the previously reported complete prevention of glycogen accumulation in the brain and muscle of both LD DKO mouse models (Fig. 1C,D) (Turnbull et al, 2011; Turnbull et al, 2014).

The abundance of glycogen in skeletal muscle facilitated the analysis of glycogen chain length distribution (CLD) and glycogen C6 phosphate. As expected, the CLD is shifted toward longer chains in LKO and MKO mice (Fig. 1E). By contrast, CLDs of PKO and DKO are slightly shifted toward shorter chains in both cohorts compared to WT (Fig. 1E-F). The data confirm that accumulated glycogen in LD mice is poorly branched and reveal that PKO has a positive effect on branching frequency. In addition, glycogen is hyperphosphorylated in LKO and MKO, as previously shown (DePaoli-Roach et al, 2015; Tagliabracci et al, 2008; Tiberia et al, 2012; Turnbull et al, 2010), while PKO mice have normal phosphate levels. The two cohorts differ regarding glycogen phosphate levels in DKO mice, being elevated in the laforin line (DKO1) but normalized in the malin line (DKO2) (Fig. 1G). These results indicate that glycogen hyperphosphorylation and insolubility are uncoupled, further corroborating previous findings in mice expressing phosphatase-inactive laforin (Gayarre et al, 2014; Nitschke et al, 2017).

## PTG knockout prevents LD-related neuroinflammation

Neuroinflammation is a cardinal feature of LD in mice and inflammatory and immune system response genes were identified whose extent of upregulation aligns with disease progression (Della

Vecchia et al, 2022; Sanz and Serratosa, 2020). To assess the extent of neuroinflammation, we selected nine inflammatory response genes that showed strong upregulation in an RNAseq study in both LD models (Lahuerta et al, 2020). We confirmed elevated expression of all nine genes in LKO and MKO. Furthermore, PKO mice exhibited normal mRNA levels, as did DKO mice, showing no neuroinflammatory response (Fig. 2A–I). This implies that neuroinflammation is driven by the presence of PBs, the absence of which in DKO completely prevents the neuroinflammatory response in line with rescued astrogliosis previously shown in these mice (Turnbull et al, 2011; Turnbull et al, 2014).

In addition, *Chi3l1* expression was studied, as it was also upregulated in the aforementioned RNAseq study and because its human homolog, YKL40, is a leading CSF biomarker in other neuroinflammatory and neurodegenerative diseases such as Creutzfeld-Jacob and Alzheimer's disease (Baldacci et al, 2019; Llorens et al, 2017; Mavroudis et al, 2022). We confirm *Chi3l1* upregulation in both LKO and MKO and show normalization of expression in DKO, implying that glycogen accumulation is a prerequisite for the induction of this disease marker (Fig. 2J). Furthermore, immunodetection of brain CHI3L1 showed significantly increased levels in both LKO (Fig. 2K) and MKO (Fig. 2L) mice, suggesting the human analog YKL40 as a promising monitoring biomarker in LD.

## Brain metabolic pathways are only modestly impacted in LKO, but not in MKO mice

Previous publications show metabolic changes in LD mouse brains (Brewer et al, 2019; Duran et al, 2021). However, complete metabolomics data sets were only published for LKO mice (Brewer et al, 2019) and were obtained from CP-fixed brain, not accounting for rapid postmortem metabolic changes. We quantified 250 metabolites in the brains of LKO and MKO mice and respective

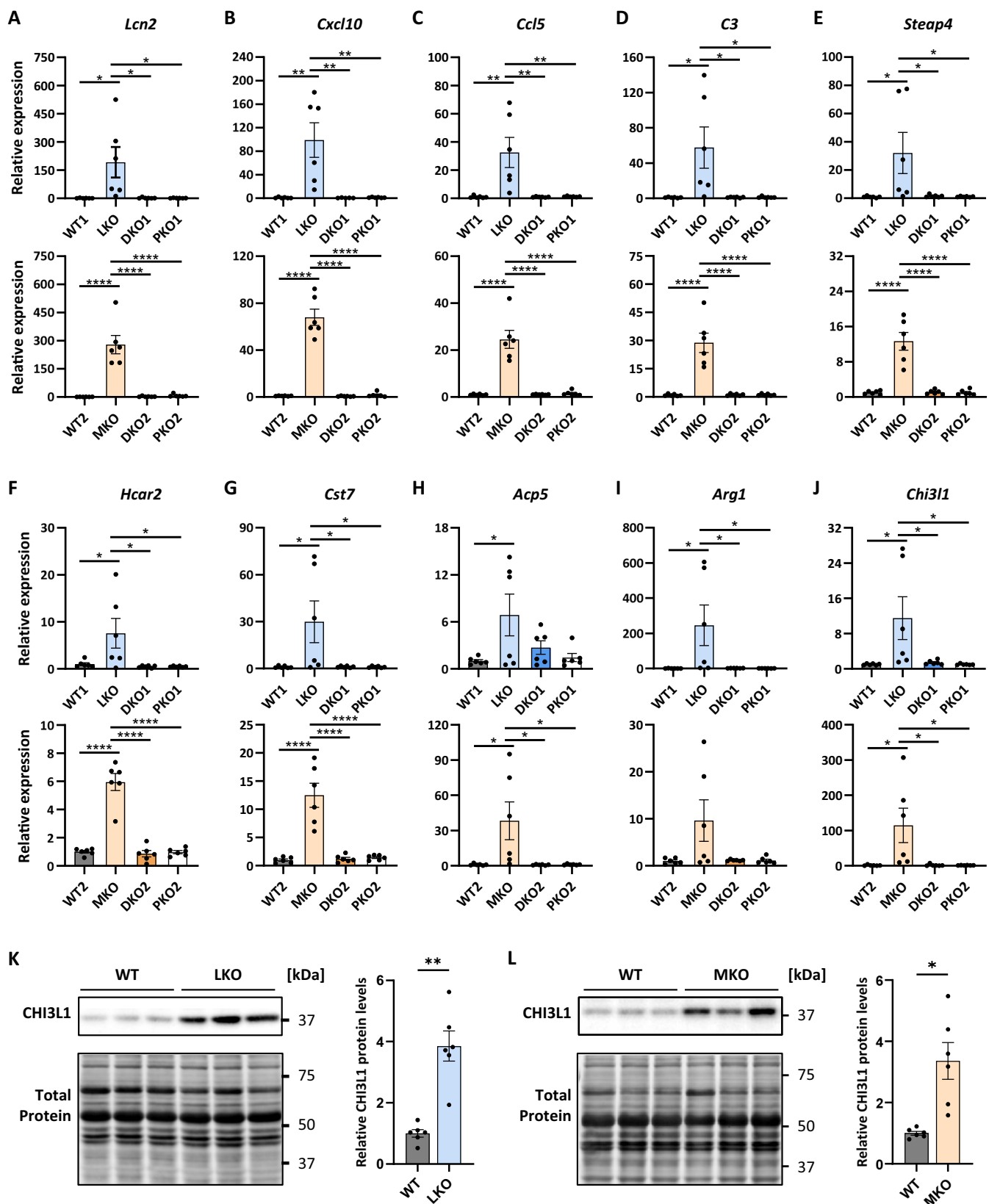

**Figure 2.  PTG knockout completely prevents neuroinflammatory response in LD mice.**

(A–J) LD-associated induction of neuroinflammatory response gene expression rescued in DKO mice of both LD cohorts subjected to FM fixation. $n = 6$. (K, L) CHI3L1 protein levels in LKO (K) and MKO (L) mice compared to WT. $n = 6$. Data information: All data were presented as mean ± SEM. *$p < 0.05$; **$p < 0.01$; ****$p < 0.0001$ by two-way ANOVA with Tukey post hoc analysis (A–J) or by Welch's $t$-test (K, L). n signifies the biological replication of measurements in different animals. P values: (A) WT1 vs. LKO 0.0166, LKO vs. DKO1 0.0173, LKO vs. PKO1 0.0168, WT2 vs. MKO 5.6E-07, MKO vs. DKO2 6.1E-07, MKO vs. PKO2 7.9E-07; (B) WT1 vs. LKO 0.0011, LKO vs. DKO1 0.0017, LKO vs. PKO1 0.0011, WT2 vs. MKO 7.7E-11, MKO vs. DKO2 7.1E-11, MKO vs. PKO2 9.6E-11; (C) WT1 vs. LKO 0.0025, LKO vs. DKO1 0.0025, LKO vs. PKO1 0.0026, WT2 vs. MKO 2.0E-07, MKO vs. DKO2 2.1E-07, MKO vs. PKO2 3.0E-07; (D) WT1 vs. LKO 0.0133, LKO vs. DKO1 0.0136, LKO vs. PKO1 0.0139, WT2 vs. MKO 1.5E-06, MKO vs. DKO2 1.6E-06, MKO vs. PKO2 1.6E-06; (E) WT1 vs. LKO 0.0321, LKO vs. DKO1 0.0361, LKO vs. PKO1 0.0332, WT2 vs. MKO 5.5E-07, MKO vs. DKO2 6.2E-07, MKO vs. PKO2 5.6E-07; (F) WT1 vs. LKO 0.0386, LKO vs. DKO1 0.0236, LKO vs. PKO1 0.0238, WT2 vs. MKO 6.5E-09, MKO vs. DKO2 4.1E-09, MKO vs. PKO2 5.9E-09; (G) WT1 vs. LKO 0.0292, LKO vs. DKO1 0.0301, LKO vs. PKO1 0.0294, WT2 vs. MKO 1.8E-06, MKO vs. DKO2 2.4E-06, MKO vs. PKO2 3.3E-06; (H) WT1 vs. LKO 0.0402, WT2 vs. MKO 0.0182, MKO vs. DKO2 0.0176, MKO vs. PKO2 0.0184; (I) WT1 vs. LKO 0.0327, LKO vs. DKO1 0.0336, LKO vs. PKO1 0.0329; (J) WT1 vs. LKO 0.0296, LKO vs. DKO1 0.0384, LKO vs. PKO1 0.0294, WT2 vs. MKO 0.0175, MKO vs. DKO2 0.0182, MKO vs. PKO2 0.0176; (K) WT vs. LKO 0.0017; (L) WT vs. MKO 0.0107. Source data are available online for this figure.

WTs after FM fixation. We identified 36 and 32 metabolites that were significantly ($p < 0.05$) altered in LKO (Fig. 3A) and MKO (Fig. 3B), respectively, with no overlap between the two LD mouse models regarding metabolites with fold changes (FC) of >1.25 or <0.8 (Fig. 3A,B). Only nine (of 36) and one (of 32) metabolites in LKO and MKO, respectively, passed the FDR threshold $q < 0.2$. Metabolic profiles of LKO/MKO and respective WTs clustered on a PLS-DA biplot, though less strictly for MKO and WT (Fig. 3C). However, no clustering was found on unsupervised PCA plots (Fig. EV1A,B) that do not maximize clustering as much as the supervised PLS-DA plot (Ruiz-Perez et al, 2020), indicating that the changes in metabolic profiles are small. The most influential metabolites driving the PLS-DA clustering are plotted in Fig. 3D, with on average lower VIP scores in MKO. Unsupervised cluster analyses, including all significantly changed metabolites, corroborate the observation that metabolic changes are generally modest with clustering into genotype groups less complete in MKO (Fig. EV1C,D). Pathway analysis identified six significantly changed pathways in LKO, while no pathways were changed in MKO (Fig. 3E). We directly compared the previously published LKO metabolome dataset (Brewer et al, 2019) with our dataset. The significantly changed metabolites in Brewer et al were found unchanged in our study (Fig. EV2A). Furthermore, of the 20 significantly changed pathways in Brewer et al, we found only three changed (Fig. EV2B). Metabolic changes previously found in MKO (Duran et al, 2021) were not detected in our study (Fig. EV2C). It is possible that FM fixation prevented previously published death-related metabolic changes due to faster inactivation of brain metabolism to reveal true LD-related metabolic profiles. These profiles being inconsistent between the two LD mouse models, suggests that metabolic changes seen in LKO brains are not disease-related.

## PTG knockout intensifies brain metabolic changes in LKO and MKO mice

PTG knockout prevents glycogen accumulation and the associated neuroinflammatory response. We asked whether PTG knockout also prevents changes observed in metabolic profiles and pathways. Comparing PKO with WT, we found no pathways changed in PKO1 (laforin cohort) and only two altered in PKO2 (malin cohort) (Fig. 4A–C). By contrast, in DKO1 (PKO + LKO) 20 pathways were significantly changed (Fig. 4D). Interestingly, even though no or very few pathways were found changed in MKO and PKO2, respectively, 24 pathways were significantly changed in DKO2 (PKO + MKO) (Fig. 4E). The majority of significantly

changed metabolites in LKO and MKO were still changed in the same direction in the respective DKO (Figs. 4F and EV3A). Similarly, all six pathways significantly changed in LKO were still changed in DKO1 with an even higher significance level (Fig. 4G) and similar directionality (Figs. 4I,J and EV3B–G). These data imply that the observed pathway changes are independent of glycogen accumulation, which is absent in PKO and DKO (Fig. 1C).

To investigate the roles of laforin and malin independent of glycogen accumulation, we compared DKO mice with the respective PKO. We found 14 pathways significantly changed in DKO1 (LKO + PKO), which were also changed in DKO2 (MKO + PKO) together with 13 additional pathways (Fig. 4H). The 14 overlapping pathways (changed in DKO1 and DKO2 without PBs) include the six pathways found changed in LKO (with PBs present). Further inspection of DKO-associated pathway changes revealed that even though partly mediated by different metabolites and/or different extents of change (FC and/or significance), the directionality of pathway changes was the same in most cases (Figs. 4K and EV3H–U). Taken together, these data indicate partly overlapping roles of malin and laforin in several metabolic pathways that are enhanced or only revealed in the absence of PTG. As pathway changes are not consistently present in both LD models and most severe in the absence of PTG (and PB) we conclude that abnormal metabolic flux is independent of PB formation. Pathway changes being virtually absent in PKO and MKO but present in DKO2 and enhanced in DKO1 suggest redundancy between laforin, malin, and PTG action in preventing abnormal metabolic flux (see the model in Fig. 8).

## Brain metabolome changes in Gbe1YS APBD mice are modest, not overlapping with changes in LD

To investigate the association of glycogen accumulation, neuroinflammation, and metabolic changes in the brain further, we consulted another PB disease mouse model and collected FM- and CP-fixed brains from Gbe1YS APBD mice. We confirmed a strong accumulation of insoluble glycogen in these mice (Fig. 5A–C). To evaluate the success of FM fixation in this cohort, we specifically looked at non-disease WT mice which do not accumulate degradation-resistant insoluble glycogen. Not only did the WT mice have 35% higher glycogen levels after FM (Fig. 5A) compared to CP fixation (Fig. 5B), glycogen levels in the FM-fixed WT brains also remained unchanged during preincubation while completely depleted in CP-fixed WT brains (Fig. EV4A). This confirms enzymatic inactivation through focused microwave irradiation.

Similar to LD mice, a causal relationship between PB formation and neuroinflammation has been shown for APBD mice (Chown et al, 2020; Gumusgoz et al, 2021; Gumusgoz et al, 2022). To gauge

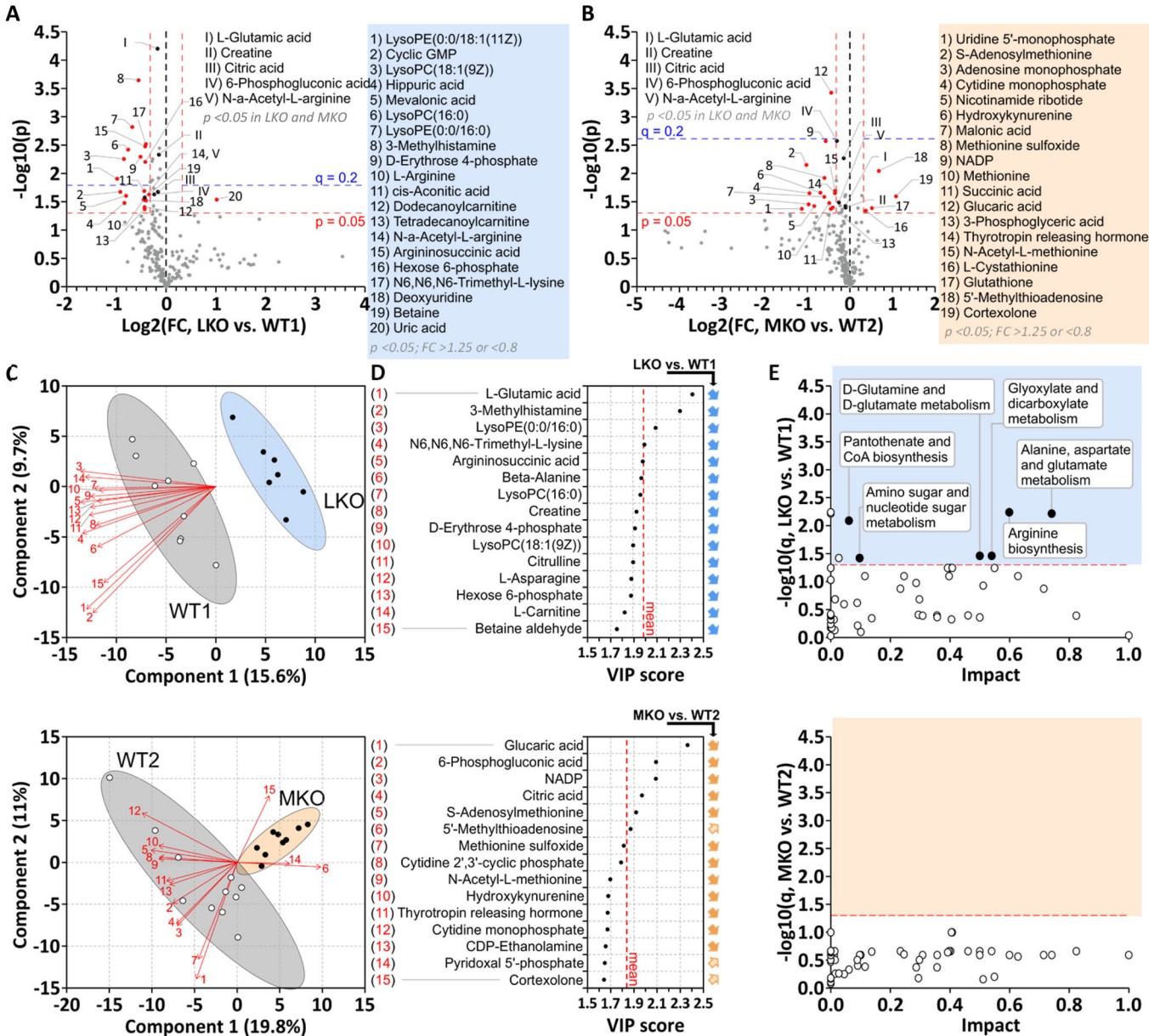

**Figure 3. LD brain metabolome is only modestly changed with metabolic pathways exclusively impacted in LKO mice.**

(A, B) Volcano plots of brain metabolites in LKO (A) and MKO (B) mice. Metabolites significantly changed in both LD mice are numbered I–IV. Significantly changed metabolites with FC >1.25 or <0.8 are indicated in red and listed in the box. FDR threshold ($q = 0.2$) is indicated. P values were calculated by unpaired two-tailed $t$-test and FDR threshold determined after Benjamini–Hochberg correction. (C) PLS-DA biplots of metabolic profiles in LKO versus WT1 and MKO versus WT2, respectively. Loadings of the 15 most influential metabolites (arrows) and scores of each animal (dots). (D) VIP score plots for LKO and MKO show the 15 most influential metabolites, driving respective PLS-DA plot clustering. Arrows indicate the directionality of change compared to WT. The mean VIP score of displayed metabolites is indicated. (E) Pathway analyses for LKO and MKO compared to respective WTs. Changed pathways are displayed as filled circles. Data information: Significance levels determined by MetaboAnalyst 5.0 as follows: $p < 0.05$ (A, B) or $q < 0.05$ (E), $n = 7$–11. $n$ signifies the biological replication of measurements in different animals. See also Figs. EV1, EV2. Source data are available online for this figure.

if CHI3L1 could be a monitoring biomarker candidate for APBD as well, we measured brain *Chi3l1* mRNA and protein levels in Gbe1YS and found them significantly increased (Fig. 5D,E), raising the possibility that YKL40 could be used as a tool to track the progression of neuroinflammation in PB diseases.

Finally, we quantified 180 metabolites in FM-fixed Gbe1YS brains with 17 significant metabolites passing the FC threshold

(>1.25 or <0.8) (Fig. 5F). Interestingly, there is almost no overlap with the metabolites found changed in LKO or MKO (Fig. 3A,B). The metabolic profiles of Gbe1YS versus WT clustered on a PLS-DA biplot (Fig. 5G), with metabolites influencing this projection the most shown with VIP scores in Fig. 5H. Overall, the changes in Gbe1YS were modest, indicated by the absence of clustering on a PCA plot (Fig. EV4B) and the identification of only one impacted

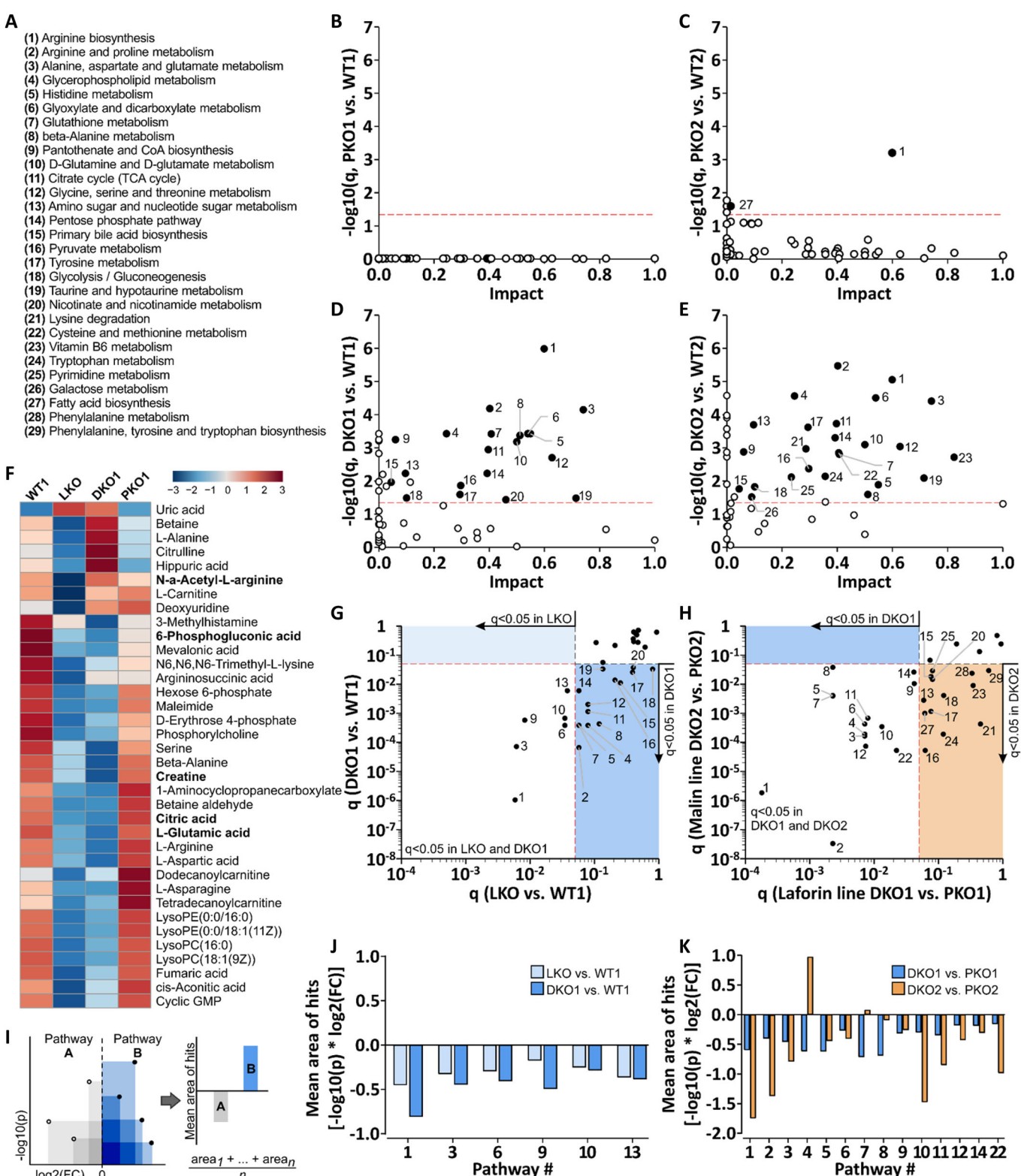

pathway (Fig. 5I), which was not changed in LKO (Fig. 3E) but found to be impacted in DKO1 and DKO2 (Fig. 4H), though with different directionality (Fig. EV4C–E). Combined, the metabolomic data of the three PB disease models show that there is no common set of brain metabolites or pathways that are changed in association with PB accumulation. However, in Gbe1YS UDP-glucose, Hex2, and Hex4 were increased (Fig. 5F, J–L) and are linked to glycogen metabolism and degradation, respectively. The first is the substrate for GYS1. The latter two, while not increased in LD mice (Fig. EV4F,G), could be glycogen degradation products, such as

**Figure 4.  PTG knockout does not normalize metabolic changes in LKO but reveals overlapping roles of PTG with laforin and malin.**

(A) List of metabolic pathways numbered #1–29 throughout this figure. (B–E) Pathway analyses for PKO1 (B), PKO2 (C), DKO1 (D), and DKO2 (E) compared to respective WTs. Changed pathways are displayed as filled circles (numbering as in A). (F) Heatmap for laforin cohort, comparing all significantly changed metabolites in LKO between the four genotypes. Metabolites significantly changed in both LD mice in bold. (G, H) Two-dimensional pathway analyses, comparing pathway changes in LKO and DKO1 (G, both compared to WT1) or DKO1 and DKO2 (H, both compared to respective PKO). Pathways significantly changed in at least one dimension are numbered as in (A). (I) Schematic illustrating how data were processed in (J, K) to estimate directionality. For hypothetical pathways A and B, mean areas (log2(FC) * −log10(p)) of metabolite hits on hypothetical volcano plots are calculated. $P$ values are calculated by unpaired two-tailed $t$-test. (J, K) The mean area of hits for pathways significantly changed in both genotype comparisons in (G, H). Directionality in LKO vs. WT compared to that in DKO1 vs. WT (J), and directionality in DKO1 vs. PKO1 compared to that in DKO2 vs. PKO2 (K), with pathways numbered as in (A). $P$ values were calculated by unpaired two-tailed $t$-test. Data information: Significance levels determined by MetaboAnalyst 5.0 as follows: $q < 0.05$ (B–E, G, H) or $p < 0.05$ (F), $n = 6$–11. $n$ signifies the biological replication of measurements in different animals of the same group. See also Fig. EV3. Source data are available online for this figure.

Glc4 detected in urine from patients with Pompe disease (Young et al, 2012) and in Pompe disease mice (Fig. EV4H). Strikingly, we show urine Glc4 strongly increased in Gbe1YS (Fig. 5M), but not in MKO mice (Fig. EV4I), suggesting urine Glc4 as a potential systemic biomarker for APBD as well.

## FM fixation reveals metabolically volatile α-glucan in the brains of three mouse models of PB disease

Quantification of soluble and insoluble glycogen from muscle showed accumulation of insoluble glycogen in LKO, MKO, and Gbe1YS with soluble glycogen levels indistinguishable from WT (Sullivan et al, 2019). To determine whether this is the same in the brain, we slightly modified the previously established glycogen separation method, used FM fixation to preserve the degradation-prone brain glycogen, and, more accurately, called the measured glucan material α-glucans. As in muscle, only soluble α-glucan was present in the WTs, while insoluble α-glucan accumulated in LKO, MKO, and Gbe1YS (Fig. 6A). Levels of insoluble glycogen determined after preincubation in age-matched CP-fixed brains were comparable (Figs. EV5A,B and 5C), confirming that FM fixation does not interfere with the detection of insoluble brain glycogen.

In contrast to previous findings in muscle, all three mutants contained significantly increased levels of soluble brain α-glucan compared to WT (Fig. 6A). However, in age-matched CP-fixed brains soluble α-glucan was not elevated in the mutants (Fig. 6B). Very similar to our results in Fig. 1A, FM fixation prevented a 30–50% loss of brain α-glucan in the WTs but, strikingly, the mutants had 80–90% less soluble brain α-glucan after CP fixation, while, as expected, insoluble α-glucan levels remained unchanged (Fig. 6A,B). These findings indicate that soluble α-glucan/glycogen pools in the three mutants are more metabolically volatile and degradation-prone, leading to a faster postmortem depletion than in the respective WTs.

## Low-molecular weight malto-oligoglucans constitute the metabolically volatile α-glucan

Strikingly, in all three mutants total α-glucan contents determined as amyloglucosidase-released glucose after extraction in a buffer with only mechanical force (Figs. 6A and EV5C) were consistently higher than when total glycogen was measured after extraction in heated 30% KOH with subsequent repeated ethanol precipitations (Figs. 1C [LKO and MKO], 5A [Gbe1YS]). The latter method determines macromolecular precipitable glycogen (a), while the former detects all α-glucans (b, glucosyl chains ≥2 glucose units), and therefore the difference between the methods $\Delta$(b-a) corresponds to lower molecular weight non-precipitable α-glucans (Fig. EV5C). We

hypothesized that these glucan moieties constitute the metabolically volatile excess soluble α-glucan in the three mutants (Fig. 6A). In line with this, $\Delta$(b-a) was elevated in Gbe1YS after FM (Fig. EV5D) and much reduced after CP fixation (Fig. EV5E).

To prove our hypothesis, we analyzed the soluble α-glucans of FM-fixed mutant and WT brains after ultra filtration, separating macromolecular glycogen and lower molecular weight α-glucan by a 30 kDa size cut-off. In the WTs virtually all soluble α-glucans were macromolecular (>30 kDa), while LKO and MKO had comparable and Gbe1YS even decreased levels compared to WT (Fig. 6C). This means that in the three disease models substantial amounts of soluble α-glucan were <30 kDa (Fig. 6D), confirming that the excess soluble α-glucans found in the mutants after FM fixation are in fact low-molecular weight α-glucans, namely malto-oligoglucans (MOGs).

## Malto-oligoglucans are not generally associated with low levels of macromolecular glycogen

In Gbe1YS macromolecular soluble glycogen was drastically reduced (Fig. 6C), indicating that most of the soluble α-glucan material in the brains of these mice are MOGs. Immunoblotting for brain GYS1 revealed that soluble GYS1 protein levels were greatly decreased in Gbe1YS (Fig. 6E) with RNA levels unchanged (Fig. 6F). By contrast, GYS1 levels in the insoluble fraction were strongly increased (Fig. 6G). The reduced GYS1 availability for glycogen synthesis in the soluble fraction likely explains the low levels of macromolecular glycogen in Gbe1YS and is in line with the elevated UDP-glucose levels we found (Fig. 5F,J). To examine whether a deficiency in macromolecular glycogen is generally associated with MOG formation as an alternative energy source, we determined soluble α-glucan levels in FM-fixed brains of DKO and PKO mice, both exhibiting low amounts of precipitable macromolecular brain glycogen (Fig. 1C). Neither DKO nor PKO mice showed a substantial increase in α-glucans compared to levels of precipitable glycogen (Fig. EV5F–I), indicating no or very minor MOG production. Therefore, MOG formation is unlikely a direct consequence of low macromolecular glycogen levels and notably is prevented in both DKO1 and DKO2 compared to LKO and MKO, respectively.

## Malto-oligoglucans are associated with later stages of LD and correlate with the extent of neuroinflammation

Next, we asked whether MOGs in LD mice are dependent on the disease stage. We determined α-glucans in FM-fixed brains of 5-week-old LKO and MKO mice, respectively, and found no

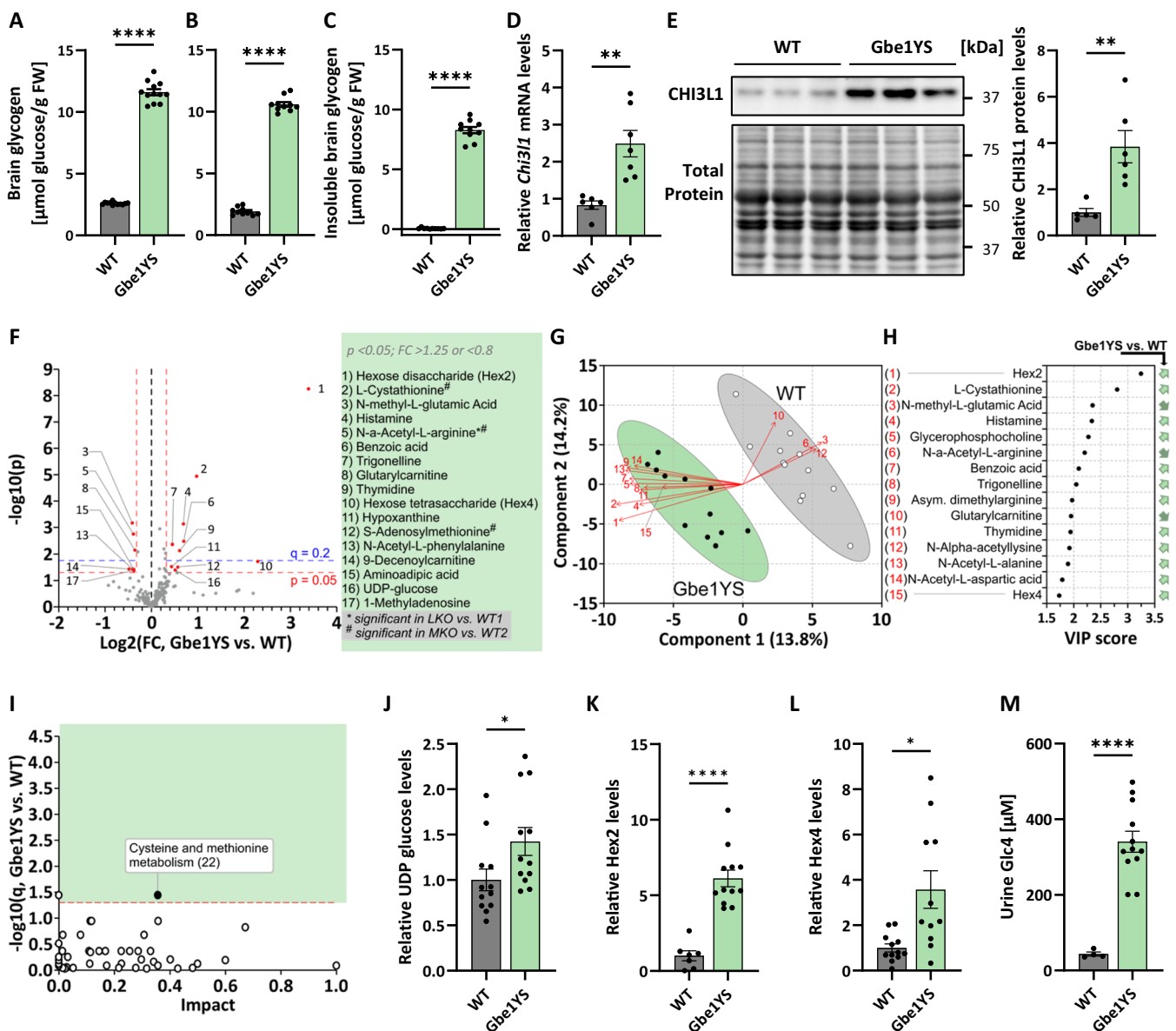

**Figure 5. Gbe1YS brain metabolomics reveals increased UDP-glucose, Hex2, and Hex4 levels but otherwise only modest changes not overlapping with LD.**

(A–C) Brain total glycogen in Gbe1YS after FM (A) and CP (B) fixation and insoluble (preincubated) glycogen in Gbe1YS mice after CP fixation (C). n = 10–12. (D, E) *Chi3l1* mRNA (D) and CHI3L1 protein levels (E) in Gbe1YS compared to wildtype. n = 5–7. (F) Volcano plot of brain metabolites in Gbe1YS compared to WT. Significantly changed metabolites with FC >1.25 or <0.8 are indicated in red and listed in the box. * and # indicate metabolites significantly changed in LKO or MKO, respectively. FDR threshold (q = 0.2) is indicated. *P* values were calculated by unpaired two-tailed *t*-test and FDR threshold determined after Benjamini–Hochberg correction. (G) PLS-DA biplots of metabolic profiles in Gbe1YS versus WT. Loadings of the 15 most influential metabolites (arrows) and scores of each animal (dots). (H) VIP score plots for Gbe1YS, showing the 15 most influential metabolites driving PLS-DA biplot clustering. Arrows indicate the directionality of change compared to WT. (I) Pathway analyses for Gbe1YS compared to WT. Changed pathways are displayed as filled circles. (J–L) Relative metabolite levels of UDP glucose (J), Hex2 (K), and Hex4 (L) in Gbe1YS compared to WT. n = 7–12. (M) Urine Glc4 levels in Gbe1YS compared to WT. n = 4–12. Data information: Data were presented as mean ± SEM (A–E, J–M). *p < 0.05; **p < 0.01; ****p < 0.0001 by Welch's *t*-test (A–E, J–M). Significance levels were determined by MetaboAnalyst 5.0 as follows: p < 0.05 (F) or q < 0.05 (I), n = 12. n signifies biological replication of measurements in different animals of the same group. See also Fig. EV4. Source data are available online for this figure. *P* values: (A) 3.4E-13, (B) < 1.0E-15, (C) 1.8E-10, (D) 0.0028, (E) 0.0084, (J) 0.0408, (K) 5.1E-07, (L) 0.0114, 2.5E-07. Source data are available online for this figure.

increase yet in the soluble fraction (Fig. 6H). Supporting the notion that MOGs are essentially absent at this age, Δ(b-a) values did not indicate a dramatic increase in non-precipitable α-glucans (Fig. EV6A). These results demonstrate that MOGs in LD mice are not present at a young age yet, when only very few PBs are present (Fig. EV6B,C) but are at later disease stages when PBs have

accumulated and/or neuroinflammation has progressed further. Interestingly, MOG contents in LKO and MKO mice, respectively, correlated particularly well with the neuroinflammatory markers *Lcn2*, *Cxcl10*, and *Ccl5* with correlation coefficients largely >0.6 (Figs. 6I and EV6D–G), indicating a connection between neuroinflammation and MOG production.

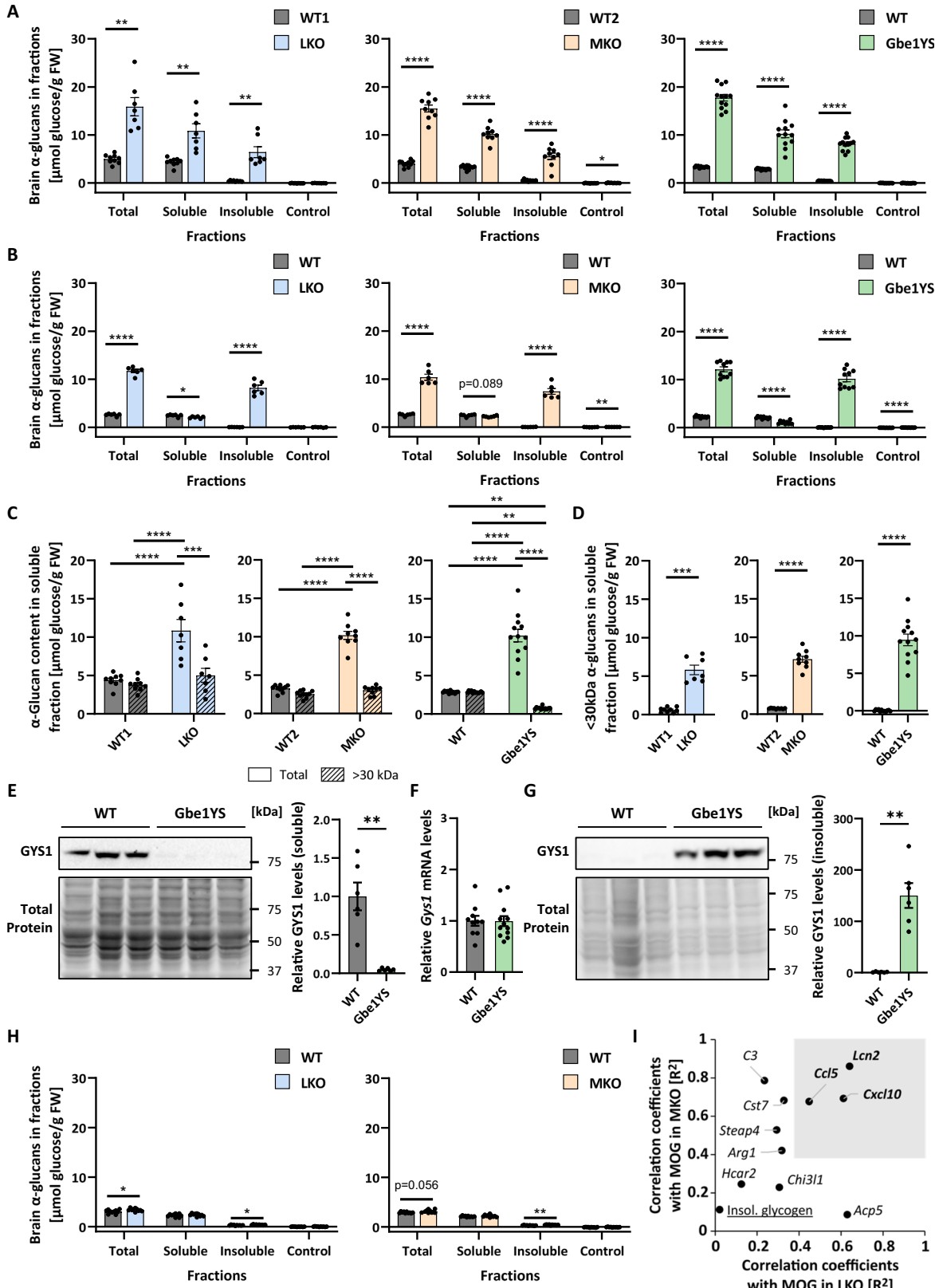

◄

**Figure 6.   Metabolically volatile malto-oligoglucans in brains of three PB disease mouse models associated with disease progression.**

(A, B) Total, soluble, and insoluble α-glucan in LKO, MKO, and Gbe1YS mice after FM (A) and CP (B) fixation. (C) Fractionation of soluble α-glucan from (A) using a 30 kDa size cut-off filter. (D) Calculated <30 kDa fraction corresponding to (C). (E–G) Brain soluble GYS1 protein levels (E), *Gys1* mRNA levels (F), and insoluble GYS1 protein levels (G) in Gbe1YS mice compared to wildtype. (H) Total, soluble, and insoluble α-glucan in five-week-old LKO and MKO mice after FM fixation. (I) Correlation coefficients ($R^2$) between malto-oligoglucans (MOGs, D) and neuroinflammatory marker expression (Fig. 2A–J) or insoluble glycogen (A), respectively, in LKO and MKO. See also Fig. EV6D–G. Data information: All data were presented as mean ± SEM. *$p < 0.05$; **$p < 0.01$; ***$p < 0.001$; ****$p < 0.0001$ by Welch's *t*-test (A, B, D–H) or by two-way ANOVA with Tukey post hoc analysis (C). $n = 6$–12. *n* signifies the biological replication of measurements in different animals. *P* values: (A) WT1 vs. LKO (Tot. 0.0011, Sol. 0.0042, Ins. 0.0018), WT2 vs. MKO (Tot. 6.6E-08, Sol. 2.7E-07, Ins. 5.7E-05, Contr. 0.021), WT vs. Gbe1YS (Tot. 2.6E-10, Sol. 2.0E-06, Ins. 4.2E-10); (B) WT1 vs. LKO (Tot. 2.7E-07, Sol. 0.0168, Ins. 3.4E-05), WT2 vs. MKO (Tot. 4.8E-05, Sol. 0.0898, Ins. 6.2E-05, Contr. 0.0085), WT vs. Gbe1YS (Tot. 7.5E-09, Sol. 7.4E-07, Ins. 4.5E-08, Contr. 2.6E-06); (C) Tot. (WT1 vs. LKO) 1.7E-05, WT1 > 30 kDa vs. LKO Tot. 3.9E-06, LKO (Tot. vs. >30 kDa) 0.00017, Tot. (WT2 vs. MKO) < 1.0E-15, WT2 > 30 kDa vs. MKO Tot. <1.0E-15, MKO (Tot. vs. >30 kDa) < 1.0E-15, Tot. (WT vs. Gbe1YS) 8.5E-13, WT > 30 kDa vs. Gbe1YS 8.5E-13, Gbe1YS (Tot. vs. >30 kDa) 8.4E-13, WT Tot. vs. Gbe1YS > 30 kDa 0.0041, >30 kDa (WT vs. Gbe1YS) 0.0043; (D) WT1 vs. LKO 1.2E-04, WT2 vs. MKO 1.9E-07, WT vs. Gbe1YS 9.4E-08; (E) 0.0032; (G) 0.0016; (H) WT vs. LKO (Tot. 0.0265, Ins. 0.0129), WT vs. MKO (Tot. 0.0561, Ins. 0.0019). Source data are available online for this figure.

## Phosphorylation state of PB-bound GYS1 is significantly reduced in LKO, MKO, and Gbe1YS

GYS1 and PTG have been suggested as malin targets (Vilchez et al, 2007; Worby et al, 2008) which can cause changes in GYS1 levels and phosphorylation state (activity) in LD, promoting longer glycogen chains and hence insolubility, which is in line with abnormally long chains seen in LD brain and muscle glycogen (Nitschke et al, 2017; Sullivan et al, 2019). However, global changes in GYS1 levels and/or activity could not be detected in LD mice (DePaoli-Roach et al, 2010; Sullivan et al, 2019; Tagliabracci et al, 2008; Tagliabracci et al, 2007; Tiberia et al, 2012; Valles-Ortega et al, 2011). As rapid postmortem changes to protein phosphorylation (O'Callaghan and Sriram, 2004) may have interfered with the detection of differences in GYS1 phosphorylation state and hence activity, we investigated the phosphorylation-dependent activation state of GYS1 in FM-fixed tissue. We determined the phospho-GYS1/GYS1 (P-GYS1/GYS1) ratio in total lysates via immunoblotting which, if decreased, would indicate increased GYS1 activity (Hunter et al, 2011; Lin et al, 2012). We found P-GYS1/GYS1 slightly reduced in 5-week-old LKO mice (Fig. 7A), indicating increased activity of soluble GYS1 because at this age, PBs are scarce and PB-bound GYS1 does not contribute to total GYS1 levels yet (Fig. EV6B). It is questionable whether this change in GYS1 phosphorylation is pathogenic, as we did not see a similar effect in MKO mice at this age (Fig. 7B), although LKO and MKO mice exhibit similar PB accumulation in the brain (Figs. EV6B, EV5A,B and 6A,B). By contrast, in both aged LKO and MKO mice the P-GYS1/GYS1 ratio is significantly reduced (Fig. 7C-D), resulting from a very strong increase in total GYS1 and a much smaller increase in P-GYS1. This increase in GYS1 is solely driven by PB-bound GYS1, since at this age soluble GYS1 levels are unchanged and GYS1 accumulates in PBs (Tagliabracci et al, 2008; Valles-Ortega et al, 2011). Therefore, these results show that PB-bound GYS1 in LKO and MKO is less phosphorylated than the soluble GYS1 in WT which could indicate that it was more active at the time of co-precipitation with abnormal glycogen. In aged Gbe1YS mice, the P-GYS1/GYS1 ratio is even more dramatically decreased (Fig. 7E) in line with a study by Kakhlon et al that showed an increased GYS1 activity in a *Gbe1*-deficient neuronal cell model (Kakhlon et al, 2013). The finding that all three aged PB disease models exhibit this reduced phosphorylation state in largely PB-associated GYS1, suggests that overactivation of GYS1 at eventually precipitating glycogen molecules is a major determinant of the glycogen chain length abnormality seen in PBs of LD and APBD.

## Discussion

Glycogen, the largest cytosolic molecule, is synthesized in many different cell types, tissues, and organs, serving as an energy source in the form of glucose either systemically (liver) or locally (brain and muscle). Glycogen metabolism requires soluble glycogen, allowing glycogen-metabolizing enzymes access to their substrate. Insoluble glycogen which gradually aggregates into PBs is associated with disease (Hedberg-Oldfors et al, 2019; Hedberg-Oldfors and Oldfors, 2015; Lausberg et al, 2021; Roach et al, 2012; Wong et al, 2021), including devastating neurological diseases such as LD and APBD. One overarching characteristic of insoluble glycogen is the presence of abnormally long glycogen chains (Nitschke et al, 2022; Sullivan et al, 2019). Additional character-istics have been described, which in LD mice include glycogen hyperphosphorylation, neuroinflammation, and metabolic altera-tions (Brewer et al, 2019; Duran et al, 2021; Lahuerta et al, 2020; Tagliabracci et al, 2008). However, the cause-effect relationship between these characteristics and PB accumulation is not well understood. Therefore, we analyzed muscle glycogen structure and phosphate as well as brain glycogen, brain metabolome, and neuroinflammation in LD mice with or without PTG knockout to study the effects of laforin or malin deficiency in the absence and presence of PBs, respectively. For comparison, we included Gbe1YS APBD mice, in which PB accumulation has a different genetic cause. Furthermore, we employed brain FM fixation to prevent postmortem glycogen degradation, metabolomic changes, and alterations of protein phosphorylation.

It is well established that PTG knockout prevents PB formation in both LD models (Turnbull et al, 2011; Turnbull et al, 2014). However, its effect on glycogen chain length (CLD) and phosphorylation has not been studied. Interestingly, PTG knockout rescued hyperphosphorylation in MKO (DKO2) but not in LKO (DKO1) mice (Fig. 8A). This result further corroborates that it is not the missing glycogen phosphatase function and hence hyperphosphorylation in the absence of laforin that leads to glycogen insolubility. Despite the persisting hyperphosphorylation in DKO1, glycogen remained soluble, like hyperphosphorylated glycogen in phosphatase-inactive laforin mice, which do not accumulate PBs (Gayarre et al, 2014; Nitschke et al, 2017). The normalized phosphorylation of DKO2 glycogen is completely in line with the fact that in MKO mice hyperphosphorylation is only found in insoluble glycogen (Sullivan et al, 2019) which is absent in DKO2 (Fig. 8A). Since glycogen hyperphosphorylation is a

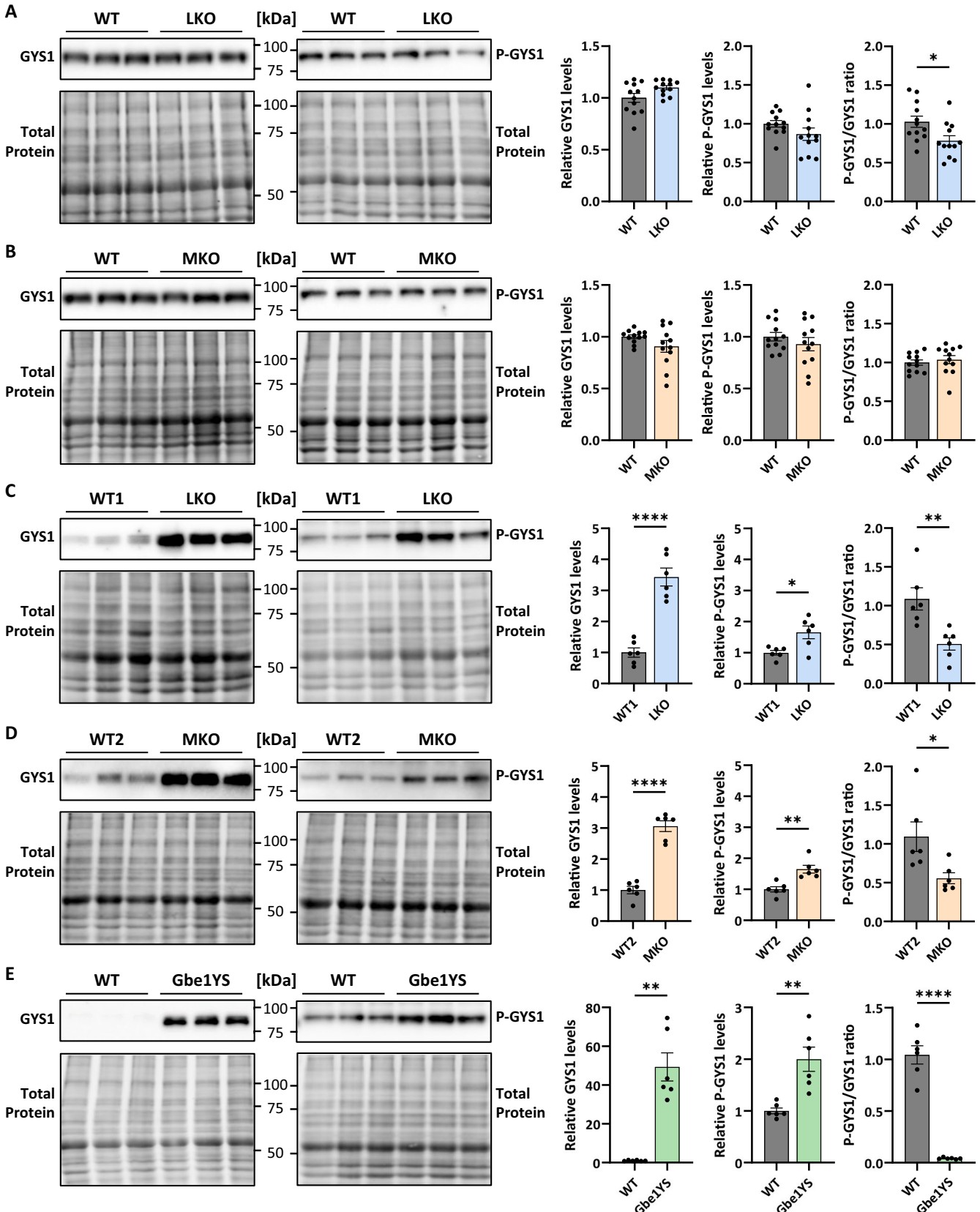

**Figure 7.  The phosphorylation state of polyglucosan-bound GYS1 is significantly reduced in LKO, MKO, and Gbe1YS.**

(A–E) Brain GYS1 and phosho-GYS1 protein levels in total homogenates (after FM fixation) from 5-week-old LKO (**A**) and MKO (**B**) and aged LKO (**C**), MKO (**D**), and Gbe1YS (**E**) mice. $n = 6$–12. Data information: All data were presented as mean ± SEM. *$p < 0.05$; **$p < 0.01$; ****$p < 0.0001$ by Welch's $t$-test. $n$ signifies the biological replication of measurements in different animals. $P$ values: (**A**) P-GYS1/GYS1 0.0203; (**C**) GYS1 9.2E-05, P-GYS1 0.0223, P-GYS1/GYS1 0.0081; (**D**) GYS1 4.5E-06, P-GYS1 0.0021, P-GYS1/GYS1 0.0363; (**E**) GYS1 0.0012, P-GYS1 0.0071, P-GYS1/GYS1 9.6E-05. Source data are available online for this figure.

hallmark of several PB diseases (Nitschke et al, 2022; Sullivan et al, 2019), the question remains what role the phosphate plays. Instead of being harmful, the phosphate could be introduced to keep glycogen soluble, like in starch where phosphorylation keeps glucan branches hydrated and accessible to degrading enzymes (Edner et al, 2007; Hejazi et al, 2008; Kötting et al, 2009; Zeeman et al, 2010).

In contrast to glycogen phosphate, LKO- and MKO-typical poor branching was rescued in both DKO mice (Fig. 8A). PTG knockout reduces the GYS1 activation state by about 50% (Israelian et al, 2021; Turnbull et al, 2014), which shifted CLDs of DKO glycogen toward chain lengths even shorter than WT, preventing PB formation. These data demonstrate once more that the balance between glycogen chain elongation and branching determines glycogen solubility. This knowledge has been widely applied in preclinical trials for different PB diseases with therapeutic down-regulation of GYS1 being extremely successful in preventing PB accumulation (Ahonen et al, 2021; Donohue et al, 2023; Gumusgoz et al, 2021; Gumusgoz et al, 2022; Nitschke et al, 2022). Strikingly, soluble and insoluble glycogen differ strongly with regard to CLD, with soluble glycogen being well-branched like WT (even in Gbe1YS) and exclusively insoluble glycogen displaying chain length abnormality. To explain this contrast, it was suggested that failed control of glycogen chain length pertains only to a small subgroup of glycogen molecules, which acquire overlong chains, eventually precipitate, but due to their low abundance, are not detectable globally (Sullivan et al, 2017). This was supported by the demonstration of glycogen molecules in WT with higher precipitation risk and longer chains than average WT glycogen (Sullivan et al, 2019). Our data show that PKO decreases glycogen chain length globally, even in comparison to WT, suggesting that PTG knockout in LD mice prevents PB formation by decreasing the occurrence of precipitation-prone ("at-risk") molecules (Fig. 8A).

A longstanding question in the field is why abnormally branched glycogen molecules are formed. While it seems obvious that GBE1 deficiency is the cause in APBD, no changes in GBE1 and GYS1 levels or activities could be detected in LD (DePaoli-Roach et al, 2010; Sullivan et al, 2019; Tagliabracci et al, 2008; Tagliabracci et al, 2007; Tiberia et al, 2012; Valles-Ortega et al, 2011), even though GYS1 and PTG have been proposed as malin targets (Vilchez et al, 2007; Worby et al, 2008). However, considering that insoluble glycogen likely originates from a very small pool of "at-risk" molecules and accumulates over time, we asked if the shift from "at-risk" to insoluble glycogen molecules could be attributed to locally increased GYS1 activity. In this case, co-precipitated PB-bound GYS1 may show marks of a previously higher activation state. Interestingly, lysates from FM-fixed brains of 10-month-old LKO and MKO mice, where total GYS1 is dominated by the PB-bound fraction of the enzyme, show a significantly smaller proportion of the phosphorylated form of GYS1 (reduced P-GYS1/GYS1). This is in line with a model where a few more active (less phosphorylated) GYS1 molecules would increase the

chain length of a small subset of glycogen molecules, which become "at-risk" molecules and eventually precipitate and simultaneously co-precipitate the less-phosphorylated form of GYS1. Aged GBE1-deficient mice show a similar effect implying that the association of overactive GYS1 with eventually precipitating glycogen molecules is a general mechanism underlying glycogen insolubility, which needs to be counteracted by frequent branching and potentially local regulation of GYS1 by the laforin-malin complex (Fig. 8A). It is intriguing to see a higher activation state of PB-bound GYS1 in Gbe1YS mice. The lack of branching activity would sufficiently explain glycogen insolubility in APBD. However, it seems that *Gbe1* deficiency also has an impact on the GYS1 activation state, in line with the increased GYS1 activity found in a *Gbe1*-deficient neuronal cell model (Kakhlon et al, 2013), the mechanism of which is unknown. In LD, the higher GYS1 activation state found in PB-bound GYS1 could potentially be explained by reduced ubiquitination and consequent degradation of the malin target PTG (Vilchez et al, 2007; Worby et al, 2008) on "at-risk" molecules that succumb to precipitation because of that. Laforin's CBM20 domain and preference for longer glucan chains (Chan et al, 2004) should target the laforin-malin complex particularly well to "at-risk" molecules that have longer chains. Therefore, localized lack of PTG degradation could be either, directly, due to malin deficiency (MKO) or, indirectly, due to impaired targeting of malin to ("at-risk") glycogen when laforin is absent (LKO).

To answer the question of whether global changes in GYS1 activation state have been previously overlooked in LD mice because of rapid postmortem changes to protein phosphorylation (DePaoli-Roach et al, 2010; Tagliabracci et al, 2008; Tagliabracci et al, 2007; Tiberia et al, 2012; Valles-Ortega et al, 2011), we studied P-GYS1/GYS1 ratios in FM-fixed brains of five-week-old LD mice. Firstly, FM fixation preserves global protein phosphorylation (O'Callaghan and Sriram, 2004). Secondly, at this young age, essentially all GYS1 is soluble, i.e., not bound to PBs, which are still sparse (Tiberia et al, 2012). We found a slightly reduced P-GYS1/GYS1 ratio in young LKO but not in young MKO mice. Since only seen in LKO, the reduced Ser641-phosphorylation of GYS1 could potentially be explained by reduced GSK3β activity due to reduced Ser9 dephosphorylation by laforin (Lohi et al, 2005; Wang et al, 2006). However, even though our results indicate a globally increased activation state of GYS1 (Hunter et al, 2011; Lin et al, 2012) in LKO we would argue that this level of GYS1 activation was not sufficient to contribute to PB formation since the levels of insoluble glycogen are indistinguishable between LKO and MKO at this age (Fig. EV6B). We further conclude that decreased GSK3β activity due to lack of laforin action is unlikely the cause of PB formation in LD as it would only explain PBs in laforin-deficient LD mice (Lohi et al, 2005; Wang et al, 2006). However, aged LKO and MKO mice are indistinguishable in terms of P-GYS1/GYS1 ratio (Fig. 7C,D), levels of insoluble glycogen (Figs. 6A,B and EV5A,B) as well as chain length abnormality (Nitschke et al, 2017). Moreover, mouse models expressing phosphatase-inactive laforin

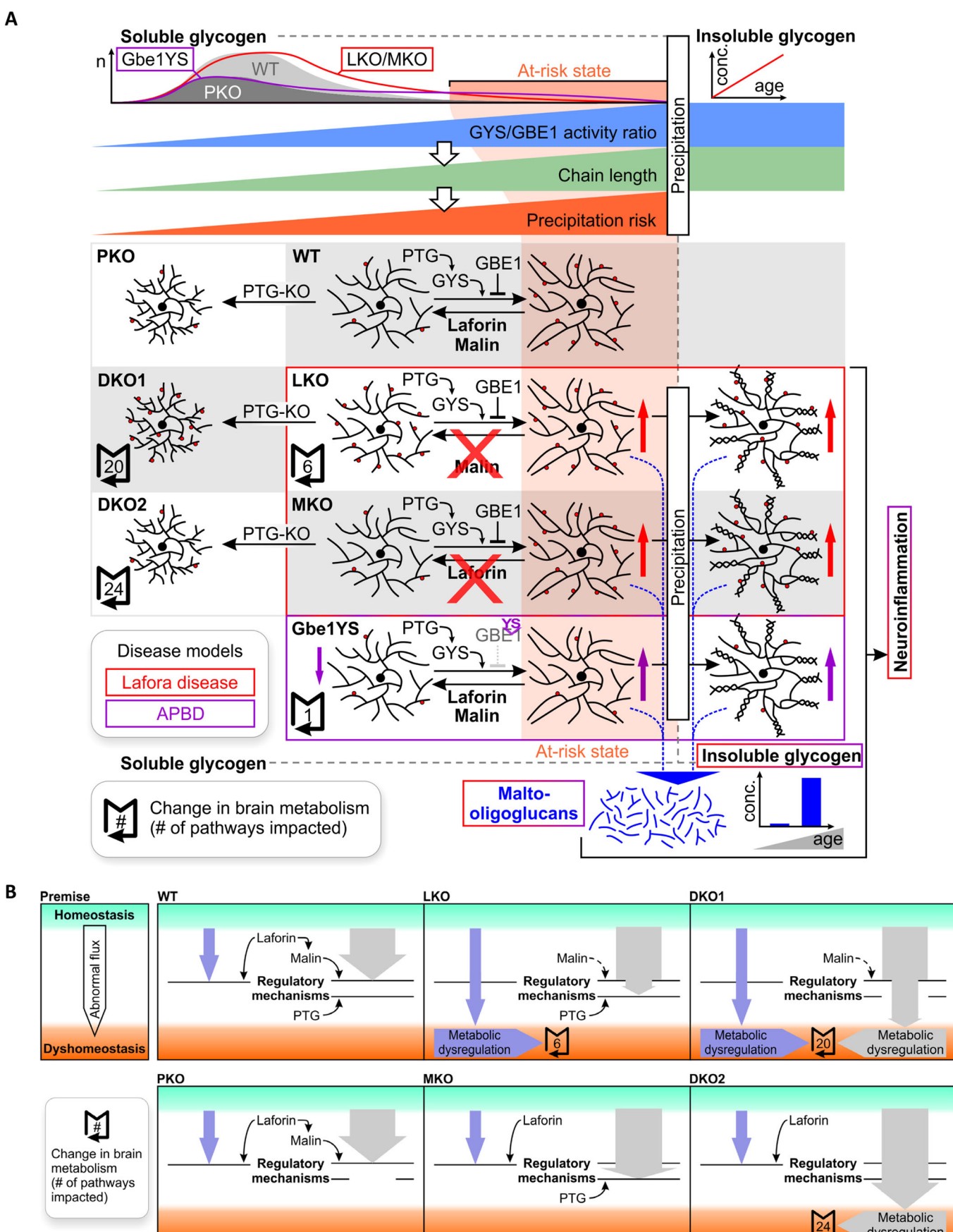

**Figure 8.  Modeling the roles of metabolic and glycogen-related changes in LD and APBD.**

(A) A small subgroup of glycogen molecules is at risk of precipitate due to the occasional occurrence of an increased ratio between glycogen chain elongation and branching. In the absence of laforin, malin, or sufficient GBE1, the amount of "at-risk" state glycogen is proposed to increase, leading to precipitation and age-dependent accumulation of insoluble glycogen that resembles the "at-risk" glycogen regarding GYS/GBE1 ratio, chain length and glycogen phosphate (red circles). PTG-KO leads to overall decreased GYS activation and chain elongation, thereby preventing the formation of "at-risk" glycogen. In all three disease models, malto-oligoglucans are formed in an age-dependent fashion, when insoluble glycogen is accumulating. Neuroinflammation correlates with the presence of insoluble glycogen and malto-oligoglucans. Metabolic pathways are impacted independently of insoluble glycogen and MOG, with the largest number of pathways being abnormal when PTG is knocked out in the LD mouse models, and insoluble glycogen and neuroinflammation is prevented. (B) Model explaining changes in brain metabolism by a partly redundant regulatory mechanism that prevents abnormal metabolic flux in a laforin-, malin-, and PTG-dependent fashion. Laforin is proposed to have a regulatory role independent from malin and PTG, leading to several impacted pathways in LKO. Regulatory mechanisms of malin and PTG are redundant in preventing abnormal flux with no pathways impacted in the single knockouts (PKO, MKO). Upon additional PTG knockout in MKO (DKO2), abnormal flux becomes possible. PTG knockout also exacerbates the metabolic phenotype in LKO possibly due to partial reduction of the proposed laforin-dependent malin function. This reduction of malin function is compensated in the presence of PTG (LKO), but manifests in its absence (DKO1).

do not accumulate PBs (Gayarre et al, 2014; Nitschke et al, 2017; Skurat et al, 2024), proving that laforin's phosphatase function is not needed to prevent PB formation. Another player in the modulation of GYS1 activity is its allosteric activator glucose-6-phosphate (G6P) (Roach et al, 2012), which is also subjected to rapid postmortem degradation in the brain (Juras et al, 2023). However, our LKO and MKO metabolomics data did not show increased G6P levels (detected as hexose 6-phosphate) that could indicate global GYS1 overactivity.

Our two FM-fixed LD cohorts enabled us to study neuroinflammatory responses as well as metabolomic changes in LD mice with and without PB accumulation after halting metabolism instantaneously by high-power-focused microwave irradiation. Neuroinflammation is a major characteristic of LD, worsening with disease progression (Lahuerta et al, 2020; Lopez-Gonzalez et al, 2017). To assess the level of neuroinflammation, we selected a comprehensive set of inflammatory response genes and measured their expression levels at 10 months. With no exception, the expression of neuroinflammation marker genes was normalized in both LKO and MKO mice with PTG knockout. This clearly shows that neuroinflammation is not a direct consequence of laforin or malin deficiency, respectively, but depends on the formation of PBs and/or MOG (see below), which results from the loss of function of both genes (Fig. 8A). These findings are in line with other studies that, by genetic or therapeutic approaches downregulating glycogen synthesis, show PB (and likely MOG) formation in LD fully or partially prevented together with neuroinflammatory responses (Ahonen et al, 2021; Duran et al, 2021; Gumusgoz et al, 2021; Gumusgoz et al, 2022; Israelian et al, 2021; Nitschke et al, 2020; Pederson et al, 2013; Turnbull et al, 2011; Turnbull et al, 2014; Varea et al, 2021). Further work is needed to identify the molecular mechanism of how PBs and/or MOG elicit an inflammatory response.

Knowing that metabolic changes occur within seconds after death due to rapid hypoxia-driven postmortem changes of metabolism (Juras et al, 2023), we employed FM brain fixation to rule out any effect from death-related metabolic reprogramming. Overall, we found only modest metabolome changes in LKO with none of the metabolites, previously found significantly changed in cryopreserved LKO brains, altered in our dataset and only three pathways overlapping (Brewer et al, 2019). This could indicate that previously seen metabolic changes may, at least in part, be due to a laforin-dependent response to death-associated hypoxia. Strikingly, even fewer metabolome changes were detected in MKO. Only five metabolites were significantly but slightly changed in both LKO

and MKO as well as no pathways significantly changed in MKO, indicating that metabolic changes detected in mice are largely specific to laforin or malin deficiency, respectively, and not associated with LD pathogenesis, which leads to PBs and neuroinflammation in both models to a similar extent. Importantly, metabolic changes in LKO and MKO were largely not normalized in the absence of PTG despite the prevention of PBs and neuroinflammation, corroborating the notion of disease-unrelated metabolic changes. Importantly, metabolic pathway changes in DKO mice rather intensified, pointing to a shared role in metabolic homeostasis between PTG and laforin or malin, respectively. In malin-deficient DKO2, 27 pathways were affected, while none (MKO) or one (PKO2), respectively, were impacted in the single knockouts, indicating functional redundancy in the regulation of these pathways. However, changes in these pathways do not have disease connection because DKO2 mice are not only rescued in terms of PB accumulation and neuroinflammation (Fig. 8A) but also in terms of disease-defining myoclonus (Turnbull et al, 2014). We consolidated these findings in a model, where laforin, malin, and PTG act inside of a partly redundant regulatory mechanism that prevents abnormal metabolic flux (Fig. 8B). We propose laforin to have a role independent of malin and PTG, leading to several impacted pathways in LKO. Regulatory mechanisms of malin and PTG are redundant in preventing abnormal flux with virtually no pathways impacted in the single knockouts (PKO, MKO). Additional PTG knockout in MKO (DKO2) permits abnormal flux. PTG knockout exacerbating the metabolic phenotype in LKO may be due to partial reduction of the recently proposed laforin-dependent malin function (Skurat et al, 2024). This reduction of malin function is compensated in the presence of PTG (LKO), but manifests in its absence (DKO1) (Fig. 8B).

Lastly, our Gbe1YS metabolomics dataset showed only very minor metabolic changes, which means that there seems to be no common metabolic signature in the brains of the studied PB disease mouse models. We interrogated our data with respect to a potential impact on central carbon metabolism, with the prospect of identifying a potential metabolic biomarker for PB diseases. Though individual metabolites in glycolysis, the pentose phosphate pathway (PPP), and the tricarboxylic acid (TCA) cycle were slightly decreased, these changes were not consistent between the three PB disease models. For instance, citrate and 6-phosphogluconate were slightly decreased in LKO and MKO, but not in Gbe1YS. Erythrose 4-phosphate and G6P were decreased in LKO, but not in MKO or Gbe1YS. 3-phosphogyceric acid was slightly decreased in MKO, but not in LKO or Gbe1YS. On the pathway level, glycolysis, the PPP, and the TCA cycle do not seem to be significantly

affected in single mutants (LKO, MKO, Gbe1YS, PKO). However, they seem to be impacted in the DKO mice. Here, also some of the emanating pathways show decreases in metabolites, including most prominently several in amino acid metabolism that feed off of glycolysis or the TCA cycle. This suggests a connection between laforin, malin, and PTG with flux in central carbon metabolism in the brain. However, these pathway changes may not be associated with pathological changes in glycogen metabolism because of the inconsistent display of these changes in the different PB disease models. Taken together, the use of individual metabolites of central carbon metabolism as sensitive PB disease-specific biomarkers may be limited.

We analyzed glycogen content in FM- and CP-fixed brains from three disease models with PB accumulation. While microwave-fixation prevents postmortem degradation, brain glycogen is only 40–50% reduced when cryopreservation occurs within 45 s of death. Absolute FM-fixed brain glycogen levels were 60–75% lower than those previously reported (Oe et al, 2016), but are in line with relative glycogen levels in FM- and CP-fixed brains published recently (Juras et al, 2023). We propose monitoring and publishing death-to-cryopreservation time when analyzing brain glycogen after CP fixation.

Separating soluble and insoluble brain α-glucans (including glycogen), we identified malto-oligoglucans (MOGs) as a metabolically volatile metabolite pool that is clearly associated with neuroinflammatory processes in all three PB disease models. FM fixation of the brain was necessary to detect MOGs in the soluble α-glucan fraction, while they were already completely depleted in CP-fixed brains. This shows that MOGs are metabolized much faster than glycogen, which may suggest a role in hypoxia/hypoglycemia protection, a role that had previously been ascribed to brain glycogen (Choi et al, 2003; Herzog et al, 2008; Saez et al, 2014). MOGs are low-molecular weight α-glucans and, therefore, non-precipitable at ethanol concentrations that precipitate macromolecular glycogen (Balto et al, 2016; Good et al, 1933). They are absent in young LD mice (LKO and MKO) as well as aged LD mice with PTG knockout (DKO), which suggests that MOGs are associated with a more advanced disease state, potentially accumulating along with PBs and progressing neuroinflammation (Fig. 8A).

Analysis of the relationship between MOGs, PB accumulation, and neuroinflammation on the individual animal level revealed a strong positive correlation of MOG content with the expression levels of neuroinflammatory marker genes *Cxcl10*, *Ccl5*, and *Lcn2*, i.e., with the severity of the neuroinflammatory response in both LD mouse models. This indicates that MOG production is linked to neuroinflammatory processes, but whether in a causative fashion or as a consequence remains unclear. It is conceivable that MOGs contribute to neuroinflammation as they are raising the intracellular levels of osmotically active molecules, which could elicit an osmotic stress signal that modulates inflammatory and immune responses, similar to extracellular hyperosmolarity (Brocker et al, 2012). Alternatively, MOG production could be the benign result of an attempt to reduce overlong glycogen chains in animals with more pronounced neuroinflammation. This would require neuroinflammation-stimulated endo-acting enzymes such as amylases that have access to cytosolic glycogen. Amylases are known as secreted enzymes, but recent studies have shown their implication in Alzheimer's disease, their presence within astrocytes and neurons, and their potential association with glycogen degradation (Byman et al, 2021; Byman et al, 2018; Byman et al, 2019). An alternative mechanism to produce MOG could be glycogenin (GYG)-independent de novo synthesis by GYS1. Note, ultrafiltration showed that MOGs are smaller than 30 kDa. Therefore, they must be GYG-free since GYG has a

molecular weight of 37 kDa. However, GYG knockout mice accumulated glycogen molecules that were significantly larger than WT glycogen (Testoni et al, 2017) and hence much larger than MOGs. Investigations into the origin and exact function of MOGs as well as their role in neuroinflammation are ongoing.

After the removal of MOGs, macromolecular glycogen levels were WT-like in both LD models, similar to soluble glycogen levels in LD and APBD muscle (Sullivan et al, 2019). By contrast, we found macromolecular brain glycogen drastically decreased in APBD mice, probably caused by >95% reduced soluble GYS1 levels in the brain, which likely limits glycogen synthesis. The increased UDP glucose levels support the notion that flux toward glycogen is diminished, similar to Gys1-KO (Saez et al, 2014) or Gys1-R582A knock-in mice (Bouskila et al, 2010). GYS1 reduction can be explained by the strong sequestration of GYS1 to the insoluble fraction, where it is likely bound to PBs. GYS1 is also PB-bound in LD mice and accumulates in the insoluble fraction with disease progression but not to the same extent and without depletion of GYS1 in the soluble fraction (Tagliabracci et al, 2008; Valles-Ortega et al, 2011; Varea et al, 2021). It remains to be elucidated how GBE1 deficiency, but not laforin or malin deficiency, causes GYS1 sequestration into the insoluble fraction at similar levels of insoluble glycogen.

The relevance of appropriate GYS1 activity regulation is illustrated by the aforementioned success of several preclinical approaches that aimed at reducing GYS1 activity to prevent glycogen insolubility (Ahonen et al, 2021; Donohue et al, 2023; Gumusgoz et al, 2021; Gumusgoz et al, 2022; Nitschke et al, 2022) with an antisense oligonucleotide therapy going into a clinical trial for LD (ClinicalTrials.gov ID NCT06609889). Other promising approaches include *EPM2A* gene replacement, removal of PBs by bioengineered hydrolases, and the use of an autophagic activator with an FDA orphan designation for the treatment of APBD (Brewer et al, 2019; Kakhlon et al, 2021; Zafra-Puerta et al, 2024). Efforts of translation into the clinic are greatly facilitated by the availability of biomarkers that correlate with disease state and demonstrate drug efficacy (Califf, 2018). In the absence of an established disease biomarker for LD and APBD, this study provides several candidates. The relatively large pool of MOGs, their tight disease association, especially with neuroinflammation, suggests MOGs as a biomarker for PB diseases. Soluble MOGs could be detectable by non-invasive GlycoNOE MRI, which has been successfully used to quantify liver glycogen and is currently being extended to the brain (Zeng et al, 2024; Zhou et al, 2020). Furthermore, our work suggests the investigation of YKL40 as a monitoring biomarker of neuroinflammation in PB diseases. YKL40 is a robust CSF biomarker in other neuroinflammatory diseases, such as Creutzfeld-Jakob and Alzheimer's disease (Baldacci et al, 2019; Llorens et al, 2017; Mavroudis et al, 2022), and the mouse homolog of human YKL40, CHI3L1, was elevated in brains of LD and APBD models and expression was normalized in rescued LD (DKO) mice. Prompted by our Gbe1YS metabolomics data we investigated urine Glc4 and found it elevated in APBD mice. Glc4 is an established, easily accessible clinical biomarker for Pompe disease (Young et al, 2012) with potential use in other diseases with systemic glycogen accumulation, such as GSDIII (Young et al, 2020) and now APBD, as a systemic biomarker.

Besides providing promising biomarker candidates for LD and APBD, this work delineates the cause-effect relationship between the physicochemical properties of glycogen (solubility, hyperphosphorylation, chain length), metabolic dysregulation, and neuroinflammation. Our data support the idea of a localized imbalance between chain elongation and branching reactions at the root of PB

diseases by showing (1) PB-bound GYS1 to have a higher activation state (explaining the longer glycogen chains seen in LD glycogen) and (2) LD mice rescued by PTG-KO to exhibit a significantly shorter glycogen chain length (likely explaining the reduced precipitation risk in these mice). Furthermore, our results corroborate the notion that glycogen hyperphosphorylation and glycogen insolubility are uncoupled. We discover MOGs as a new rapidly metabolized energy source in the brain that is increasing with disease progression. While the presence of PBs and MOGs are strictly linked to neuroinflammation, metabolic dysregulation appears uncoupled from the aberrant glycogen phenotype. Metabolic dysregulation is not consistently present between the three mouse models of LD and APBD, and it seems to be enhanced by the additional absence of PTG, which prevents PB formation completely (Fig. 8A). The comprehensive set of brain metabolomes acquired in this study allows the proposal of a model in which laforin, malin, and PTG prevent abnormal metabolic flux by a partly redundant regulatory mechanism (Fig. 8B).

# Methods

### Reagents and tools table

| Reagent/resource | Reference or source | Identifier or catalog number | |
|---|---|---|---|
| **Experimental models** | | | |
| Mouse: *Epm2a*⁻/⁻: (B6.129-*Epm2a*^tm1.1Kzy) | (Ganesh et al, 2002) | N/A | |
| Mouse: *Epm2b*⁻/⁻: (B6-*Nhlrc1*^tm1Bmin) | (Turnbull et al, 2010) | N/A | |
| Mouse: *Ppp1r3c*⁻/⁻: (B6.Cg-*Ppp1r3*^ctm1Adpr) | (Zhai et al, 2007) | N/A | |
| Mouse: *Gaa*⁻/⁻: (B6;129-*Gaa*^tm1Rabn/J) | (Raben et al, 1998) | JAX:004154 | |
| Mouse: *Gbe1*^ys/ys: (B6.129-*Gbe1*^tm2.1Hoa) | (Akman et al, 2015) | N/A | |
| **Antibodies** | | | |
| Rabbit monoclonal anti-GYS [15B1] | Cell Signaling | 3886 | |
| Rabbit monoclonal anti-phospho GYS | Cell Signaling | 3891 | |
| Rabbit monoclonal anti-YKL40/CHI3L1 | Abcam | ab259322 | |
| Donkey polyclonal anti-Rabbit IgG-HRP | Thermo Fisher Scientific | 31458 | |
| **Oligonucleotides and other sequence-based reagents** | | | |
| Genotyping primers | This study | N/A | |

| Genotype | Direction | Sequence (5′ to 3′) | Amplicon size (bp) |
|---|---|---|---|
| *Epm2a* KO (LKO) | FWD | GCATCGGCTG TAAGTTAGCC | 430 |
| | REV | AGCGTATTCAA TAACCCTTAAT | |
| *Epm2a* WT | FWD | GCATCGGCTGT AAGTTAGCC | 620 |
| | REV | CGTGTGTCCATT CTCCAGAA | |
| *Epm2b* KO (MKO) | FWD | AAGCGAAGGAGCA AAGCTGCTATTGGCC | 573 |
| | REV | CATGTCTCAATC TTTAATCCTGGAAT CTTCC | |

| Reagent/resource | Reference or source | Identifier or catalog number | |
|---|---|---|---|
| *Epm2b* WT | FWD | TGTGACCTTCG ATCACCAAG | 500 |
| | REV | TGAATGCTCTG GTCTGTCT | |
| Gbe1YS and Gbe1-WT | FWD | AGTGACCATGA TTGGCTAGCTT | 256 (Gbe1YS) |
| | REV | GTCTATGTCCAG CACAGTATTAAGGA | 321 (Gbe1-WT) |
| *Ppp1r3c* KO (PKO) | FWD | AGATCTCATCACC CCAGTGC | 192 |
| | REV | TAGTTCCCAGG CTGTCCTTG | |
| *Ppp1r3c* WT | FWD | GAGCTGTGTCAG ACTTGTTCAGAT AGAGC | 400 |
| | REV | TTGAAAACCATTG TAAGGACCCAGGA AACTC | |
| qPCR primers | This study | N/A | |

| Gene | Direction | Sequence (5′ to 3′) | Amplicon size (bp) | Primer efficiency |
|---|---|---|---|---|
| *Rpl4* | FWD | CCCTTACGCCA AGACTATGC | 124 | 100.4% |
| | REV | TGGAACAACCT TCTCGGATT | | |
| *Ppia* | FWD | GCTGGACCAAA CACAAACG | 92 | 101.6% |
| | REV | TTCACCTTCCC AAAGACCAC | | |
| *Lcn2* | FWD | GCCTCAAGGAC GACAACATC | 75 | 101.7% |
| | REV | CACACTCACC ACCCATTCAG | | |
| *Cxcl10* | FWD | AAGTGCTGCCG TCATTTTCT | 158 | 99.9% |
| | REV | ATAGGCTCGCA GGGATGATT | | |
| *C3* | FWD | CTGTGTGGGT GGATGTGAAG | 102 | 103.0% |
| | REV | TCCTGAGTGTC GTTTGTTGC | | |
| *Ccl5* | FWD | TGCCAACCCA GAGAAGAAGT | 111 | 95.7% |
| | REV | AGCAAGCAATG ACAGGGAAG | | |
| *Steap4* | FWD | CGTGAATGGGA AAACAGATG | 185 | 109.3% |
| | REV | AAAGCATCCAA TGGTCAAGC | | |
| *Hcar2* | FWD | TCTGCTCAAAG TCACGGATG | 156 | 91.8% |
| | REV | ATACTGGTCCC ACCGAACAC | | |
| *Chi3l1* | FWD | CAACCTGAAG ACCCTCCTG | 138 | 100.8% |
| | REV | CCATCAAAGCC ATAAGAACG | | |
| *Acp5* | FWD | CCATTGTTAGC CACATACGG | 99 | 101.6% |
| | REV | ACTCAGCACA TAGCCCACAC | | |
| *Cst7* | FWD | TGCCTTGAAGC GGACTCTAT | 172 | 94.1% |
| | REV | CAGGATGGGG TGGAAAGTAA | | |

| Reagent/resource | Reference or source | Identifier or catalog number | | |
|---|---|---|---|---|
| *Arg1* | FWD | GGGCAACCTG TGTCCTTTCT | 139 | 87.9% |
| | REV | AGTGTTCCCCA GGGTCTACG | | |
| *Gys1* | FWD | CGCAAACAACT ATGGGACAC | 116 | 96.9% |
| | REV | TCCTCCTTGTC CAGCATCTT | | |
| **Chemicals, enzymes, and other reagents** | | | | |
| Isoamylase | Sigma-Aldrich | 08124 | | |
| Amyloglucosidase (3.26 U/mL) | Megazyme | E-AMGDF-40ML | | |
| Glucose-6-phosphate dehydrogenase (G6PDH) from *Leuconostoc* | Sigma-Aldrich | G5760 | | |
| Glucose-6-phosphate dehydrogenase (G6PDH) from *Leuconostoc* | Roche | 1016587001 | | |
| Hexokinase (HK) from yeast "overproducer" | Roche | 11426362001 | | |
| Precision Plus Protein™ Dual Color Standards | Bio-Rad | 1610374 | | |
| 3-(4,5-dimethylthiazol-2-yl)-2,5-diphenyltetrazolium (MTT) | Sigma-Aldrich | M2128 | | |
| Phenazine-methosulfate (PMS) | Sigma-Aldrich | P9625 | | |
| Adenosine 5'-monophosphate (AMP) | Sigma-Aldrich | A1752 | | |
| Adenosine 5'-triphosphate (ATP) | Roche | 10127523001 | | |
| Nicotinamide adenine dinucleotide phosphate (NADP) | Roche | 10128058001 | | |
| Glucose-6-phosphate (G6P) | Sigma-Aldrich | G7879 | | |
| Protease inhibitor cocktail | Roche | 11836170001 | | |
| Phosphatase inhibitor cocktail | Roche | 04906837001 | | |
| Blotting grade blocker non-fat dry milk | Bio-Rad | 1706404 | | |
| TRiZol reagent | Invitrogen | 15596018 | | |
| **Software** | | | | |
| CaseViewer 2.4 | 3DHISTECH Ltd. | N/A | | |
| MetaboAnalyst 5.0 | Metaboanalyst.ca | N/A | | |
| Thermo Scientific™ Dionex™ Chromeleon 7.2 | Thermo Fisher Scientific | N/A | | |
| Prism 10.0 | GraphPad | N/A | | |
| ImageLab 6.0.1 | Bio-Rad | N/A | | |
| Microsoft Excel 2013 | Microsoft | N/A | | |
| QuantStudio 7 Pro software (Design & Analysis Software 2.3.3) | Thermo Fisher Scientific | N/A | | |
| **Other** | | | | |

| Reagent/resource | Reference or source | Identifier or catalog number |
|---|---|---|
| Clarity™ Western ECL Substrate | Bio-Rad | 170-5061 |
| Revert 700 Total Protein Stain | LI-COR | 92611021 |
| DC protein assay kit II | Bio-Rad | 5000112 |
| Pierce BCA protein assay | Thermo Fisher Scientific | 23225 |
| PureLink RNA Mini-Kit | Invitrogen | 12183018 A |
| PureLink DNase kit | Invitrogen | 12185010 |
| iTaq Universal SYBR Green Master Mix | Bio-Rad | 1725121 |
| TissueRuptor II | Qiagen | #9002755 |
| TissueRuptor Disposable Probes | Qiagen | #990890 |
| Rodent brain microwave-fixation system | Muromachi | TMW-6402C and TMW-4012C |
| Amicon Ultra-0.5 Centrifugal Filter Unit, 30 kDa | MilliporeSigma | UFC510096 |
| SpectraMax iD3 plate reader | Molecular Devices | iD3 |
| Dionex HPAEC-PAD | Thermo Scientific | ICS5000 |
| ChemiDoc MP Imaging System | Bio-Rad | N/A |

## Methods and protocols

### Mice

The mouse models used in this study have been previously described: (1) laforin knockout, $Epm2a^{-/-}$, herein termed LKO (Ganesh et al, 2002), (2) malin knockout, $Epm2b^{-/-}$ (also known as $Nhlrc1^{-/-}$), herein termed MKO (Turnbull et al, 2010), (3) GBE1-deficient mice, homozygous *Gbe1* c.986 A > C (p.Y329S), herein termed Gbe1YS (Akman et al, 2015), (4) protein targeting to glycogen (PTG) knockout, $Ppp1r3c^{-/-}$, herein termed PKO (Turnbull et al, 2011; Zhai et al, 2007), (5) acid alpha-glucosidase knockout, $Gaa^{-/-}$ (Raben et al, 1998). Mice were housed with environmental enrichment in ventilated cages at 20–22 °C and fed a commercially available diet with water accessible ad libitum, either at The Centre for Phenogenomics, Toronto, or at the University of Texas Southwestern Medical Center's Animal Resource Center facility. All animal procedures were approved by The Centre for Phenogenomics Animal Care Committee (protocol #21-0044H) or by the Institutional Animal Care and Use Committee of the University of Texas Southwestern Medical Center (protocol # 2021-103163) and are in compliance with the Canadian Council for Animal Care Guidelines and the OMAFRA Animals for Research Act. Several littermate cohorts were generated for this study: (Cohort 1) 10-month-old WT1, LKO, PKO1, and DKO1 (laforin and PTG double knockout) by crossing mice heterozygous for LKO and PKO; (Cohort 2) 10-month-old WT2, MKO, PKO2, and DKO2 (malin and PTG double knockout) by crossing mice heterozygous for MKO and PKO; (Cohort 3) 10-month-old WT, MKO, and LKO by crossing mice heterozygous for MKO and LKO; (Cohort 4) 7-month-old WT and Gbe1YS by crossing mice heterozygous for Gbe1YS; (Cohort 5) 5-week-old WT and LKO by crossing mice heterozygous for LKO;

(Cohort 6) 5-week-old WT and MKO by crossing mice heterozygous for MKO; (Cohort 7) 6-month-old WT and MKO by crossing mice heterozygous for MKO; (Cohort 8) 4–5-month-old WT and Gbe1YS by crossing mice heterozygous for Gbe1YS; (Cohort 9) 3-month-old WT and *Gaa*$^{-/-}$ generated by and obtained from The Jackson Laboratory. Mice of both sexes were used, except in cohort 3, where only females were available. Genotyping primers for LKO, MKO, Gbe1YS, and PKO are provided in the Reagent and Tools Table.

## Mice specimen and methods of sacrifice

Animals were sacrificed by one of the two following methods: (1) designated as CP, by cervical dislocation with fast dissection of brain and hindlimb muscle and cryopreservation (CP) of the tissue by immediate freezing in liquid nitrogen or fixation of the tissue for at least 48 h in 10% buffered formalin before embedding in paraffin; (2) designated as FM, by ultra-fast brain fixation with a 0.95–1.4-s head-focused microwave (FM) pulse, delivered by a Muromachi microwave rodent brain fixation system (10 kW-model TMW-4012C for cohorts 1 and 2; 5-kW-model TMW-6402C for cohorts 4, 5, and 6), followed by dissection of the fixed brain as well as of unfixed hindlimb muscle, both of which frozen in liquid nitrogen and stored at −80 °C until further processing. Cohorts 1 and 2 were sacrificed using FM, and cohort 3 by CP. Animals of cohorts 4, 5, and 6 were sacrificed using CP and FM at similar animal numbers per genotype group. Cohorts 7, 8, and 9 were used for urine collection. Urine was obtained from the mice in the afternoon by placing the animal over a plastic wrap and allowed to urinate spontaneously for 30 s. If necessary the animal was restrained using the one-handed scruff technique and a bladder massage was performed by gently stroking the abdomen for up to 30 s. If urine was not obtained, repeat collection was attempted in the following days. Care was taken not to contaminate the urine with stool. The urine was transferred from the plastic to a microcentrifuge tube with a micropipette and snap-frozen in liquid nitrogen within two hours of collection. Samples were kept at −80 °C until further processing.

## Glycogen and α-glucan analyses

The correct interpretation of this study's results regarding glycogen and α-glucan quantification requires a solid understanding of the terminology. Therefore, a few definitions are provided here. α-Glucans are oligo- and polymeric compounds composed of glucose units that are linked via α-glycosidic linkages. α-Glucans include glycogen, which essentially is a glucose polymer containing exclusively α1,4- and α1,6-linkages. The enzyme amyloglucosidase (AMG) cleaves exclusively α1,4- and α1,6-glycosidic linkages and releases all glucose units from α-glucans that are α1,4- or α1,6-linked and allows their quantification for instance by an enzymatic (glucose-specific) assay. Historically, glycogen has been extracted from animal tissue by boiling in 30% KOH, which is followed by several rounds of ethanolic precipitation (Good et al, 1933). Later, the ethanol-insoluble isolate (glycogen) has been characterized as a spherical macromolecule containing up to 60,000 glucose units linked via α1,4- and α1,6-linkages. It is logical to assume that repeated ethanolic precipitation removes all α-glucans from glycogen that are soluble in the ethanol conditions applied, i.e., smaller α-glucans in addition to free glucose. Quantification methods that omit ethanolic precipitation do not distinguish between ethanol-soluble (lower molecular weight) and ethanol-insoluble α-glucans (higher molecular weight). In this study essentially two quantification paradigms were employed, one using ethanolic precipitation determining precipitable macromolecular glycogen (termed total glycogen), and the other omitting ethanolic precipitation quantifying all α1,4- and α1,6-linked α-glucans that are composed of two or more glucose units (termed α-glucans) (Fig. EV5C). Additional steps, such as the degradation of all metabolically accessible α-glucans by preincubation at room temperature for 1 h prior to extraction and the fractionation of extracted α-glucans by centrifugation allowed the determination of degradation-resistant glycogen and distinction of soluble and insoluble α-glucans, respectively.

### Total and insoluble glycogen extraction

Glycogen content was determined essentially as described previously (Suzuki et al, 2001). Briefly, frozen tissue was ground in liquid nitrogen, boiled in 30% [w/v] KOH, precipitated once in 67% [v/v] ethanol and 14.3 mM sodium sulfate, and precipitated thrice in 67% [v/v] ethanol and 15 mM LiCl (at least 2 h at −20 °C). After redissolving in water, aliquots were subjected to glycogen quantification. For analysis of insoluble brain glycogen, ground tissue from cryopreserved cohorts was incubated at room temperature for 1 h (preincubation) prior to glycogen extraction in KOH (Nitschke et al, 2020). As the described extraction method involves several rounds of ethanolic precipitation the obtained glycogen consists of ethanol-precipitable macromolecular glycogen only.

### Fractionation of α-glucans

α-Glucan separation into soluble and insoluble fractions was performed as previously described with a few adjustments for brain tissue (Nitschke et al, 2022). Approximately 100 mg of frozen ground brain were homogenized in 1 mL of cold glycogen isolation buffer (GIB; 50 mM Tris, pH 8, 150 mM NaCl, 2 mM EDTA, 50 mM NaF, and 5 mM sodium pyrophosphate), samples were homogenized in an ice bath using a tissue homogenizer (TissueRuptor II, Qiagen). After the removal of an aliquot for total α-glucan quantification, Triton X-100 was added to the homogenate to reach 0.95% [w/v], and recurrent vortexing of the ice-cold sample for 15 min was conducted to optimize the separation of soluble and insoluble glycogen. Subsequently, the remainder of the homogenate was centrifuged at 4 °C and 13,000×*g* for 10 min. The supernatant was collected with an aliquot removed for soluble α-glucan determination. The pellets were repeatedly washed by resuspension in 1.2 mL GIB and subsequent centrifugation (see above). As a control, the sixth supernatant was collected with an aliquot removed for quantification of α-glucans removed from the pellet during the last wash. The pellets were finally resuspended in 0.75 mL GIB, and an aliquot was removed for quantification of the insoluble α-glucans, i.e., insoluble glycogen. All fractions and aliquots were immediately snap-frozen and stored at −80 °C until further processing. Prior to glycogen quantification in the aliquots, the Triton X-100 concentration was adjusted (to be equal in each fraction), and a heat treatment (15 min, 95 °C) was added to eliminate all enzymatic activities. Furthermore, an aliquot of the soluble α-glucan fraction was filtrated using an ultra-centrifugal filter with a size cut-off of 30 kDa, the retentate being washed by three passages of 400 uL water. α-Glucan input (1) as well as α-glucan in the retentate (2) were quantified. The low-molecular weight α-glucans that passed the filter were calculated as the difference between (1) and (2) and termed malto-oligoglucans

(MOG, "malto" signifying the possible AMG-sensitive linkage type α1,4 and α1,6 as found in maltose and iso-maltose). Contents of MOG and insoluble glycogen quantified in the insoluble α-glucan fraction, as well as mRNA expression of neuroinflammatory markers were correlated across mutant mice (Figs. 6I and EV6D–G).

### α-Glucan and glycogen quantification

α-Glucan and glycogen content was determined as glucose after subjecting an aliquot to enzymatic degradation using AMG. Non-glucan-associated (free) glucose in α-glucan or glycogen samples was quantified without prior glycogen degradation and found to be negligible in total and insoluble glycogen preparations. Digestion method A was used to digest α-glucans in aliquots derived from the α-glucan fractionation method described above in a total volume of 150 μL: sample was topped up to 80 μL with GIB (see above), mixed with 37 μL water, 23 μL acetate mix (1:8 mixture of 10% acetic acid:200 mM sodium acetate buffer (pH 4.5)) and 10 μL of a 1:4 diluted solution of AMG, and finally incubated at 55 °C for 60 min. Free glucose in the fractions was quantified in mixtures where AMG was replaced by water and that were left on ice for 60 min. Digestion method B was used to digest aliquots of total and insoluble glycogen as well as of soluble retentate α-glucans after 30 kDa ultrafiltration in a total volume of 100 μL: aliquots were topped up to 50 μL with water, mixed with 40 μL of 200 mM sodium acetate buffer (pH 4.5) and 10 μL of a 1:20 diluted solution of AMG, and finally incubated at 55 °C for 1 h. Control digests, containing GIB (method A) or water (method B) instead of sample, were analyzed to account for free glucose in the AMG preparation.

### Glucose quantification

After centrifugation (room temperature (RT), $16,000 \times g$, 20 min), glucose levels were determined in supernatants of AMG digests using a previously reported method (Lowry and Passonneau, 1972) in a 96-well format. All glycogen samples as well as seven serially diluted D-glucose standards were mixed with 170 μL of glucose-6-phosphate dehydrogenase (G6PDH, Roche) reaction mix (150 μL of 200 mM tricine/KOH (pH 8), 10 mM $MgCl_2$; 18 μL of deionized water; 1 μL of 112.5 mM NADP; 1 μL of 180 mM of ATP; 0.5 U of G6PDH (Roche). After recording the background absorbance for 20 min at 340 nm, 4 μL of hexokinase (HK; 0.75 U in 4 μL of 200 mM tricine/KOH (pH 8), 10 mM $MgCl_2$) were added to each well, and again the absorbance was recorded for 30 min at 340 nm. The average background absorbance was subtracted from the absorbance plateau after completion of the HK/G6PDH reaction. The glucose standard curve was used to calculate the glucose concentration in the samples. Glucose determination was performed in duplicates for each digest and standard. This assay highly reproducibly detects 0.25 to 30 nmol glucose per well with absorbance-concentration correlation coefficients $R^2 > 0.999$.

### Total glycogen C6P phosphate quantification

Glycogen C6 phosphate (GC6P) was used as a proxy for total phosphate esterification of glycogen as previously performed (Sullivan et al, 2019). GC6P was determined by measuring glucose-6-phosphate (G6P) in neutralized hydrolysates of purified glycogen samples using the enzymatic cycling assay previously described (Nitschke et al, 2013). Briefly, acid hydrolysis of total glycogen was performed to release C6 phosphorylated glucosyl

residues from total glycogen preparations (0.7 M HCl, 3 h at 95 °C) and was followed by neutralization using 5 M KOH. Aliquots of hydrolysates, containing up to 30 nmol glucose, as well as authentic G6P standards were adjusted with water to 25 μL. After adding 20 μL of buffer A (50 mM tricine/KOH (pH 8.0), 2.5 mM $MgCl_2$, and 0.325 mM NADP), the mixture was equilibrated at 30 °C for 10 min. About 10 μL of 0.25 M HCl were added, followed by 10 min incubation at 30 °C and neutralization with 10 μL of 0.25 M NaOH. Subsequently, 20 μL of buffer B (50 mM tricine/KOH (pH 8.0), 0.6 μg BSA, 0.043 U G6PDH (Roche)) were added, followed by incubation at room temperature (RT) for 10 min. Finally, 20 μL of 0.5 M NaOH were added prior to incubation for 5 min at 95 °C, cooling in ice water, and neutralization with 20 μL of 0.5 M HCl. In a cold room, the entire mixture was transferred to a clear 96-well plate. Then 80 μL of freshly prepared buffer C (200 mM tricine/KOH (pH 8.0), 0.279 mg EDTA, 0.0195 mg PMS, 0.162 mg G6P, 0.063 mg MTT, 0.0765 mg BSA, and 6 U G6PDH (Sigma)) were added, and absorbance (570 nm) was recorded in 1 min intervals for 2 h at 30 °C. Linear ranges of formazan formation were used to correlate G6P standards with samples of unknown G6P contents.

### Total glycogen chain length distribution (CLD) analyses

Total muscle glycogen (10–20 μg) was completely debranched by incubating overnight at 37 °C with 200 U isoamylase in 110 μL 10 mM sodium acetate (pH 5). The enzyme was inactivated by heating at 95 °C for 10 min. Samples were centrifuged (10 min, $20,000 \times g$) and 90 μL of the supernatant were applied to high-performance anion exchange chromatography with pulsed amperometric detection (HPAEC-PAD, Thermo Fisher, ICS5000). Oligoglucan chains were separated as described previously (Nitschke et al, 2017) and chromatograms were analyzed with Dionex Chromeleon 7.2. The relative peak areas for each chain length (specific number of glucose units) were determined and averaged among biological replicates. Significance between genotypes was tested for each individual chain length using one-way analysis of variance (ANOVA) followed by post hoc analysis using homoscedastic Student's t- or heteroscedastic Welch's tests, both unpaired, two-tailed, and with Holm–Bonferroni correction. The adequate post hoc test was selected based on equal or unequal variances of groups tested with F-tests.

### Protein analyses

Focused microwave (FM) fixation deactivates all proteins in the brain and renders virtually all proteins insoluble. This precludes FM tissue from separate analyses of soluble and insoluble protein fractions. While FM stabilizes labile protein phosphorylation, analyses of phosphorylated proteins require the use of total protein lysates with re-solubilization of all suspended proteins during boiling in SDS sample buffer prior to SDS-PAGE. Therefore, analyses of the glycogen synthase (GYS1) phosphorylation state were conducted in total protein lysates of FM-fixed brains from LKO, MKO, and Gbe1YS mice and their corresponding WTs (cohorts 1, 2, 4, 5, and 6). The sequestration of GYS1 to the insoluble fraction in Gbe1YS mouse brains was studied using soluble and insoluble lysate fractions from cryopreserved (CP) tissue (cohort 5). CHI3L1 was studied in the soluble lysate fraction of CP-fixed brains from LKO, MKO, and Gbe1YS mice and the corresponding WTs (cohorts 3 and 4).

### Lysate preparation

About 40–60 mg of frozen liquid nitrogen-ground mouse brain tissue was homogenized on ice in 100 μL per 10 mg tissue extraction buffer (50 mM Tris, pH 8.0, 1 mM EDTA, 1 mM EGTA, 50 mM NaF, 10 mM Na beta-glycerol phosphate, 5 mM Na pyrophosphate, 2 mM DTT, protease inhibitor cocktail, phosphatase inhibitor cocktail) using a tissue homogenizer (TissueRuptor II, Qiagen). Aliquots of the total lysate (containing all proteins, irrespective of their association with insoluble material) were transferred to a new vial and frozen on dry ice, and the remainder was centrifuged (20 min, 4 °C, 16,000 × g) to remove insoluble material. The resulting supernatant (containing soluble protein) was aliquoted and frozen on dry ice until further use. The remaining pellet containing the proteins associated with insoluble material was resuspended in extraction buffer prior to SDS-PAGE and Western blot analyses. Protein content was determined using the DC Protein assay (Bio-Rad). Note, that focused microwave (FM) fixation deactivates all proteins in the brain and renders virtually all proteins insoluble. This precludes FM tissue from separate analyses of soluble and insoluble protein fractions.

### Immunodetection of proteins

Equal amounts of protein were boiled in 5X SDS sample buffer (final concentrations (1X): 2% SDS, 10% glycerol, 50 mM Tris/HCl pH6.8, 8 mM EDTA, 0.02% bromophenol blue, and 0.1 M dithiothreitol) before loading equal volumes alongside with Precision Plus Protein™ Dual Color Standards (Bio-Rad) for separation on SDS-PAGE gels (10% separation gel, 4% stacking gel). Protein amounts loaded per well were (1) 30 ug of soluble or insoluble protein lysate fraction or 20 ug of total protein lysate for GYS1 detection or (2) 30 ug of soluble protein lysate fraction for CHI3L1 detection. Separated proteins were transferred to activated PVDF membranes via wet transfer at 4 °C. Immediately after the transfer, total proteins were visualized using Revert 700 Total Protein Stain (LI-COR, Cat.# 92611021) following the manufacturer's instructions. The standard immunodetection protocol included blocking in 5% (w/v) non-fat milk (Bio-Rad) in TBS-T for 1 h, the addition of primary antibodies, incubation for 1 h at RT (anti-GYS1) or overnight at 4 °C (anti-YKL40/CHI3L1), incubation with secondary antibody in 5% milk in TBS-T for 1 h, and visualization of horseradish peroxidase (HRP) activity applying the Clarity™ Western ECL Substrate (Bio-Rad) and using a ChemiDoc gel imager (Bio-Rad). Protein bands were quantified using ImageLab software (Bio-Rad). Finally, protein band intensity was normalized against the total protein lane. Primary antibodies anti-GYS1, anti-phospho-GYS1, and anti-CHI3L1 were diluted 1:1000. Secondary donkey anti-rabbit antibodies were diluted 1:4000 (GYS1 detection) or 1:1000 (CHI3L1 detection).

### mRNA expression analyses

Aliquots of frozen liquid nitrogen-ground brain tissue were homogenized using a 1 cc syringe and 21 g needle in TRiZol reagent (Invitrogen, 15596018) and RNA purified using a PureLink RNA Mini-Kit (Invitrogen, 12183018A) following the manufacturer's instructions. Genomic DNA was digested (on column) using the PureLink DNase kit (Invitrogen Cat# 12185010). cDNA was synthesized from 1 μg RNA using the iScript Reverse Transcription

SuperMix (Bio-Rad, #1708840). qPCR was performed using iTaq Universal SYBR Green Master Mix (Bio-Rad, 1725121) with a QuantStudio 7 Pro real-time cycler (Thermo Fisher Scientific). Cycling conditions were 95 °C for 30 s, followed by 40 cycles of 95 °C for 15 s and 60 °C for 1 min, completed by a melt curve analysis using a temperature range from 95 to 60 °C. Primer efficiency analyses were conducted for each primer pair used (Reagent and Tools Table). *Ppia* and *Rpl4* were used as reference genes. ΔCt values were determined, calculating $Ct_{gene\ of\ interest}$ - $Ct_{reference\ gene}$ (geometric mean was used for two reference genes), followed by transformation into $2^{-\Delta Ct}$. Expression levels were further normalized to the respective WT group average. All qPCR primers are listed in the Reagent and Tools Table.

## Histology

Paraffin-embedded brain tissue from cryopreserved 5-week-old LKO, MKO, and their respective WT (cohorts 5 and 6) was sectioned and stained using the periodic acid-Schiff diastase (PASD) method to visualize polyglucosan bodies. Embedding, sectioning, and staining was performed by the University of Texas Southwestern Medical Center Histo Pathology Core. Stained slides were scanned using a Hamamatsu Nanozoomer 2.0 HT digital slide scanner (40x objective) and representative images were taken.

## Metabolite extraction and detection

Metabolites were extracted from FM-fixed brain tissue, ground to a fine powder using a liquid nitrogen-cooled mortar and pistil, using cold 80% methanol. About 70 mg tissue aliquots were resuspended well in 1 mL of the solvent. The suspension was diluted 1:5 in a cold solvent, with extraction continuing in the cold by extensive intermittent vortexing prior to centrifugation of a defined volume (14,000 × g, 4degC, 10 min) and drying of 1 mL of supernatant in a speedvac. The pellet was extracted in 1 mL 1 M NaOH, and protein was determined using the Pierce BCA protein assay (Cat.# 23225). Protein values were later used for the normalization of metabolic profiles. After submission of dried extracts to the Metabolomics Facility at Children's Medical Center Research Institute at UT Southwestern data acquisition was performed by reverse-phase chromatography on a 1290 UHPLC liquid chromatography (LC) system interfaced to a high-resolution mass spectrometry (HRMS) 6550 iFunnel Q-TOF mass spectrometer (MS) (Agilent Technologies, CA). The MS was operated in both positive and negative (ESI+ and ESI-) modes. Analytes were separated on an Acquity UPLC® HSS T3 column (1.8 μm, 2.1 × 150 mm, Waters, MA). The column was kept at room temperature. Mobile phase A composition was 0.1% formic acid in water and mobile phase B composition was 0.1% formic acid in 100% ACN. The LC gradient was 0 min: 1% B; 5 min: 5% B; 15 min: 99%; 23 min: 99%; 24 min: 1%; 25 min: 1%. The flow rate was 250 μL min⁻¹. The sample injection volume was 5 μL. ESI source conditions were set as follows: dry gas temperature 225 °C and flow 18 L min⁻¹, fragmentor voltage 175 V, sheath gas temperature 350 °C and flow 12 L min⁻¹, nozzle voltage 500 V, and capillary voltage +3500 V in positive mode and −3500 V in negative. The instrument was set to acquire over the full m/z range of 40–1700 in both modes, with the MS acquisition rate of 1 spectrum s⁻¹ in profile format. Raw data files (.d) were processed using Profinder B.08.00 SP3 software (Agilent Technologies, CA) with an in-house database containing

retention time, accurate mass information, and isotopic pattern on 600 standards from Mass Spectrometry Metabolite Library (IROA Technologies, MA) which was created under the same analysis conditions. The in-house database matching parameters were: mass tolerance of 0 ppm; retention time tolerance of 0.5 min. Peak integration result was manually curated in Profinder for improved consistency and exported as a spreadsheet (.csv). Peak intensities were protein-normalized across samples.

## Metabolic profile analyses

Protein-normalized metabolic profiles generated in this study were analyzed using MetaboAnalyst 5.0 modules for statistical and pathway analysis (Pang et al, 2022), with data exported for graph generation in Microsoft Excel. Biological replicates (profiles from individual animals) were as follows: WT1 $n = 9$, LKO $n = 7$, DKO1 $n = 10$, PKO1 $n = 8$, WT2 $n = 11$, MKO $n = 9$, DKO2 $n = 10$, PKO2 $n = 6$. For Gbe1YS mice and respective WT controls $n = 12$. MetaboAnalyst 5.0 analyses included the calculation of fold changes and $p$ value for all metabolites when comparing genotype groups. Data sets were autoscaled (1) for unsupervised cluster analyses and the generation of heatmaps, which included only the metabolites that were significantly changed in comparison to WT controls ($p < 0.05$), and (2) for principal component analysis (PCA) as well as for partial least squares-discriminant analysis (PLS-DA) with variable importance in projection (VIP) scores calculated for the 15 most influential metabolites. Points in PCA and PLS-DA biplots each represent one mouse with (1) a 95% confidence interval plotted around each genotype group, and (2) loadings of the 15 most influential metabolites being displayed as arrows. Pathway analyses were conducted comparing the autoscaled individual metabolite profiles between two indicated genotype groups. Analysis parameters were always set as follows: global test as enrichment method, relative-betweenness centrality for topology analysis, with all compounds in the Mus musculus pathway library of the Kyoto Encyclopedia of Genes and Genomes (KEGG) used for reference. False discovery rate-adjusted $p$ values (q) were plotted against impact score of individual pathways, with impact a measure of importance of the quantified metabolites in the pathway. A pathway was considered and marked as significantly changed if it was represented by ≥2 metabolites in the metabolomics dataset and with impact $>0$ and $q < 0.05$ (red dashed line in the pathway analysis plots). To compare pathway changes occurring in two different genotype comparisons, two-dimensional pathway analyses were conducted, where $q$ values of one genotype comparison was plotted against those from another comparison (e.g., $q$ of LKO vs. WT plotted against q of DKO vs. WT). To quantify the severity and direction of a pathway change, areas in the volcano plot for all individual metabolites were calculated by multiplication of log2(-fold-change) and -log10(p). The average direction and severity of select pathways were estimated by calculating the mean area in the volcano plot (comparing two genotypes) for all detected metabolites in the corresponding pathways. Volcano plot area of individual metabolites from select pathways were then plotted for two genotype-genotype comparisons to allow inspection of genotype impacts on individual metabolites in select pathways. A schematic in Fig. 4I illustrates for hypothetical pathways A and B how the mean volcano area of metabolite hits are calculated from hypothetical volcano plots.

## Statistical analysis

Data were presented as mean ± SEM with biological replicates (n) indicated in figure legends. For CLD analyses, mean ± SD is displayed. For comparison of four groups differing in the genetic status of two genes (laforin and PTG or malin and PTG) two-way ANOVA with Tukey post hoc analysis was conducted, except for CLD data, where one-way ANOVA was conducted for each individual chain length with post hoc analysis using unpaired two-tailed $t$-tests and Holm–Bonferroni correction. For comparison of two groups, parametric unpaired two-tailed $t$-tests with Welch's correction were used. Analyses of metabolome data sets was conducted with MetaboAnalyst 5.0, including the calculation of raw $p$ values and false discovery rate-adjusted $p$ values (q) as well as fold changes. Individual statistical tests employed are indicated in all figure legends.

## Data availability

Source data for the generation of Figs. 3, 4 contains protein-normalized relative abundances of brain metabolites in (A) genotype groups WT1, LKO, DKO1, PKO1 (used for the generation of Figs. 3, 4 and EV1–3) and (B) genotype groups WT2, MKO, DKO2, PKO2 (used for the generation of Figs. 3, 4 and EV1–3). The dataset is available online at Mendeley Data (https://doi.org/10.17632/vv9n57g87g.1).

Source data for the generation of Fig. 5 contains protein-normalized relative abundances of brain metabolites in genotype groups WT and Gbe1YS (used for the generation of Figs. 5 and EV4). The dataset is available online at Mendeley Data (https://doi.org/10.17632/sw4r4m66gk.1).

All other data reported in this paper are included in this publication as source data files. Any additional information required to reanalyze the data reported in this paper is available from the lead contact upon request.

The source data of this paper are collected in the following database record: biostudies:S-SCDT-10_1038-S44318-024-00339-3.

## Peer review information

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

## Acknowledgements

The work followed The ARRIVE guidelines 2.0 and was funded by NIH grants (P01NS097197 to BAM and R01 NS128437 to FN), and by Chan-Zuckerberg Initiative Patient-Partnered Collaborations for Rare Neurodegenerative Disease grant (2022-316703 to BAM and FN). BAM holds the University of Texas Southwestern Jimmy Elizabeth Westcott Chair in Pediatric Neurology. We also thank Dr. Jennifer Kohler (Biochemistry Department, UT Southwestern Medical Center) for granting access to the HPAEC-PAD equipment and Dr. Hieu Vu (St. Jude Children's Research Hospital) for helpful advice on metabolomics data processing. We thank UT Southwestern's HistoPathology Core, Whole Brain Microscopy Facility, Metabolomics Facility at Children's Medical Center Research Institute, and Preclinical Pharmacology Core for histological staining, slide scanning services, metabolic analyses, and Glc4 determinations, respectively.

## Author contributions

**Silvia Nitschke**: Conceptualization; Data curation; Formal analysis; Supervision; Validation; Investigation; Visualization; Methodology; Writing—original draft; Writing—review and editing. **Alina P Montalbano**: Data curation; Formal analysis; Supervision; Investigation; Visualization; Methodology; Writing—review and editing. **Megan E Whiting**: Formal analysis; Investigation. **Brandon H Smith**: Formal analysis; Investigation. **Neije Mukherjee-Roy**: Formal analysis; Investigation; Writing—review and editing. **Charlotte R Marchioni**: Formal analysis; Investigation. **Mitchell A Sullivan**: Conceptualization; Formal analysis; Investigation; Writing—review and editing. **Xiaochu Zhao**: Formal analysis; Investigation. **Peixiang Wang**: Investigation. **Howard Mount**: Resources; Investigation; Methodology; Writing—review and editing. **Mayank Verma**: Conceptualization; Formal analysis; Investigation; Methodology; Writing—review and editing. **Berge A Minassian**: Conceptualization; Supervision; Funding acquisition; Project administration; Writing—review and editing. **Felix Nitschke**: Conceptualization; Resources; Data curation; Formal analysis; Supervision; Funding acquisition; Validation; Investigation; Visualization; Methodology; Writing—original draft; Project administration; Writing—review and editing.

Source data underlying figure panels in this paper may have individual authorship assigned. Where available, figure panel/source data authorship is listed in the following database record: biostudies:S-SCDT-10_1038-S44318-024-00339-3.

## Disclosure and competing interests statement

The authors declare no competing interests.

# Expanded View Figures

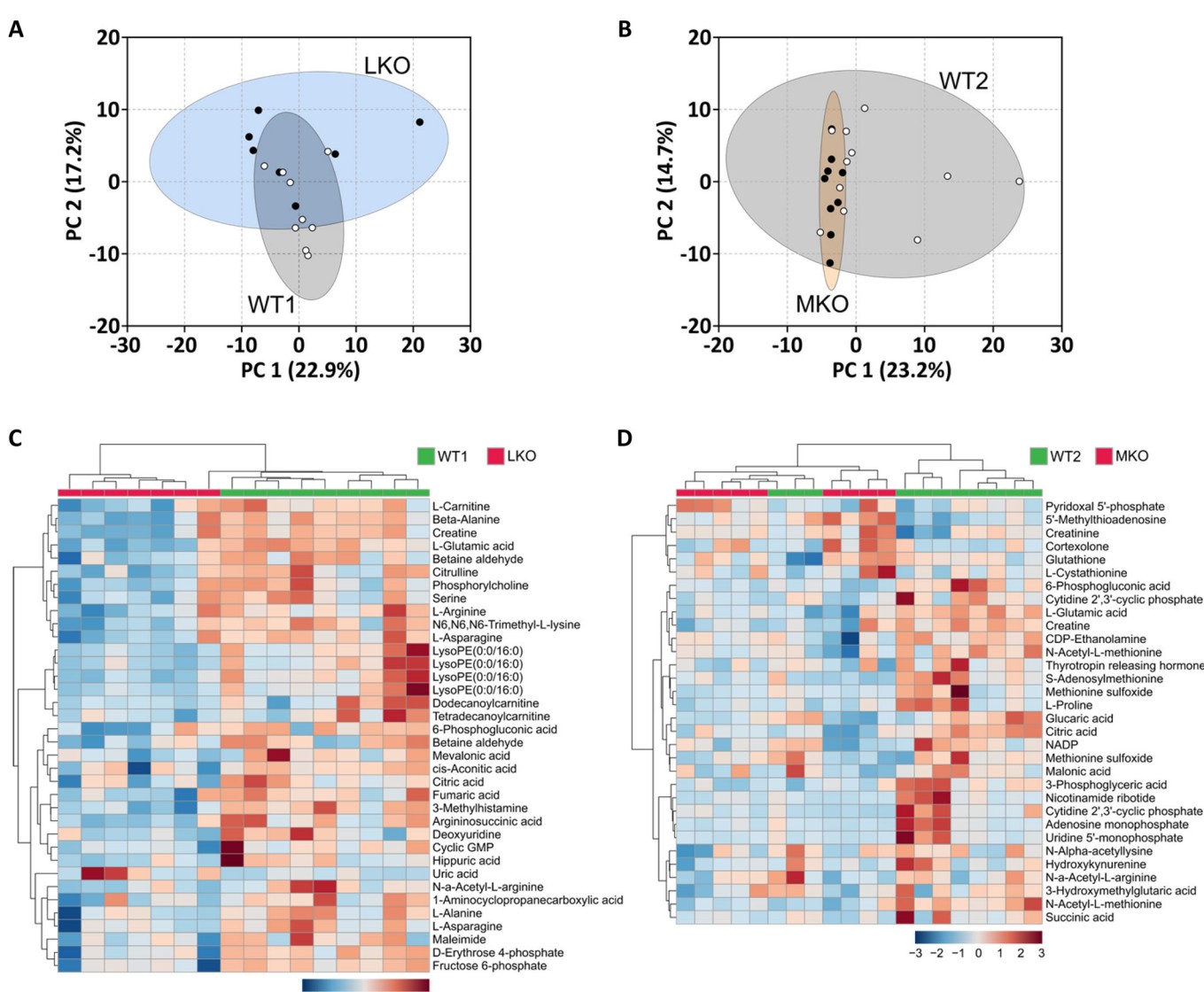

**Figure EV1. Cluster analyses using brain metabolite profiles from LD mice showing no or incomplete clustering.**

(A,B) PCA plots of metabolic profiles in LKO (A) and MKO (B) compared to respective WTs. (C, D) Heatmaps, comparing all significantly changed ($p < 0.05$) brain metabolites in LKO (C) and MKO (D) to respective WTs in individual animals from unsupervised Ward cluster analysis with Euclidean distance measure based on unpaired two-tailed *t*-test. Data information: Significance levels determined by MetaboAnalyst 5.0 as follows: $p < 0.05$ (C, D), $n = 7$–11. Corresponding to Fig. 3. Source data are available online for this figure.

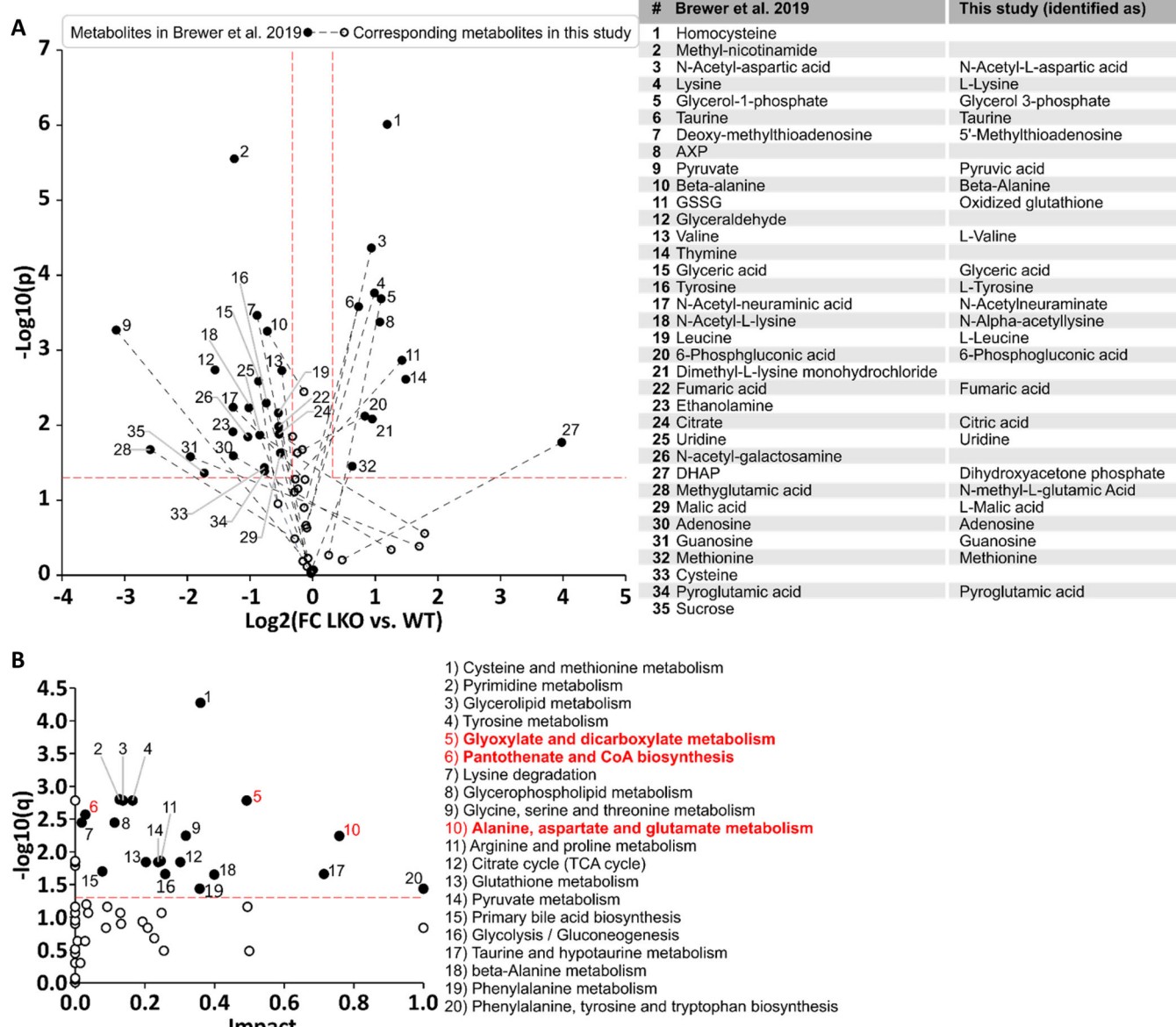

**A**

| # | Brewer et al. 2019 | This study (identified as) |
|---|---|---|
| 1 | Homocysteine | |
| 2 | Methyl-nicotinamide | |
| 3 | N-Acetyl-aspartic acid | N-Acetyl-L-aspartic acid |
| 4 | Lysine | L-Lysine |
| 5 | Glycerol-1-phosphate | Glycerol 3-phosphate |
| 6 | Taurine | Taurine |
| 7 | Deoxy-methylthioadenosine | 5'-Methylthioadenosine |
| 8 | AXP | |
| 9 | Pyruvate | Pyruvic acid |
| 10 | Beta-alanine | Beta-Alanine |
| 11 | GSSG | Oxidized glutathione |
| 12 | Glyceraldehyde | |
| 13 | Valine | L-Valine |
| 14 | Thymine | |
| 15 | Glyceric acid | Glyceric acid |
| 16 | Tyrosine | L-Tyrosine |
| 17 | N-Acetyl-neuraminic acid | N-Acetylneuraminate |
| 18 | N-Acetyl-L-lysine | N-Alpha-acetyllysine |
| 19 | Leucine | L-Leucine |
| 20 | 6-Phosphgluconic acid | 6-Phosphogluconic acid |
| 21 | Dimethyl-L-lysine monohydrochloride | |
| 22 | Fumaric acid | Fumaric acid |
| 23 | Ethanolamine | |
| 24 | Citrate | Citric acid |
| 25 | Uridine | Uridine |
| 26 | N-acetyl-galactosamine | |
| 27 | DHAP | Dihydroxyacetone phosphate |
| 28 | Methyglutamic acid | N-methyl-L-glutamic Acid |
| 29 | Malic acid | L-Malic acid |
| 30 | Adenosine | Adenosine |
| 31 | Guanosine | Guanosine |
| 32 | Methionine | Methionine |
| 33 | Cysteine | |
| 34 | Pyroglutamic acid | Pyroglutamic acid |
| 35 | Sucrose | |

**B**

1) Cysteine and methionine metabolism
2) Pyrimidine metabolism
3) Glycerolipid metabolism
4) Tyrosine metabolism
5) **Glyoxylate and dicarboxylate metabolism**
6) **Pantothenate and CoA biosynthesis**
7) Lysine degradation
8) Glycerophospholipid metabolism
9) Glycine, serine and threonine metabolism
10) **Alanine, aspartate and glutamate metabolism**
11) Arginine and proline metabolism
12) Citrate cycle (TCA cycle)
13) Glutathione metabolism
14) Pyruvate metabolism
15) Primary bile acid biosynthesis
16) Glycolysis / Gluconeogenesis
17) Taurine and hypotaurine metabolism
18) beta-Alanine metabolism
19) Phenylalanine metabolism
20) Phenylalanine, tyrosine and tryptophan biosynthesis

**C**

| Duran et al. 2021 | Change in MKO | This study (identified as) | MKO vs. WT2 FC | MKO vs. WT2 p | DKO2 vs. WT2 FC | DKO2 vs. WT2 p | |
|---|---|---|---|---|---|---|---|
| Glycine | down | Glycine | 1.04 | 0.847 | 0.99 | 0.975 | |
| Creatinine | up | Creatinine | 1.14 | 0.041 * | 1.02 | 0.824 | Rescued in DKO |
| Ribose-5-phosphate | up | D-Ribose 5-phosphate | 0.37 | 0.151 | 0.46 | 0.204 | |
| 6-Phosphogluconic acid | up | 6-Phosphogluconic acid | 0.81 | 0.003 ** | 0.81 | 0.004 ** | |
| Glycolic acid | up | Glycolate | 0.60 | 0.101 | 0.66 | 0.212 | |
| Alanine | up | L-Alanine | 1.02 | 0.767 | 0.92 | 0.129 | |
| Lactate | up | L-Lactic acid | 0.71 | 0.054 | 0.78 | 0.138 | |
| Hexose | up | D-Glucose | 1.01 | 0.925 | 0.91 | 0.114 | |
| Lactamide | up | not detected | | | | | |
| Cysteine | up | not detected | | | | | |

FC
0.33          0.0          3.0

**Figure EV2.  Meta-analysis of our metabolomic data and published data from CP-fixed LD brains shows no common metabolic signature.**

(A) Volcano plot of the 35 metabolites significantly changed ($p < 0.05$, FC >1.25 or <0.8) in CP-fixed LKO brains in Brewer et al dataset (Brewer et al, 2019), displayed as filled circles and listed next to the plot. 25 of these metabolites were measured in the current study, are displayed as empty circles, and connected to the corresponding data point from Brewer et al using dashed lines. $P$ values were calculated by unpaired two-tailed $t$-test. (B) Pathway analysis for published LKO dataset (Brewer et al, 2019). Changed pathways are displayed as filled circles and numbered according to the list next to the plot. Pathways 5, 6, and 10 (red) were also significantly changed in LKO in the current study (see Fig. 3E). (C) Table comparing ten metabolites found significantly changed in CP-fixed MKO brains in Duran et al (Duran et al, 2021) with identical metabolites detected in our study with fold changes (FC) and $p$ values for MKO or DKO2, respectively, compared to WT2 ($p > 0.05$ or FC of <1.25/>0.8). $P$ values were calculated by unpaired two-tailed $t$-test. Data information: Significance levels determined by MetaboAnalyst 5.0 as follows: $p < 0.05$ (A, B) *$p < 0.05$, **$p < 0.01$ (C, red), $n = 7$–11. Corresponding to Fig. 3. Source data are available online for this figure.

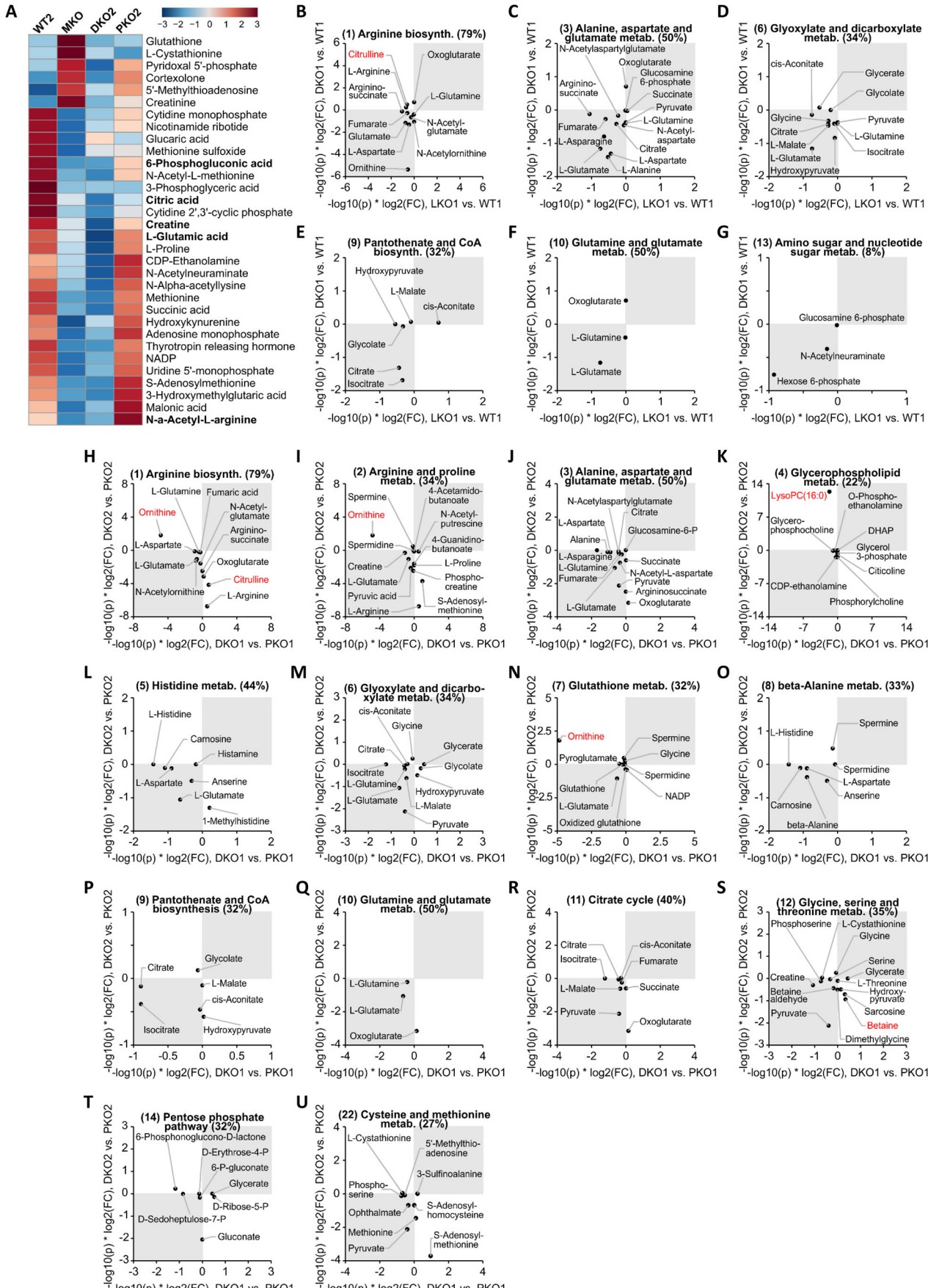

◄  **Figure EV3.  Metabolite and pathway changes in LKO and MKO not rescued by PTG knockout.**

(A) Heatmap for malin cohort, comparing all significantly changed metabolites in MKO between the four genotypes. Metabolites significantly changed in both LD mice in bold. (B–G) Volcano plot areas of all detected metabolites from pathways #1 (B), #3 (C), #6 (D), #9 (E), #10 (F), and #13 (G) used for calculation of mean areas in Fig. 4J plotted for LKO vs. WT1 comparison with DKO1 vs. WT1. *P* values for each metabolite were calculated by unpaired two-tailed *t*-test. (H–U) Volcano plot areas of all detected metabolites from pathways #1 (H), #2 (I), #3 (J), #4 (K), #5 (L), #6 (M), #7 (N), #8 (O), #9 (P), #10 (Q), #11 (R), #12 (S), #14 (T), and #22 (U) used for calculation of mean areas in Fig. 4K plotted for DKO1 vs. PKO1 comparison with DKO2 vs. PKO2. *P* values for each metabolite were calculated by unpaired two-tailed *t*-tests. Data information: Off-centered dots in **B–U** correspond to metabolites with increased fold-change (FC) and/or significance. Red-font metabolites were significantly changed in both genotype comparisons but with different directionality. Pathway numbers as in Fig. 4A. Corresponding to Fig. 4. Percentages indicate detected metabolites of all metabolites in the pathway. Source data are available online for this figure.

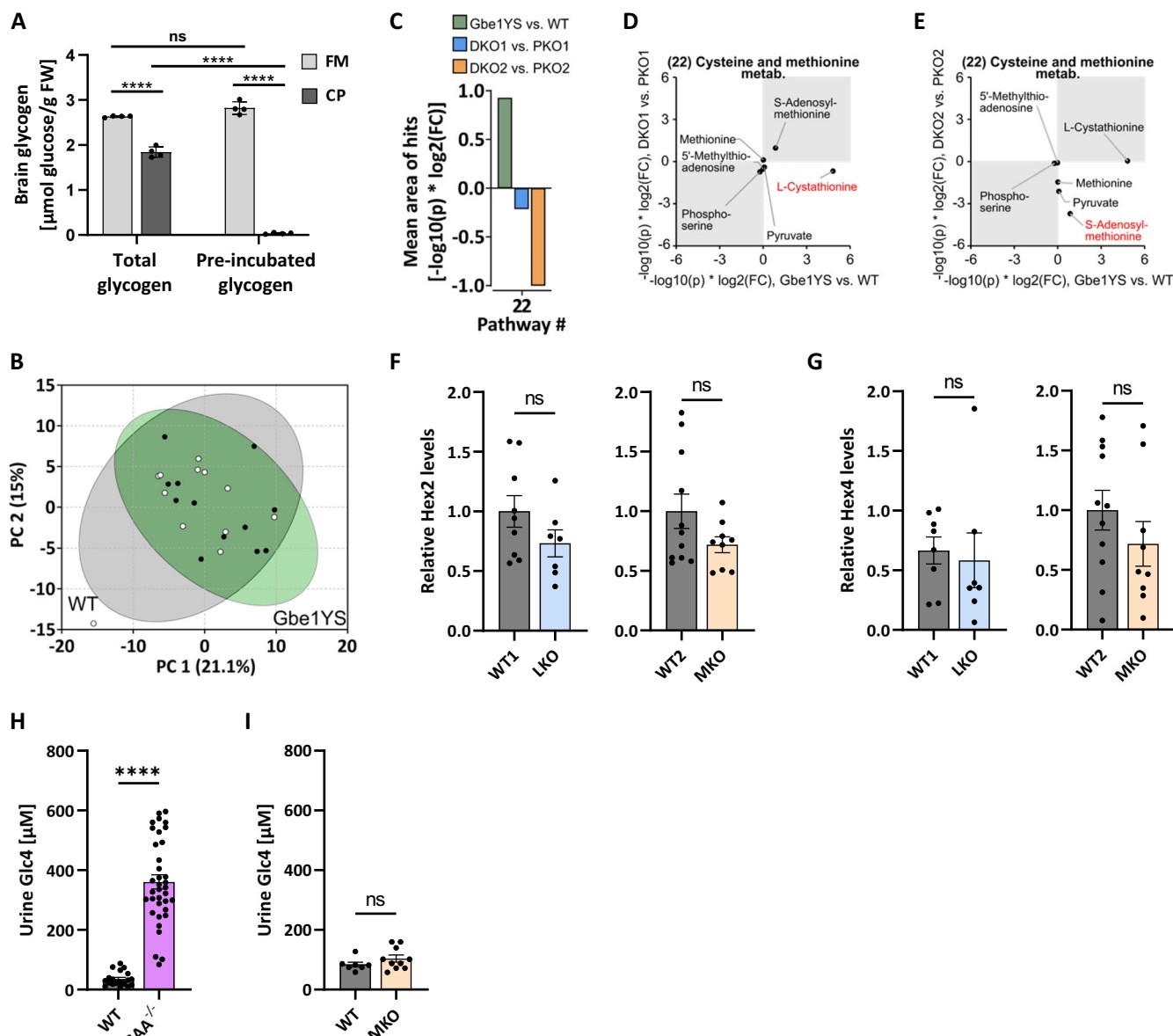

**Figure EV4.** **Increased Hex4/Glc4 levels in Gbe1YS, but not LD mice are comparable to changes in Pompe disease.**

(A) Brain total and preincubated glycogen in CP- and FM-fixed brains of WT mice from Gbe1YS cohort. $n = 4$. (B) PCA plot of metabolic profile in Gbe1YS compared to respective WT. $n = 12$. (C) The mean area of hits to show directionality of change (calculations as shown in Fig. 4I) displayed for pathway #22 that was significantly changed in all three indicated genotype comparisons. P values for each metabolite in the pathway were calculated by unpaired two-tailed t-tests. (D, E) Volcano plot areas of metabolites from pathway #22 used for calculation of mean areas in G plotted for Gbe1YS ($n = 12$) vs. WT ($n = 12$) comparison with DKO1 ($n = 10$) vs. PKO1 ($n = 8$) (D) or DKO2 ($n = 10$) vs. PKO2 ($n = 6$) (E), respectively. Off-centered dots correspond to metabolites with increased fold-change (FC) and/or significance. Red-font metabolites were significantly changed in both genotype comparisons but with different directionality. P values for each metabolite were calculated by unpaired two-tailed t-tests. (F, G) Relative metabolite levels of hexose disaccharide Hex2 (F) and hexose tetrasaccharide Hex4 (G) in LKO and MKO mice compared to their respective WTs. $n = 7$–11. (H, I) Urine Glc4 levels in Pompe disease mice GAA$^{-/-}$ (H) and MKO mice (I). $n = 19$–36 (H), $n = 7$–10 (I). Data information: Data in (A) and (F–I) are presented as mean ± SEM. ****$p < 0.0001$ by two-way ANOVA with Tukey post hoc analysis (A) or by Welch's t-test (F–I); ns not significant, CP cryopreservation, FM focused microwave. Corresponding to Fig. 5. Source data are available online for this figure. P values: (A) Tot. (FM vs. CP) 2.2E-07, Pre. (FM vs. CP) 2.4E-13, CP (Tot. vs. Pre.) 7.6E-12, FM (Tot. vs. Pre.) 0.058; (H) <1.0E-15. Source data are available online for this figure.

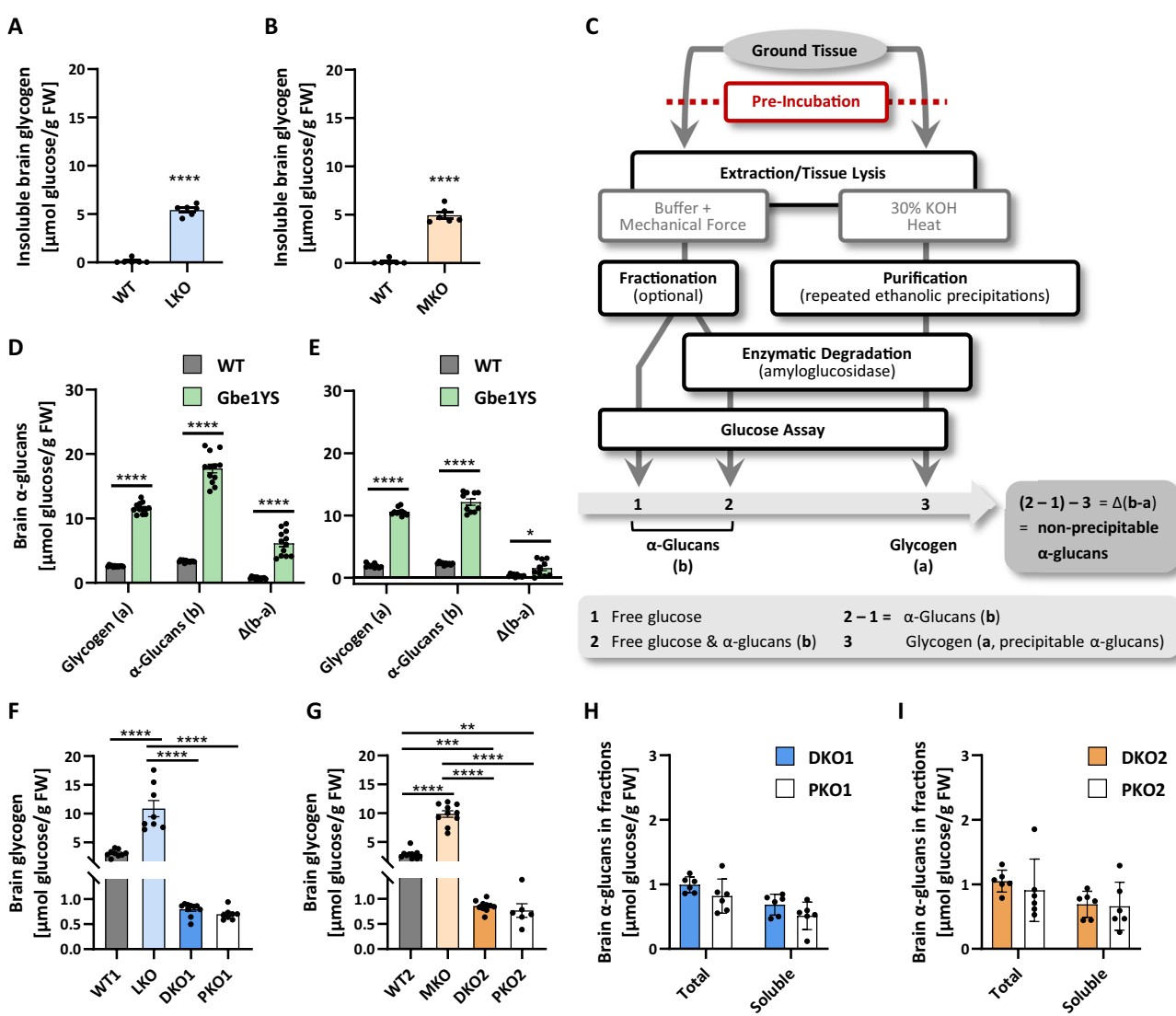

**Figure EV5.  Metabolically volatile α-glucan is present in Gbe1YS but absent in DKO mice showing rescue compared to LD mice.**

(A, B) Insoluble (preincubated) glycogen in LKO (A) and MKO (B) mice after CP fixation. $n = 6$. (C) Schematic showing methodology of glycogen determination as precipitable glycogen (a) or α-glucan (b) which additionally contains lower molecular weight glucans (≥2 glucose units). (D, E) Brain α-glucans in Gbe1YS mice after FM (D) or CP (E) fixation. Terminology is explained in (C). $n = 10$–12. (F, G) Brain total glycogen following FM fixation in laforin (F) and malin (G) cohort as in Fig. 1C but with different y-axis scale. $n = 6$–11. (H, I) Total and soluble brain α-glucan in PKO and DKO from the laforin (H) and malin (I) cohort after FM fixation. $n = 6$. Data information: All data were presented as mean ± SEM. *$p < 0.05$; **$p < 0.01$; ***$p < 0.001$; ****$p < 0.0001$ by two-way ANOVA with Tukey post hoc analysis (F, G) or by Welch's *t*-test (A–E, H, I). Corresponding to Fig. 6. *P* values: (A) 1.7E-07; (B) 1.03E-05; (D) Glyc. 3.4E-13, α-Gluc. 2.6E-10, Δ(a-b) 9.7E-07; (E) Glyc. <1.0E-15, α-Gluc. 7.5E-09, Δ(a-b) 0.0147; (F) WT1 vs. LKO 1.3E-08, LKO vs. DKO1 1.7E-11, LKO vs. PKO1 4.9E-11; (G) WT2 vs. MKO 5.6E-14, MKO vs. DKO2 5.5E-14, MKO vs. PKO2 5.5E-14, WT2 vs. DKO2 0.00063, WT2 vs. PKO2 0.0022. Source data are available online for this figure.

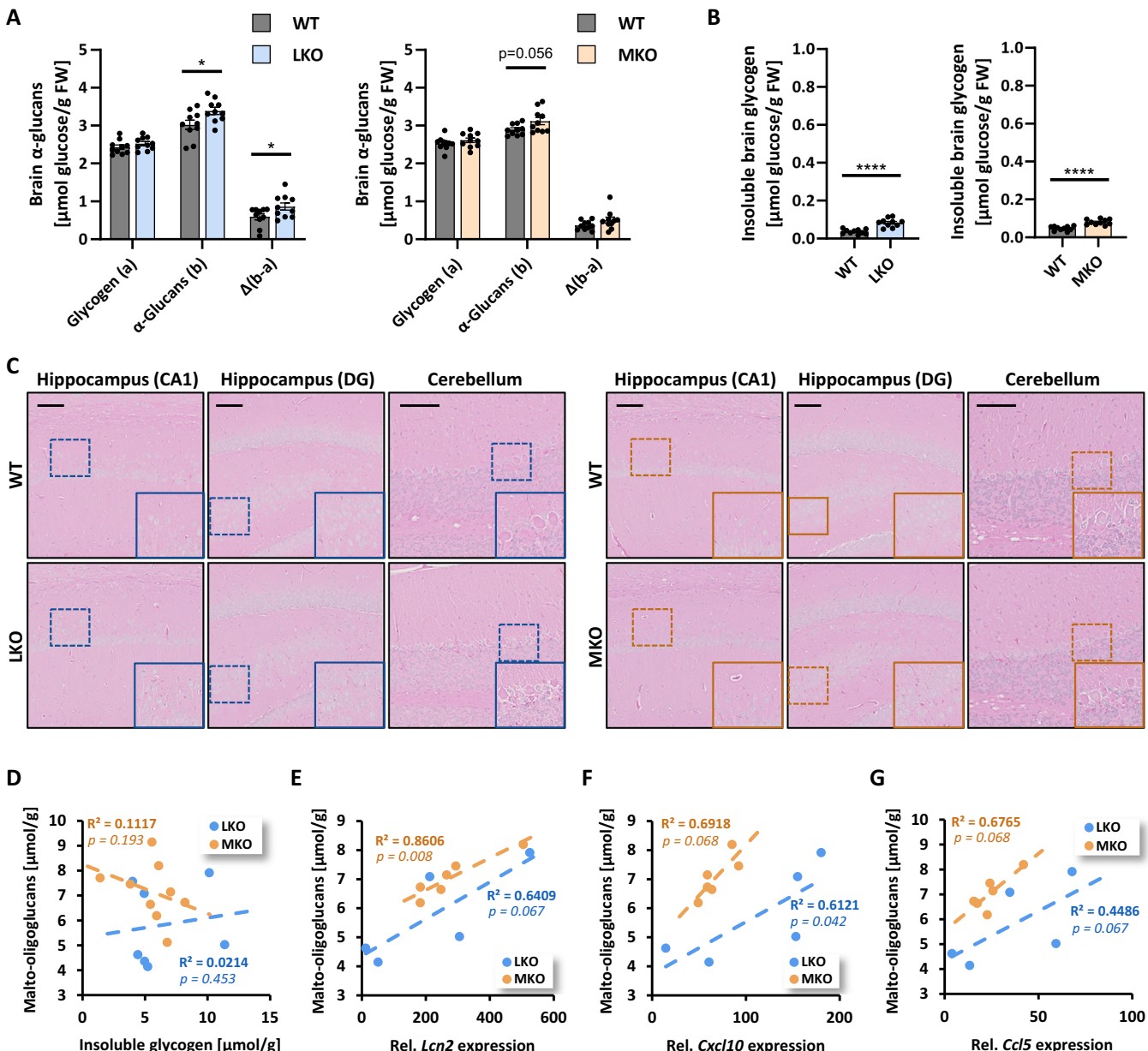

**Figure EV6. Malto-oligoglucan accumulation only found in aged LD mice correlates well with neuroinflammatory marker expression.**

(**A**) Brain α-glucans in 5-week-old LKO and MKO, respectively, compared to the respective WT after FM fixation. Terminology explained in Fig. EV5C. $n = 10$. (**B**) Insoluble (preincubated) glycogen in 5-week-old LKO and MKO mice, respectively after CP fixation. $n = 10$. (**C**) Representative PASD images of hippocampus and cerebellum in LKO and MKO mice, respectively. Scale bar, 100 μm. The region highlighted by small square is shown enlarged at the right bottom of each image (bigger squares outlined in blue or orange). (**D–G**) Correlation coefficients ($R^2$) between malto-oligoglucans and insoluble glycogen (**D**) or neuroinflammatory markers *Lcn2* (**E**), *Cxcl10* (**F**), and *Ccl5* (**G**), respectively, in aged LKO and MKO. $n = 5$–9. Data information: Data were presented as mean ± SEM (**A, B**) or individual data points (**D–G**). \*$p < 0.05$; \*\*\*\*$p < 0.0001$ by Welch's *t*-test. Corresponding to Fig. 6. *P* values: (**A**) α-Gluc.0.0265, Δ(a-b) 0.0439; (**B**) WT vs. LKO 5.0E-05, WT vs. MKO 2.5E-05. Source data are available online for this figure.

