## [Peer Review File · The EMBO Journal]

Glycogen synthase GYS1 overactivation contributes to glycogen insolubility and malto-oligoglucan-associated neurodegenerative disease

Silvia Nitschke, Alina Montalbano, Megan Whiting, Brandon Smith, Neije Mukherjee-Roy, Charlotte Marchioni, Mitchell Sullivan, Xiaochu Zhao, Peixiang Wang, Howard Mount, Mayank Verma, Berge Minassian, and Felix Nitschke

Corresponding author(s): Felix Nitschke (felix.nitschke@utsouthwestern.edu)

Review Timeline:

Submission Date:	1st May 24
Editorial Decision:	5th Jul 24
Revision Received:	11th Oct 24
Editorial Decision:	12th Nov 24
Revision Received:	19th Nov 24
Accepted:	20th Nov 24

Editors: Kelly M Anderson and Ioannis Papaioannou

Transaction Report:

Dear Prof. Nitschke,

Thank you for submitting your manuscript for consideration by the EMBO Journal. It has now been seen by three referees whose comments are shown below.

Given the referees' recommendations, I would like to invite you to submit a revised version of the manuscript, addressing the comments of all three reviewers. I should add that it is EMBO Journal policy to allow only a single round of revision, and acceptance of your manuscript will therefore depend on the completeness of your responses in this revised version. It would be good to discuss your plan to address the referee concerns and I am available to do so via zoom or email in the coming weeks.

Thank you for the opportunity to consider your work for publication. I look forward to your revision.

Yours sincerely,

Kelly M Anderson, PhD
Editor, The EMBO Journal
k.anderson@embojournal.org

Please remember: Digital image enhancement is acceptable practice, as long as it accurately represents the original data and conforms to community standards. If a figure has been subjected to significant electronic manipulation, this must be noted in the figure legend or in the 'Materials and Methods' section. The editors reserve the right to request original versions of figures and

the original images that were used to assemble the figure.

We realize that it is difficult to revise to a specific deadline. In the interest of protecting the conceptual advance provided by the work, we recommend a revision within 3 months (3rd Oct 2024). Please discuss the revision progress ahead of this time with the editor if you require more time to complete the revisions. Use the link below to submit your revision:

Referee #1:

This work by Nitschke et al presents new data linking GYS1 overactivity, glycogen insolubility and neuroinflammation in neurological amylopectinoses. Using PTG knockout mice, in combination with malin and laforin knockout mice, the work provides further support to the conclusion in the authors previous work, which capitalized on phosphatase-mutated laforin, that glycogen chain length, governed by GYS1 activity, and not its phosphorylation status drive polyglucosan formation in the brain. The main innovations of this work are: (1) A comparative metabolomic analysis of brains from laforin, malin and Gbe knockout mice, which shows that significant metabolomic changes are not normalized by genetic rescue of the disease and are therefore not disease related; (2) Use of microwave brain fixation instead of cryopreservation to halt all rapid postmortem metabolic changes, first and foremost glycogen degradation, and thus obtain more accurate and biologically valid results; (3) The discovery of malto-oligoglucans as new biomarker for amylopectinoses. The work is very well written and presents new and exciting information on amylopectinosis pathogenesis. It merits publication in EMBO J, provided the authors address the following issues (not necessarily shown in order of importance, or place in the manuscript):

Major points

- 1) Taken together, these data indicate partly overlapping roles of malin and laforin in several metabolic pathways that are independent of PB formation and only revealed in the absence of PTG, indicating potential redundancy between laforin/malin and PTG action.
- 2) YKL-40 can be a cue for both anti- and pro- inflammatory processes. Can the authors please reconcile these differences and explain how YKL-40 can induce neuroinflammation?
- 3) The subject of neuroinflammation seems to be thematically detached from the rest of the manuscript, which characterizes alpha-glucan status in brains from the three transgenic strains and their metabolomic consequences. Can the authors try to integrate the two subjects in Discussion (i.e., neuroinflammation and MOGs)?
- 4) Are there serum samples available for metabolomic analysis from the mice whose brains were analyzed? Such an analysis and its integration with the brain analyses might generate some insight, which can connect the brain data with systemic metabolic pathways and shed light on key metabolic axes, such as the brain-liver axis. It can also suggest some systemic biomarkers. One of the issues within this discussion would be differential BBB permeability to different metabolites, so that the brain and blood can communicate, and the possible involvement of compromising BBB integrity by LKO, NKO, or Gbe1YS.
- 5) Can comparative (univariate and multivariate) analyses be also made on brains from LKO vs MKO animals?
- 6) The authors mention mass tolerance and retention time as quality control criteria for mass spectrometry metabolomics analysis. Wasn't the metabolites fragmentation pattern and its comparison to that of standards used as another criterion?
- 7) For some key observations, the authors should consider presenting the results as biplots, co-plotting the scores of the different animals from each genotype and the loadings of the significant metabolites. This visualization would show how the metabolites correlate with each other, their relative contributions (according to vector length) and their PCA scores in the respective genotype. This form of presentation is usually quite informative. Please plot only significant metabolites because otherwise the visualization would be too cluttered.
- 8) While overall metabolomic changes in LKO and MKO were modest, I think that the authors should still discuss their implication on central carbon metabolism. Metabolites such as UDP-Glucose and Fructose-6-Phosphate, as well as the changes in glycogen mobilization caused by LKO, MKO, Gbe1YS, and Gbe1YS are bound to modify central carbon metabolism. The authors should therefore add in Discussion some speculation on the implications expected in glycolysis and its correlation with the TCA cycle, glucose-6-phosphate and its emanating pathways and more. Such an analysis might also suggest hypothetical biomarkers, especially systemic ones.

Minor points

- 1) p. 2: Please change to "An imbalance between the activity of both enzymes"
- 2) p. 3: "In the absence of evidence for a dysregulation of glycogen chain elongation as an explanation for glycogen insolubility in LD," Isn't misregulated chain elongation presented as a cause for glycogen insolubility in PMID: 31042462 ?
- 3) p. 3: "By modulating GYS1 activity through PTG knockout, these phenotypes can be studied in the presence and absence of PBs." Why not opting for a direct inhibition of GYS1, e.g., using CRISPR, or shRNA?
- 4) p. 3: "Allosteric binding of glucose 6-phosphate (G6P) overrides the phosphorylation-imposed inhibition, restoring full GYS1 activity (Hunter et al, 2015)." - Please also cite Bouskila wt al (2010) Cell Metab. 12:456-66. doi: 10.1016/j.cmet.2010.10.006.

- 4) p. 4: Do you know what is the glycogen degradation status in PFA fixed tissues?
- 5) p. 4: How long is the pre-incubation time?
- 5) p. 5: "We identified 36 and 32 metabolites that were significantly altered in LKO (Fig. 3A) and MKO (Fig. 3B), respectively, with no overlap between the two LD mouse models regarding metabolites with fold changes (FC) of >1.25 or <0.8 (Fig. 3A-B)." Were p values in the volcano plots adjusted for false negatives? If so, what were the FDR (q) values? Usually FDR<0.2 is acceptable.
- 6) p. 5: "Metabolic profiles of LKO/MKO and respective WTs clustered on a PLS-DA plot, though less strictly for MKO and WT (Fig. 3C)." Has PLS-DA been tested for forced classification (for instance by a permutation analysis)?
- 7) p. 5: "with on average lower VIP scores in MKO" This difference does not clearly show in figure 3D. VIP scores for both LKO and MKO range from 1.7 to 2.3."
- 8) p. 5: "The significantly changed metabolites in Brewer et al. were found unchanged in our study (Fig. EV2A). Furthermore, of the 20 significantly changed pathways in Brewer et al., we found only three changed (Fig. EV2B). Metabolic changes previously found in MKO (Duran et al., 2021) were not detected in our study (Fig. EV2C)." Can these apparent discrepancies be explained?
- 9) Figure 5A-D: Why does figure show results from WT mice? Not clear how this is related to glycogen post-mortem preservation, which should be analyzed in Gbeys/ys mice.
- 10) Fig. EV3B-G: Are these all the metabolites that participate in the respective pathway(s), or only the ones identified? Sometimes a small percentage of all metabolites in specific pathways are identified.
- 11) p. 6: "We confirmed strong accumulation of insoluble glycogen in these mice (Fig. 5A-C) with better glycogen preservation (Fig. 5A-B) and complete pre-incubation resistance after FM fixation"
- 12) p. 6: "driving the clustering plotted". I would rephrase to "contributing to metabolic variability".
- 13) p. 9: "In aged Gbe1^{YS} mice the P-GYS1/GYS1 ratio is even more dramatically decreased (Fig. 7E), further corroborating the direct link between glycogen chain length abnormality in PB diseases and the phosphorylation state of the chain-elongating enzyme GYS1." Please note that a decrease in p-GYS/GYS and subsequent increase in GYS activity was also observed in a GBE knock-down neuronal model of APBD (Kakhlon et al (2013) J Neurochem 127: 101). Please add this reference.
- 14) p. 10: "In LD mice, additional characteristics such as glycogen hyperphosphorylation, neuroinflammation, and metabolic alterations have been described (Brewer et al., 2019; Duran et al., 2021; Lahuerta et al., 2020; Tagliabracci et al., 2008).
- 15) p. 10: "indicating a globally increased activation state of GYS1 (Hunter et al., 2011; Lin et al., 2012) solely in LKO, which could be explained by..." I didn't understand how then can the authors explain GYS1 activation solely in LKO and not in MKO. The two scenarios (GSK3 β and PTG activity) are basically ruled out. Is there an alternative explanation?
- 16) p. 11: "there is no common metabolic signature in PB diseases." This conclusion might be true for the brain. However, systemic metabolic signature of PBs might still take place, which might be disclosed why serum metabolomics.
- 17) p. 11: "Other promising approaches include EPM2A gene replacement and removal of PBs by bioengineered hydrolases (Brewer et al., 2019; Zafra-Puerta et al, 2023)". Please also add here Kakhlon et al (2021) EMBO Mol Med (see above) - for APBD compassionate use of a newly discovered safe autophagic activator with an FDA orphan designation also mentioned in Wu et al (2024) EMBO Mol Med (see above).

Referee #2:

In this work, Nitschke et al. have analyzed the effect of PTG k.o. on glycogen structure and level of phosphorylation. Relate to those analysis, the use of a novel reported fixation method has prevented brain glycogen post-mortem degradation, occurring when other methods have been used and more reliable results were obtained. In addition, they have analyzed the consequences of PTG depletion or brain metabolome or in the presence of some neuroinflammatory markers. The studies were done on Lafora Disease (LD) mouse models. The title is clear but other parts of the article are less clear and should be further clarified.

Specific points:

- In the Introduction, for the general audience, malto oligoglucans should be introduced.
- Figure 1 shows that PTG k.o. prevents glycogen accumulation in LD mouse models. This is not a novel observation, as indicated in the Introduction. Thus, the effect of PTG k.o. was focused on glycogen chain length (related to insolubility) and

hyperphosphorylation. The obtained results confirm the previous observations found in a mouse with a defective laforin, indicating that hyperphosphorylation and insolubility are uncoupled (see Gayarre 20014 or the work of the authors of this submitted manuscript).

In summary, this is a confirmatory data, that perhaps could be shortened in part.

- Figure 2 shows that PTG k.o. prevents LD-related neuroinflammation and analysis of pro and anti-inflammatory genes were carried out. The analyses were properly performed but the conclusion is mainly descriptive. It may require further explanations for a possible mechanism.

- Figure 3-Figure 6 are mainly focused on the obtained results using the reported fixation method and it has facilitated the comparison with the previous data from other groups. This is a positive point. However, there are many "crude" data. For example, Figure 3A shows the "crude" data for 20 metabolites. Figure 3B shows that for 19 metabolites. Figure 4A indicates 29 different pathways.

A clearer example could be found in Figure 4I. It indicates pathway A and pathway B. Nothing in the text (mentioned six pathways) or in the legend of the Figure is indicated about the meaning of pathway A and pathway B.

I suggest clearer conclusions rather than quantity or too many "unexplained" data.

- In summary, the article could be improved by making a clear scheme with the main (not many) conclusions, showing (end of the Discussion) one by one, the cause-effect relationship between: a) Glycogen chain elongation, b) glycogen chain branching, c) glycogen solubility, d) glycogen phosphorylation with A) metabolic dysregulation or B) neuroinflammation, like markers of LD. Now that cause-effect relationship is slightly confused and should be clarified.

Minor points:

- The analyses showing the level of GSK3 β P-Ser 9, like a control, could be suitable marker after using this novel method.

Referee #3:

Polyglucosan body (PB) diseases are clinically and genetically heterogeneous diseases that cannot be classified easily to one gene or clinical phenotype. However, these very diverse diseases have one in common Diastase resistant PAS positive carbohydrate inclusions in muscle or brain. This diversity makes it difficult to diagnose and treat those diseases. GBE1 deficiency is the only form of PB disease with a very well understood mechanism yet degree of enzyme deficiency creates a wide spectrum of clinical presentation that requires careful examination and diagnostic workup. Despite the well characterization, diagnosis, treatment, and easily detectable biomarkers are not available for patients. Dr Nitschke et al, compared the mouse models of Pb diseases and affected molecular and cellular pathways using different mouse models that are shown to be related to glycogen metabolism. Their findings identified oligoglucans that explain the carbohydrate storage and use problems in the disease state. Like normal glycogen these molecules disappear quickly when tissue is devoid of blood or supply. Therefore, Nitschke et al used a microwave mediated sacrifice which instantaneously fixes tissue and proteins preventing further metabolic changes and degradation. Methods they used are reliable and experiments are carefully designed and executed. Detection of oligoglucans and their role in the disease are impressive and can be a very good source for determining the efficiency of the treatments and the stage of the disease when a treatment method becomes available. However, I have concerns about the CHI3L1/YKL40, these markers are very common biomarkers and may not reflect the state of the disease especially in the advanced age of the patients. Considering how common these two markers are, authors must limit the use of these markers to pure polyglucosan body diseases and mention them in the text that they may cause misdiagnosis when combined with other conditions including but not limited to infections and other neurodegenerative diseases.

We thank all three reviewers for their helpful input and suggestions. We addressed each point here and revised our manuscript accordingly. Please, find our responses below.

Referee #1:

This work by Nitschke et al presents new data linking GYS1 overactivity, glycogen insolubility and neuroinflammation in neurological amylopectinoses. Using PTG knockout mice, in combination with malin and laforin knockout mice, the work provides further support to the conclusion in the authors previous work, which capitalized on phosphatase-mutated laforin, that glycogen chain length, governed by GYS1 activity, and not its phosphorylation status drive polyglucosan formation in the brain. The main innovations of this work are: (1) A comparative metabolomic analysis of brains from laforin, malin and Gbe knockout mice, which shows that significant metabolomic changes are not normalized by genetic rescue of the disease and are therefore not disease related; (2) Use of microwave brain fixation instead of cryopreservation to halt all rapid postmortem metabolic changes, first and foremost glycogen degradation, and thus obtain more accurate and biologically valid results; (3) The discovery of malto-oligoglucans as new biomarker for amylopectinoses. The work is very well written and presents new and exciting information on amylopectinosis pathogenesis. It merits publication in EMBO J, provided the authors address the following issues (not necessarily shown in order of importance, or place in the manuscript):

Major points

1) Taken together, these data indicate partly overlapping roles of malin and laforin in several metabolic pathways that are independent of PB formation and only revealed in the absence of PTG, indicating potential redundancy between laforin/malin and PTG action.

We thank the reviewer for this remark. To clarify our conclusions, we expanded the paragraph, now elaborating that pathway changes are not consistently present in both LD models and most severe in the absence of PTG (and PBs). We therefore conclude that abnormal metabolic flux is independent of PB formation. Pathway changes being virtually absent in PKO and MKO but enhanced in DKO1 and only present in DKO2 suggest redundancy between laforin, malin, and PTG action in preventing abnormal metabolic flux. We added a model as a new Fig. 8B.

2) YKL-40 can be a cue for both anti- and pro- inflammatory processes. Can the authors please reconcile these differences and explain how YKL-40 can induce neuroinflammation?

We thank the reviewer for this comment. In this manuscript we are investigating CHI3L1/YKL40 as a potential marker for neuroinflammation. Our reasoning was that YKL40 is a reactive glycoprotein whose expression has been shown to be upregulated in correlation with inflammation in several neurodegenerative diseases. While we are aware of studies proposing that YKL-40 has been implicated in pro-inflammatory processes, in the brain or anywhere else, these studies seem to be few and mostly based on correlative suppositions. The scope of this paper was not to investigate the role of YKL40 in neuroinflammation, but merely to assess it as a potential biomarker for neuroinflammation in PB diseases. Even though it is unclear if and how YKL-40 might be driving neuroinflammation, YKL-40 expression and elevated levels are strongly correlated with disease

progression/pathogenesis/neuroinflammation and has therefore been proposed to be a potential biomarker for many neurodegenerative diseases, which now also include PB diseases. We ensured our manuscript does not make any unsupported claims on the role of YKL40 in neuroinflammation and clarified that YKL40 could potentially serve as monitoring biomarker, i.e., a tool to monitor the progression of neuroinflammation in PB diseases. Please note that we found Glc4 elevated in urine from APBD mice but not LD mice which is in line with a stronger peripheral involvement in APBD than in LD (at least at the adolescent age LD patients present with neurological problems). We highlighted urine Glc4 as a potential systemic biomarker for APBD in the Discussion.

3) The subject of neuroinflammation seems to be thematically detached from the rest of the manuscript, which characterizes alpha-glucan status in brains from the three transgenic strains and their metabolomic consequences. Can the authors try to integrate the two subjects in Discussion (i.e., neuroinflammation and MOGs)?

We thank the reviewer for this comment. We revised the discussion of neuroinflammation and MOG to integrate both subjects more overtly. We now state “Analysis of the relationship between MOGs, PB accumulation, and neuroinflammation on the individual animal level revealed a strong positive correlation of MOG content with the expression levels of neuroinflammatory marker genes *Cxcl10*, *Ccl5*, and *Lcn2*, i.e., with the severity of the neuroinflammatory response in both LD mouse models. This indicates that MOG production is linked to neuroinflammatory processes, but whether in a causative fashion or as a consequence remains unclear. It is conceivable that MOGs contribute to neuroinflammation as they are raising the intracellular levels of osmotically active molecules, which could elicit an osmotic stress signal that modulates inflammatory and immune responses, similar to extracellular hyperosmolarity (Brocker et al, 2012). Alternatively, MOG production could be the benign result of an attempt to reduce overlong glycogen chains in animals with more pronounced neuroinflammation. This would require neuroinflammation-stimulated endo-acting enzymes such as amylases that have access to cytosolic glycogen. Amylases are known as secreted enzymes, but recent studies have shown their implication in Alzheimer’s disease, their presence within astrocytes and neurons, and their potential association with glycogen degradation (Byman et al, 2021; Byman et al, 2018; Byman et al, 2019). An alternative mechanism to produce MOG could be glycogenin (GYG)-independent *de novo* synthesis by GYS1. Note, ultrafiltration showed that MOGs are smaller than 30kDa. Therefore, they must be GYG-free since GYG has a molecular weight of 37 kDa. However, GYG knockout mice accumulated glycogen molecules that were significantly larger than WT glycogen (Testoni et al., 2017) and hence much larger than MOGs. Investigations into the origin and exact function of MOGs as well as their role in neuroinflammation are ongoing.”

4) Are there serum samples available for metabolomic analysis from the mice whose brains were analyzed? Such an analysis and its integration with the brain analyses might generate some insight, which can connect the brain data with systemic metabolic pathways and shed light on key metabolic axes, such as the brain-liver axis. It can also suggest some systemic biomarkers. One of the issues within this discussion would be differential BBB permeability to different metabolites, so that the brain and blood can communicate, and the possible involvement of compromising BBB integrity by LKO, NKO, or Gbe1YS.

We appreciate this insightful comment. We did not collect serum for this study, but it could be interesting looking into this in the future. Even though not formally studied for LD and APBD, disruption of the blood-brain barrier has been consistently observed during the progression of epilepsies as well as neurodegenerative diseases such as Alzheimer's disease (AD), Parkinson's disease (PD), Amyotrophic lateral sclerosis (ALS), Multiple Sclerosis (MS), and Huntington's disease (HD). Therefore, it is possible that there is increased leakage of metabolites with disease progression in LD and APBD. If this is true, any metabolic signature that is found in the brain could possibly be detected in the blood as well. However, our data indicate that the metabolic changes we detected in the brain were independent of PB load, the widely accepted driver of the two diseases. We revised our statements in the Discussion to clarify that our conclusions are solely based on metabolic profiles measured in the brain.

5) Can comparative (univariate and multivariate) analyses be also made on brains from LKO vs MKO animals?

We appreciate the reviewer's suggestion. Metabolomic comparisons between LKO and MKO brains can technically be conducted using the data set we acquired. We have done so below. In the author's opinion the interpretation of this comparison is difficult, because MKO and LKO animals are not littermates but are generated from individually maintained mouse lines. To draw conclusions about the effects of LKO and MKO, comparisons have to be made with littermate WTs. As outlined in the method section we generated two sets of mice: (1) WT1, LKO, DKO1, PKO1, and (2) WT2, MKO, DKO2, PKO2. Within each set, animals are littermates and meaningful comparisons can be made. Comparing groups across the two sets (e.g. LKO vs. MKO) will always include changes that are mere line differences and that are not informative on the function of the disease proteins. For instance, the data below illustrates that there are changes even between WT1 and WT2. We therefore decided to restrict data in the paper to comparisons between littermates and would like to stress the importance of the use of littermates to assess 'true' metabolic changes especially when metabolic changes between genotype groups are moderate.

Significant changes in volcano plot

MKO vs. LKO				
	FC	log2(FC)	raw.pval	-LOG10(p)
Norvaline	2.0673	1.0477	6.50E-05	4.187
Oxoadipic acid	2.437	1.2851	0.000233	3.6318
4-Hydroxyphenyl-2-propionic acid	0.3835	-1.3827	0.000962	3.0169
Phenylactic acid	0.44275	-1.1753	0.001835	2.7355
Lumichrome	0.473	-1.0801	0.00294	2.5317
3-(2-Hydroxyphenyl)propanoic acid	0.4515	-1.1472	0.004571	2.34
2'-Deoxyguanosine 5'-monophosphate	0.35343	-1.5005	0.004683	2.3295
Propionylcholine	2.0135	1.0097	0.005675	2.2461
Hydroxykynurenine*	0.4958	-1.0122	0.00632	2.1993
L-Tyrosine	0.47925	-1.061	0.015017	1.8234
Phenylacetic acid	0.3394	-1.559	0.016373	1.7859
Hydroxyphenylacetic acid	0.43295	-1.2076	0.024193	1.6163
N-Acetyl-L-phenylalanine	0.41236	-1.278	0.026685	1.5737
Ethylmalonic acid	0.44101	-1.1811	0.039413	1.4044
L-Kynurenine	0.41392	-1.2726	0.042253	1.3741

*) also slightly downregulated in MKOvsWT

6) The authors mention mass tolerance and retention time as quality control criteria for mass spectrometry metabolomics analysis. Wasn't the metabolites fragmentation pattern and its comparison to that of standards used as another criterion?

We thank the reviewer for this question. As detailed in our method section, metabolites were identified based on a narrow mass tolerance of 10 ppm, retention time tolerance 0.5 min, and isotopic pattern. An in-house database with matching parameters was developed and validated for every detected metabolite using authentic standards. The Metabolomics Core Facility at Children's Medical Center Research Institute at UT Southwestern routinely uses this methodology to perform targeted metabolomics analysis on high-resolution mass spectrometry that routinely detects several hundred metabolites simultaneously.

7) For some key observations, the authors should consider presenting the results as biplots, co-plotting the scores of the different animals from each genotype and the loadings of the significant metabolites. This visualization would show how the metabolites correlate with each other, their relative contributions (according to vector length) and their PCA scores in the respective genotype. This form of presentation is usually quite informative. Please plot only significant metabolites because otherwise the visualization would be too cluttered.

As suggested by the reviewer, we now show biplots including the loadings of the 15 most significant metabolites for the PLSDA plots for all three mouse models (LKO, MKO, Gbe1YS).

8) While overall metabolomic changes in LKO and MKO were modest, I think that the authors should still discuss their implication on central carbon metabolism. Metabolites such as UDP-Glucose and Fructose-6-Phosphate, as well as the changes in glycogen mobilization caused by LKO, MKO, Gbe1YS, and Gbe1YS are bound to modify central carbon metabolism. The authors should therefore add in Discussion some speculation on the implications expected in glycolysis and its correlation with the TCA cycle, glucose-6-phosphate and its emanating pathways and more. Such an analysis might also suggest hypothetical biomarkers, especially systemic ones.

We thank the reviewer for this suggestion. While overall metabolomic changes are modest in brains of LKO, MKO, and Gbe1YS, we interrogated our data with respect to a potential impact on central carbon metabolism, with the prospect of identifying a potential metabolic biomarker for PB diseases. Individual metabolites in glycolysis, pentose phosphate pathway, and tricarboxylic acid (TCA) cycle were slightly decreased, but not consistently between the three PB disease models. For instance, citrate and 6-phosphogluconate were slightly decreased in LKO and MKO, but not in Gbe1YS. Erythrose 4-phosphate and G6P were decreased in LKO, but not in MKO or Gbe1YS. 3-phosphoglyceric acid was slightly decreased in MKO, but not in LKO or Gbe1YS. On the pathway level, TCA cycle, pentose phosphate pathway, and glycolysis do not seem to be significantly affected in the single mutants (LKO, MKO, Gbe1YS, PKO). However, they seem to be impacted in the DKO mice. Here, also some of the emanating pathways show decreases in metabolites, including most prominently several in amino acid metabolism that feed off of glycolysis or TCA cycle. This suggests a connection of laforin and malin, and PTG with flux in central carbon metabolism in the brain. However, these pathway changes may not be the result of the pathological changes in glycogen metabolism because of the inconsistent display of these changes in the different PB disease models. This includes the subtle changes in individual metabolites in the single mutants, which are independent of PB accumulation, and may not allow for a sensitive PB disease specific biomarker among the metabolites of central carbon metabolism. As requested, we added these considerations to the discussion.

Minor points

1) p. 2: Please change to "An imbalance between the activity of both enzymes"

Done.

2) p. 3: "In the absence of evidence for a dysregulation of glycogen chain elongation as an explanation for glycogen insolubility in LD,." Isn't misregulated chain elongation presented as a cause for glycogen insolubility in PMID: 31042462 ?

We completely agree that the wording is misleading and thank the reviewer for pointing this out.

We changed the end of this paragraph as follows: Despite the clear evidence that increased glycogen chain length is the cause for glycogen insolubility (Nitschke *et al*, 2017; Sullivan *et al*, 2019), it is unclear what causes the abnormal glycogen structure in LD as no differences in the expression or activity of

glycogen-metabolizing enzymes could be found to date. FM brain fixation prevents postmortem processes, which could alter the metabolic state and/or enzyme activities potentially eliminating the disease-related changes that caused the shift in chain length. Therefore, brain glycogen levels, GYS1 phosphorylation, and the LD brain metabolome need to be revisited in FM-fixed mouse brains.

3) p. 3: "By modulating GYS1 activity through PTG knockout, these phenotypes can be studied in the presence and absence of PBs." Why not opting for a direct inhibition of GYS1, e.g., using CRISPR, or shRNA?

In order to gain as much mechanistic insight as possible, we wanted the effect on PB formation to be as complete as possible. PTG knockout (apart from GYS1 knockout) has been shown to be the most complete in its effect to inhibit PB formation, while GYS1 knockdown approaches only achieved intermediate effects. The discreteness of the two states (with PBs, without PBs) made PTG-KO appear to be the better tool to study the effects of PBs on metabolism and neuroinflammation. Targeting *Gys1* with AAV9-CRISPR-Cas9 is technically a possibility, but currently the AAV9 technology only allows for a limited distribution of the genetic cargo (in this case CRISPR-Cas9) throughout the brain (Gumusgoz, 2021, PMID: 33830476). GYS1 knockout would have been another option but with only 10% of the homozygous GYS1-KO pups surviving birth this approach was not very practical, whereas mice heterozygous for GYS1-KO show only a partial rescue. Moreover, GYS1-KO completely prevents glycogen synthesis and makes it impossible to study glycogen levels or structure in the rescued LD mice.

4) p. 3: "Allosteric binding of glucose 6-phosphate (G6P) overrides the phosphorylation-imposed inhibition, restoring full GYS1 activity (Hunter et al, 2015)." - Please also cite Bouskila et al (2010) Cell Metab. 12:456-66. doi: 10.1016/j.cmet.2010.10.006.

Done.

4) p. 4: Do you know what is the glycogen degradation status in PFA fixed tissues?

We have never formally studied it. From our experience with post-mortem glycogen degradation, we noticed that degradation rates are tissue, time, and temperature dependent (brain faster than muscle, increasing with time to full metabolic arrest, less in the cold). When brains are dissected after cervical dislocation and fixed in PFA, we anticipate 30-40 % of soluble glycogen to be degraded in the first 45s after cervical dislocation. IF the brain is immersed in PFA after 45s, there might further be a fixation delay due to slow PFA permeation depending on the volume of the tissue piece. Taken together it is reasonable to assume that soluble glycogen is significantly degraded in PFA-fixed tissue. However, it is obvious that the PASD positive polyglucosan, i.e. the insoluble glycogen fraction, is resistant to post-mortem degradation.

5) p. 4: How long is the pre-incubation time?

For brain 1h. We double checked that it is clearly mentioned in the Methods and even added this information on an appropriate second occasion.

5) p. 5: "We identified 36 and 32 metabolites that were significantly altered in LKO (Fig. 3A) and MKO (Fig. 3B), respectively, with no overlap between the two LD mouse models regarding metabolites with fold changes (FC) of >1.25 or <0.8 (Fig. 3A-B)." Were p values in the volcano plots adjusted for false negatives? If so, what were the FDR (q) values? Usually $FDR < 0.2$ is acceptable.

The volcano plots display raw p values. However, we now also indicate the suggested FDR threshold ($q=0.2$) in all volcano plots (Fig. 3A-B, Fig. 5F). This emphasized even more that metabolic changes in MKO brains are less pronounced than in LKO.

6) p. 5: "Metabolic profiles of LKO/MKO and respective WTs clustered on a PLS-DA plot, though less strictly for MKO and WT (Fig. 3C)." Has PLS-DA been tested for forced classification (for instance by a permutation analysis)?

We conducted permutation analyses, which delivered empirical p value > 0.5 , indicating that classification in our PLS-DA plots is likely forced. For this reason, we also added unsupervised PCA plots in the supplement. These PCA plots show no multivariate differentiation between mutants and WT of all three PB disease models we studied in this publication, meaning on a large scale the differences are small. We clarified this in the text.

7) p. 5: "with on average lower VIP scores in MKO" This difference does not clearly show in figure 3D. VIP scores for both LKO and MKO range from 1.7 to 2.3."

We changed the figure to include a dashed line indicating the average VIP score of the displayed 15 most significant metabolites to better visualize the difference.

8) p. 5: "The significantly changed metabolites in Brewer et al. were found unchanged in our study (Fig. EV2A). Furthermore, of the 20 significantly changed pathways in Brewer et al., we found only three changed (Fig. EV2B). Metabolic changes previously found in MKO (Duran et al., 2021) were not detected in our study (Fig. EV2C)." Can these apparent discrepancies be explained?

We thank the reviewer for this question. Analyses of metabolic processes in the brain have been shown to be confounded by rapid post-mortem metabolic processes (Ponten et al, 1973; Wu et al, 2019). Post-mortem changes in brain metabolism can be avoided by using high-power focused microwave (FM) fixation systems that inactivate all enzymatic activity in the brain within approximately one second. Recently, the comparison of metabolomes from CP- and FM-fixed mouse brain tissue showed that postmortem several metabolites and associated pathways change within seconds, with drastic changes observed in glucose, glucose 6-phosphate, and glycogen pools (Juras et al, 2023). The previous metabolomic analyses were conducted in cryopreserved brain tissue. We would argue that previously seen metabolic changes may, at least in part, be due to metabolic reprogramming in association to death-associated hypoxia. Another potential reason for the absence of previously reported metabolic changes in our data set may lie in our strict littermate comparison approach. Especially when metabolic changes between genotype groups are moderate, which they also were in previous reports, the use of

littermates to assess 'true' genotype-related metabolic changes is of utmost importance. The use of littermates for metabolomic studies in Brewer et al. (2019) and Duran et al. (2021) has not been specified.

9) Figure 5A-D: Why does figure show results from WT mice? Not clear how this is related to glycogen post-mortem preservation, which should be analyzed in Gbeys/ys.

Figure 5A-D includes both Gbe1YS mutant mice and littermate WT mice for comparison. Post-mortem glycogen preservation by microwave fixation (FM, as compared to cryopreservation CP) can be deduced from slightly higher glycogen levels in panel A (FM) vs. panel B (CP). In addition, we show in Fig. EV4A that brain glycogen in WT littermates of the Gbe1YS cohort is depleted by pre-incubation when brains have been cryopreserved (CP), but glycogen levels are unaffected by pre-incubation when brains have been fixed by focused microwave (FM). Please note, the mutants accumulate degradation-resistant glycogen (insoluble glycogen) and degradable (soluble) glycogen. The degradation-resistant glycogen is not affected by the fixation method (see also Figs. 5C and 6A-B). That means to assess the post-mortem glycogen preservation in a cohort of mice soluble glycogen levels are the more appropriate indicator. WT mice do not significantly accumulate insoluble glycogen and are therefore the better subjects to study post-mortem degradation of glycogen after different fixation methods.

10) Fig. EV3B-G: Are these all the metabolites that participate in the respective pathway(s), or only the ones identified? Sometimes a small percentage of all metabolites in specific pathways are identified.

Pathways displayed in Fig. EV3B-U only include metabolites that were detected in our metabolomic assay. We have included this information in each panel description. Also, we added the percentage of detected metabolites from each pathway in each panel of Fig. EV3.

11) p. 6: "We confirmed strong accumulation of insoluble glycogen in these mice (Fig. 5A-C) with better glycogen preservation (Fig. 5A-B) and complete pre-incubation resistance after FM fixation"

We changed the sentence above as follows to clarify and additionally consider the reviewer's point 9: "We confirmed strong accumulation of insoluble glycogen in these mice (Fig. 5A-C). To evaluate the success of FM fixation in this cohort we specifically looked at non-disease WT mice which do not accumulate degradation-resistant insoluble glycogen. Not only did the WT mice have 35% higher glycogen levels after FM (Fig. 5A) compared to CP fixation (Fig. 5B), glycogen levels in the FM-fixed WT brains also remained unchanged during pre-incubation while completely depleted in CP-fixed WT brains (Fig. EV4A). This confirms enzymatic inactivation through focused microwave irradiation."

12) p. 6: "driving the clustering plotted". I would rephrase to "contributing to metabolic variability".

We agree that the wording is not optimal and changed the sentence as follows: "The metabolic profiles of Gbe1YS versus WT clustered on a PLS-DA biplot (Fig. 5G) with metabolites influencing this projection the most shown with VIP scores in Fig. 5H."

13) p. 9: "In aged *Gbe1*^{YS} mice the P-GYS1/GYS1 ratio is even more dramatically decreased (Fig. 7E), further corroborating the direct link between glycogen chain length abnormality in PB diseases and the phosphorylation state of the chain-elongating enzyme GYS1." Please note that a decrease in p-GYS/GYS and subsequent increase in GYS activity was also observed in a GBE knock-down neuronal model of APBD (Kakhlon et al (2013) J Neurochem 127: 101). Please add this reference.

We thank the reviewer for mentioning this point. We agree. We changed the sentence above as follows: "In aged *Gbe1*^{YS} mice the P-GYS1/GYS1 ratio is even more dramatically decreased (Fig. 7E) in line with a study by Kakhlon et al. (2013) that showed an increased GYS1 activity in a *Gbe1*-deficient neuronal cell model [Kakhlon et al (2013) J Neurochem 127: 101]. The finding that all three aged PB disease models exhibit this reduced phosphorylation state in largely PB-associated GYS1 suggests that overactivation of GYS1 at eventually precipitating glycogen molecules is a major determinant of the glycogen chain length abnormality seen in PBs of LD and APBD."

14) p. 10: "In LD mice, additional characteristics such as glycogen hyperphosphorylation, neuroinflammation, and metabolic alterations have been described (Brewer et al., 2019; Duran et al., 2021; Lahuerta et al., 2020; Tagliabracci et al., 2008).

We thank the reviewer for highlighting this sentence. We aimed to focus on the findings in the LD model, but now reworded the sentence slightly to avoid the impression that in other PB disease models (such as APBD) no additional characteristics were described. We changed the sentence as follows: "Additional characteristics have been described, which in LD mice include glycogen hyperphosphorylation, neuroinflammation, and metabolic alterations (Brewer et al., 2019; Duran et al., 2021; Lahuerta et al., 2020; Tagliabracci et al., 2008)."

15) p. 10: "indicating a globally increased activation state of GYS1 (Hunter et al., 2011; Lin et al., 2012) solely in LKO, which could be explained by..." I didn't understand how then can the authors explain GYS1 activation solely in LKO and not in MKO. The two scenarios (GSK3 β and PTG activity) are basically ruled out. Is there an alternative explanation?

We thank the reviewer for this question. It is true that global GYS1 activation can only be detected in 5-week-old LKO, not MKO. Therefore, we argue that only laforin impacts global GYS1 phosphorylation, leading to less GYS1 phosphorylation in laforin absence. Laforin deficiency would lead to less GSK3 β dephosphorylation (Lohi et al. 2005, Wang et al., 2006), decreasing the net activity of GSK3 β , and reducing the phosphorylation of GYS1 (hence GYS1 activation). The fact, however, that malin-deficient mice accumulate PBs in the brain to the same extent as laforin-deficient mice, questions that the slight change in GYS1 activation seen in young LKO (Fig. 7A) is causative of PB accumulation. However, in both LD models it appears that GYS1 bound to mature PBs is or has been more active than the average soluble GYS1 in age-matched WT mice. This could indicate an overactivation of GYS1 on those few glycogen molecules that later succumbed to precipitation, conserving the occasionally occurring, locally high GYS1 activation state in the precipitated PBs. This is consistent with the theory that laforin binds malin and targets it to particular "at-risk" glycogen molecules through its CBM domain. These molecules

for stochastic reasons may be burdened with overactive GYS1 which induces long chains. The laforin-malin complex may locally reduce PTG levels in these “at-risk” molecules to locally decrease GYS1 activity through reduced PP1-mediated GYS1 dephosphorylation. In the absence of laforin or malin, malin is not targeted to “at-risk” molecules and glycogen synthesis of these molecules continues with overactive GYS1, ultimately leading to precipitation and conservation of the higher GYS1 activation state in the precipitated glycogen. We revised the corresponding sections in the Discussion to clarify our conclusions.

16) p. 11: "there is no common metabolic signature in PB diseases." This conclusion might be true for the brain. However, systemic metabolic signature of PBs might still take place, which might be disclosed why serum metabolomics.

We agree with the reviewer and have changed the statement so that it is limited to brain metabolism: “... there seems to be no common metabolic signature in the brains of the studied PB disease mouse models.”

17) p. 11: "Other promising approaches include EPM2A gene replacement and removal of PBs by bioengineered hydrolases (Brewer et al., 2019; Zafra-Puerta et al, 2023)". Please also add here Kakhlon et al (2021) EMBO Mol Med (see above) - for APBD compassionate use of a newly discovered safe autophagic activator with an FDA orphan designation also mentioned in Wu et al (2024) EMBO Mol Med (see above).

We changed the passage as follows: “Other promising approaches include *EPM2A* gene replacement, removal of PBs by bioengineered hydrolases, and the use of an autophagic activator with an FDA orphan designation for the treatment of APBD (Brewer *et al.*, 2019; Zafra-Puerta *et al.*, 2023; Kakhlon *et al.*, 2021).

Referee #2:

In this work, Nitschke et al. have analyzed the effect of PTG k.o. on glycogen structure and level of phosphorylation. Relate to those analysis, the use of a novel reported fixation method has prevented brain glycogen post-mortem degradation, occurring when other methods have been used and more reliable results were obtained. In addition, they have analyzed the consequences of PTG depletion or brain metabolome or in the presence of some neuroinflammatory markers. The studies were done on Lafora Disease (LD) mouse models. The title is clear but other parts of the article are less clear and should be further clarified.

Specific points:

- In the Introduction, for the general audience, malto oligoglucans should be introduced.

We thank the reviewer for this helpful suggestion. We are the first to describe malto-oligoglucans as a metabolite in the eukaryotic context. Malto-oligoglucans have been described in bacteria before. We now show evidence for the existence of this metabolite in the mammalian brain in association with

neurodegenerative disease. We have added a short description of malto-oligoglucans in the Introduction.

- Figure 1 shows that PTG k.o. prevents glycogen accumulation in LD mouse models. This is not a novel observation, as indicated in the Introduction. Thus, the effect of PTG k.o. was focused on glycogen chain length (related to insolubility) and hyperphosphorylation. The obtained results confirm the previous observations found in a mouse with a defective laforin, indicating that hyperphosphorylation and insolubility are uncoupled (see Gayarre 20014 or the work of the authors of this submitted manuscript). In summary, this is a confirmatory data, that perhaps could be shortened in part.

We agree that our glycogen data in Fig. 1C-D confirm previous results (Turnbull et al., 2011; Turnbull et al., 2014). Hence, we accept the reviewer's suggestion and shortened the section on glycogen accumulation.

Regarding our chain length and phosphate results: Even though chain length and phosphate levels have been determined in LKO and MKO single knockout mice before and are therefore confirmatory data, they are the basis to evaluate the novel results in DKO and PKO mice, in which these types of analyses were not performed previously. In the respective paragraph we made sure that it is clear which part of the data is confirmatory and which part is new (by using phrases such as: "as expected", "as previously shown", "further corroborating previous findings"). We are glad that the reviewer agrees with us to conclude that "glycogen hyperphosphorylation and insolubility are uncoupled". However, this topic is still disputed in the field. Therefore, we find it important to sufficiently explain our data which show once again that hyperphosphorylation does not necessarily cause glycogen insolubility, especially when glucan chains are short. We thereby corroborated the findings by Gayarre et al. (2014) and Nitschke et al. (2017), yet by using an independent (genetic) approach.

- Figure 2 shows that PTG k.o. prevents LD-related neuroinflammation and analysis of pro and anti-inflammatory genes were carried out. The analyses were properly performed but the conclusion is mainly descriptive. It may require further explanations for a possible mechanism.

We thank the reviewer for raising this point. We appreciate that the reviewer would like to see a mechanistic explanation of the inflammatory processes in LD. However, in the scope of this work we tried to investigate the cause-effect relationship between glycogen chain elongation, glycogen branching, glycogen solubility, glycogen phosphorylation, metabolic dysregulation, and neuroinflammation. For that we needed a thorough approach to assess the extent of neuroinflammation. Therefore, we chose 9 inflammatory response genes, whose expression we measured in all our genotype groups. With this approach we undoubtedly show that neuroinflammation is not a direct consequence of laforin or malin deficiency, respectively, but depends on the formation of PBs and/or MOG (see below), which results from loss of function of one of both LD genes. Based on this solid finding we go on to show that the previously described metabolic phenotype in LD is independent of neuroinflammation. We agree that further work is necessary to establish the molecular underpinnings of the inflammatory response toward the insoluble glycogen and/or MOG. We revised the respective section in the Results and Discussion to clarify our objective.

- Figure 3-Figure 6 are mainly focused on the obtained results using the reported fixation method and it has facilitated the comparison with the previous data from other groups. This is a positive point. However, there are many "crude" data. For example, Figure 3A shows the "crude" data for 20 metabolites. Figure 3B shows that for 19 metabolites. Figure 4A indicates 29 different pathways. A clearer example could be found in Figure 4I. It indicates pathway A and pathway B. Nothing in the text (mentioned six pathways) or in the legend of the Figure is indicated about the meaning of pathway A and pathway B.

We appreciate the reviewer's feedback. For metabolomic studies it is common to show volcano plots that include all data points in a compact way and highlight those metabolites that might be of interest (e.g. Wang et al., *Nat Commun* 15, 3562 (2024); Lopez-Hernandez et al., *Sci Rep* 13, 12420 (2023)). For full transparency and best comparison between the mouse lines, we found it necessary to apply an objective criterion that dictates whether or not a data point is highlighted. We chose fold change (FC) and p-value cutoffs as explained in the figure, which pointed out 20 metabolites in LKO (Fig. 3A) and 19 in MKO (Fig. 3B). We now also include information on how many of these data points withstand adjustment for false-discovery rate (FDR). Together, Fig. 3A and B serve to illustrate that a few significantly changed metabolites can be identified in both mouse lines, but that there is essentially no overlap between individual metabolites. Moreover, in MKO fewer metabolite changes seem to occur as illustrated by the lower number of metabolites with $q < 0.2$, the lower VIP scores of the metabolites driving the PLS-DA projection, and the absence of any significantly changed pathways in MKO. All of these approaches combined support the notion that there is no common metabolic fingerprint that is present in the brains of both models of LD.

The 29 pathways indicated in Fig. 4A can be understood as a legend, explaining the numbering of pathways throughout the entire Figure 4 (now clarified in figure legend). They become relevant when PTG is knocked out in addition to laforin or malin (DKO1 vs. WT1, DKO2 vs. WT2). While PTG knockout by itself does not elicit substantial dysregulation of pathways (Fig. 4B and C), the combination of PTG knockout with laforin or malin KO does impact several pathways significantly. Interestingly, more so in MKO than in LKO. It seems that PTG knockout 'unmasks' and exacerbates metabolic dysregulation otherwise only mild (LKO) or absent (MKO) in the presence of PTG. Our vision for this paper is to provide a comprehensive comparative study of metabolic changes and lay the foundation for research that follows up on the mechanistic underpinnings of how PTG and laforin or malin control individual metabolic pathways. To this end, here we focused on the bigger picture and tried to be inclusive of all pathways and metabolites that undergo changes. For dysregulated pathways overlapping between different genotype comparisons, we wanted to estimate direction and severity. For that we calculated the mean area in the volcano plot for all detected metabolites in the corresponding pathways. This method is illustrated in Fig. 4I using two hypothetical pathways (A and B). For both pathways 4 metabolites have hypothetically been detected and plotted on a volcano plot comparing two hypothetical groups of mice. The mean area in the volcano plot of the 4 metabolites in pathway A is negative, suggesting pathway A is downregulated. The mean volcano area of the 4 metabolites in pathway B is positive, suggesting pathway B is upregulated. In Fig. 4J we compared mean volcano areas of the six pathways changed in LKO vs. WT, with those of the same pathways comparing DKO1 vs. WT. All six pathways have the same directionality and are in most cases more severely affected in the double-KO of laforin and PTG as compared to when only laforin is knocked out. Fig. 4K compares the

effects of malin or laforin knockout in the absence of PTG. Pathway directionality is similar for most pathways and malin-deficiency largely lead to increased severity of the pathway change, as compared to laforin-deficiency. We had included a description of the calculation in the Methods section but have now included even more detailed information in the Method text and clarified our explanations in the corresponding figure legend.

I suggest clearer conclusions rather than quantity or too many "unexplained" data.

We thank the reviewer for this suggestion. We have revised the manuscript to ensure presence and improve clarity of conclusions at the end of each section of the Results part. Also, we are now including a scheme that illustrates the main conclusions of the paper (Fig. 8).

- In summary, the article could be improved by making a clear scheme with the main (not many) conclusions, showing (end of the Discussion) one by one, the cause-effect relationship between: a) Glycogen chain elongation, b) glycogen chain branching, c) glycogen solubility, d) glycogen phosphorylation with A) metabolic dysregulation or B) neuroinflammation, like markers of LD. Now that cause-effect relationship is slightly confused and should be clarified.

We thank the reviewer for this suggestion. We are now including a scheme that illustrates and integrates the main conclusions of the paper (Fig. 8).

Minor points:

- The analyses showing the level of GSK3 β P-Ser 9, like a control, could be suitable marker after using this novel method.

Laforin's action as a phosphatase on GSK3 β (removal of inhibitory phosphate on Ser 9) has been disputed in the LD field. Two groups have shown data that indicate that laforin is a GSK3 β phosphatase (Lohi, 2005; Wang, 2006) while two other groups have either found no evidence of laforin's action on GSK3 β (Worby, 2006) or no change in GSK3 β activation state in laforin knockout mice (Tagliabracci, 2007). Together, the line of evidence for GSK3 β as a laforin target is controversial and therefore we do not think that P-Ser9 GSK3 β levels can serve as a solid control or a tool to validate our western blots using FM-fixed brain tissue.

Furthermore, decreased GSK3 β activity due to lack of laforin action is unlikely the cause of PB formation in LD as it would only explain PB formation in laforin-deficient LD mice. However, both LD mouse models (aged LKO and MKO) showed a comparable change in P-GYS1/GYS1 ratios (Fig. 7C-D), indistinguishable levels of insoluble glycogen (Fig. EV5A-B) and comparable chain length abnormality (Nitschke et al., 2017). Moreover, mouse models expressing phosphatase-inactive laforin instead of WT laforin do not accumulate polyglucosan bodies (Gayarre et al., 2014; Nitschke et al., 2017; Skurat et al., 2024) which proves that laforin's phosphatase function is not needed to prevent PB formation. We would further argue that it is unlikely that the reason for the increased GYS1 activation state is necessarily the same in all three instances (LKO, MKO, and Gbe1YS). Even though we think it will be interesting to study the reason for reduced phospho-GYS1 in the future, we feel this is beyond the scope of this study. There are several possible scenarios, including reduced GYS1 phosphorylation through kinases, increased GYS1

dephosphorylation through PTG or other PP1-targeting subunits, or stochastic presence or absence of lowly phosphorylated (active) GYS1 on at-risk glycogen molecules. Getting a definite answer will be a study of its own.

Referee #3:

Polyglucosan body (PB) diseases are clinically and genetically heterogeneous diseases that cannot be classified easily to one gene or clinical phenotype. However, these very diverse diseases have one in common Diastase resistant PAS positive carbohydrate inclusions in muscle or brain. This diversity makes it difficult to diagnose and treat those diseases. GBE1 deficiency is the only form of PB disease with a very well understood mechanism yet degree of enzyme deficiency creates a wide spectrum of clinical presentation that requires careful examination and diagnostic workup. Despite the well characterization, diagnosis, treatment, and easily detectable biomarkers are not available for patients. Dr Nitschke et al, compared the mouse models of Pb diseases and affected molecular and cellular pathways using different mouse models that are shown to be related to glycogen metabolism. Their findings identified oligoglucans that explain the carbohydrate storage and use problems in the disease state. Like normal glycogen these molecules disappear quickly when tissue is devoid of blood or supply. Therefore, Nitschke et al used a microwave mediated sacrifice which instantaneously fixes tissue and proteins preventing further metabolic changes and degradation. Methods they used are reliable and experiments are carefully designed and executed. Detection of oligoglucans and their role in the disease are impressive and can be a very good source for determining the efficiency of the treatments and the stage of the disease when a treatment method becomes available. However, I have concerns about the CHI3L1/YKL40, these markers are very common biomarkers and may not reflect the state of the disease especially in the advanced age of the patients. Considering how common these two markers are, authors must limit the use of these markers to pure polyglucosan body diseases and mention them in the text that they may cause misdiagnosis when combined with other conditions including but not limited to infections and other neurodegenerative diseases.

We agree with the reviewer that CHI3L1/YKL40 by no means can serve as diagnostic biomarker, not only because of it being impacted by age but also because of its upregulation in other conditions with involvement of inflammation. As mentioned by the reviewer the age likely matters and is very different between LD and ABPD patients. Therefore, it will be important that CHI3L1/YKL-40 levels in patients are always compared to age-matched controls. We still think CHI3L1/YKL40 could potentially serve as monitoring biomarker to evaluate disease progression and/or therapeutic efficacy of any treatment in clinical trials. However, it will be crucial to record baseline measurements at disease onset/diagnosis and before the administration of a new therapeutic agent, respectively, for each individual patient. We revised our statements regarding the use of CHI3L1/YKL-40 as a biomarker to clarify that it could potentially serve as monitoring biomarker, i.e., a tool to monitor the progression of neuroinflammation in PB diseases and suggest further investigation.

Dear Prof. Nitschke,

Thank you again for submitting your revised manuscript (EMBOJ-2024-117757R) to The EMBO Journal for our consideration. It has now been seen by the three referees who previously assessed the original version of your manuscript (their comments are included below), and I am glad to say that they are satisfied with the revision, find all previously raised concerns satisfactorily addressed, and now support publication of the manuscript. I am thus happy to say that your manuscript has been in principle accepted for publication in The EMBO Journal. Congratulations on an excellent work and thank you for your contribution to our journal!

There are a few minor formatting/editorial changes that we need from you in a final version of your manuscript before we can proceed with the next steps of typesetting and production of proofs for your approval:

- The author contributions statement should be removed from the manuscript file. Instead, we use CRediT to specify the contributions of each author in the journal submission system. Please feel free to use the free text box to provide more detailed descriptions during submission. See also our guide to authors for more information: <https://www.embopress.org/page/journal/14602075/authorguide#authorshippinguidelines>.
- Please incorporate the information about genotyping and qPCR primers in your Reagents and Tools table (section: "Oligonucleotides and other sequence-based reagents"), instead of uploading them as "Appendix Table S1" and "Appendix Table S2" in an Appendix. Please update accordingly all callouts throughout the manuscript if necessary.
- Please remove "Supplementary Material: Appendix Tables S1 and Table S2" from your manuscript.
- Please remove your Reagents and Tools table from your main manuscript file, and only upload it to our manuscript tracking system as a separate file (as a "Reagent Table" file).
- Please re-organize your uploaded Source Data in one (zip) folder per Figure. For example, all Source Data for Figure 1 panels should be contained in a single zip folder named "SD Figure 1.zip" etc. Please ZIP together all Source data for EV Figures in an "SD EV Figures.zip" folder.
- During our routine pre-acceptance checks, our data editors have raised the following queries regarding figures, data, and legends. Please completely address all requests in the final version of your manuscript:
 1. Please provide the exact p values in the legends of Figures 1a, c-g; 2a-l; 5a-e, j-m; 6a-h; 7a, c-e; EV 4a, h; EV 5a-b, d-i; EV 6a-b.
 2. Please indicate the statistical test used for data analysis in the legends of Figures 3a-b; 4i-k; 5f; 6c-d; EV 1c-d; EV 2a, c; EV 3b-u; EV 4c-e.
 3. Please note that information related to "n" is missing in the legends of Figures EV 4d-e.
 4. Although "n" is provided, please describe the nature of entity for "n" in the legends of Figures 1c-d, g; 4i; 5d-e, j-m.

Please also note that as part of the EMBO publications' Transparent Editorial Process, The EMBO Journal publishes online a Peer Review File along with each accepted manuscript. This File will be published in conjunction with your paper and will include the referee reports, your point-by-point response and all pertinent correspondence relating to the manuscript. You can opt out of this by letting the editorial office know (contact@embojournal.org). If you do opt out, the Peer Review File link will point to the following statement: "No Peer Review File is available with this article, as the authors have chosen not to make the review process public in this case."

We look forward to seeing a final version of your manuscript as soon as possible. Please let us know if you have any questions and use this link to submit your revision: <https://emboj.msubmit.net/cgi-bin/main.plex>

Best regards,

Ioannis

Ioannis Papaioannou, PhD
Editor, The EMBO Journal

Referee #1:

The authors have adequately addressed all my comments and I recommend acceptance of this manuscript.

Referee #2:

The authors have address all the points raised by this reviewer. The work is acceptable for publication

Referee #3:

Dr. Nitschke and colleagues identified molecules associated with polyglucosan accumulation and related diseases. Using a technique that rapidly fixed tissues to preserve these metabolites and markers, With this method, they uncovered a pool of metabolically volatile malto-oligoglucans (MOGs). These MOGs were characterized as potential PB- and neuroinflammation-associated brain energy sources. The study also identified promising biomarker candidates for Lafora disease (LD) and adult polyglucosan body disease (APBD), in addition to the MOGs, a neurodegeneration marker CHI3L1/YKL40. Manuscript has been improved after addition of the comments and explanations of the reviewers.

All editorial and formatting issues were resolved by the authors.

Dear Felix,

Congratulations on an excellent manuscript! I am very pleased to inform you that it has been accepted for publication in The EMBO Journal. Thank you for your comprehensive responses to the referees' comments and for addressing all editorial and formatting requests.

If you have any questions, please do not hesitate to contact the Editorial Office. Thank you for your contribution to The EMBO Journal. Working with you has been a pleasure!

Best regards,

Ioannis
